

# AgriCarbon-EO v1.0.1: Large Scale and High Resolution Simulation of Carbon Fluxes by Assimilation of Sentinel-2 and Landsat-8 Reflectances using a Bayesian approach

Taeken Wijmer[1,2,*], Ahmad Al Bitar[1,*], Ludovic Arnaud[1], Remy Fieuzal[1], and Eric Ceschia[1]

[1]CESBIO, Université de Toulouse, CNES/CNRS/INRAE/IRD/UPS, 18 Avenue Edouard Belin, bpi 2801, CEDEX 09, 31401 Toulouse, France
[2]DYNAFOR, Université de Toulouse, INRAE, INPT, INP-PURPAN, Castanet-Tolosan, France

**Correspondence:** Taeken Wijmer (taeken.wijmer@inrae.fr) and Ahmad Al Bitar (ahmad.albitar@gmx.com)

**Abstract.** Soil carbon storage is a well identified climate change mitigation solution. The extensive in-situ monitoring of the soil carbon storage in cropland for agricultural policy and offset carbon markets is prohibitive, especially at intra-field scale. For this reason, comprehensive Monitoring, Reporting and Verification (MRV) of soil carbon and its explanatory variables at large scale needs to rely on remote sensing and modelling tools that provide the spatio-temporal dynamics of the carbon budget and it's components at high resolution with associated uncertainties. In this paper, we present AgriCarbon-EO v1.0:

an end-to-end processing chain that enables the estimation of carbon budget components of major crops and cover crops at intra-field resolution (10 m) and large scale (over 110×110 km) by assimilating remote sensing data in physically-based radiative transfert and agronomic models. The data assimilation in AgriCarbon-EO is based on a novel Bayesian approach that combines Normalised Importance Sampling (NIS) and Look-Up Table (LUT) generation. This approach propagates the

uncertainties across the processing chain from the reflectances to the output variables. The chain considers as input a land cover maps, multi-spectral reflectance maps from the Sentinel-2 and Landsat-8 satellites, and daily weather forcing. The PROSAIL radiative transfer model is inversed in a first step to obtain Green Leaf Area Index ($GLAI$). The $GLAI$ time series are then assimilated into the SAFYE-CO2 crop model taking into consideration their uncertainty. The chain is applied over winter wheat in the south-west of France during the cropping seasons 2017 and 2019. We compare the results agaisnt the net ecosystem

exchange measured at the FR-AUR ICOS site (RMSE = 1.69 - 2.4 $\mathrm{gCm^{-2}}$, $R^2$ = 0.88 - 0.88), biomass (RMSE = 250 $\mathrm{g\,m^{-2}}$, $R^2$ = 0.9), and combine harvester yield maps. We quantify the difference between pixel and field and pixel scale simulations of biomass (bias = -47 $\mathrm{g\,m^{-2}}$, -39 % variability), and the impact of the number of remote sensing acquisitions on the outputs (-66 % of mean uncertainty of biomass). Finally, we conduct a coherency analysis at regional scale to test the consistency of the observed patterns with soil texture, altitude and exposition variability. Results show higher biomass for higher clay soils

and earlier emergence and senescence for south western exposition.

Keywords: MRV; Carbon Farming; crop modelling; Sentinel-2; Normalised Importance Sampling



## 1 Introduction

Agriculture and land use changes accounts for 15% *ie* (8.7 Gt $CO_2$ $yr^{-1}$) of human induced Green House Gas (GHG) emis-
sions (Pörtner et al., 2022; Skea et al., 2022). On the other hand, agriculture has also been identified as a sector where climate
mitigation solutions can be implemented (Porter et al., 2017; Matthews et al., 2022). Among those solutions, soil carbon stor-
age has the potential to remove 0.6 to 9.3 Gt $CO_2$ $yr^{-1}$) from the atmosphere through the implementation of carbon farming
practices worldwide (Skea et al., 2022). For cropland, soil carbon storage can be achieved by the addition of cover crops in
crop rotations (Poeplau and Don, 2015; Lugato et al., 2020), reduced tillage (Haddaway et al., 2017), application of organic
amendments (Vereecken et al., 2016) and biochar (Steinbeiss et al., 2009). Moreover, organic carbon storage in agricultural
soils has additional benefits in terms of Ecosystem Soil Services (ESS) like increasing soil fertility (Su et al., 2006), enhanced
water holding capacity (Karhu et al., 2011) or higher biodiversity (Wall et al., 2015). Soil Organic Carbon (SOC) storage could
also account for an additional source of revenue for farmers through carbon credits and subsidies.

To assess the amounts of sequestered carbon as well as the impact on agro-ecosystems an objective and reliable quantifica-
tion of the carbon budget and crop growth variables is needed. Following the Intergovernmental Panel on Climate Change
guidelines for national GHG inventories, methodologies for assessing SOC stock changes and GHG emissions have been de-
veloped. They are based on a tiered approach with increasing complexity involving activity and soil data compilation up to soil
monitoring networks and process-based modelling (e.g. Yasso07 in Finland, RothC in Japan, DayCent in the USA) tailored
to national context. The need to monitor soil carbon at Farm level to inform individual farmers, guide policies and develop-
ment of carbon markets led to the development of Monitoring Reporting and Verification (MRV) schemes based on similar
aproaches at a higher resolution (Smith et al., 2020; Paustian et al., 2019). Those approaches are mainly used in carbon farming
projects following national or regional initiatives (e.g. Label Bas Carbone in France). They rely on a soil centered quantification
approaches which has limitations in terms of accuracy and reliability of the soil and biomass input data and a field scale reso-
lution that often does not match the spatial resolution of in-situ soil and plant growth variability (de Gruijter et al., 2016; Ellili
et al., 2019). Tools including coupled plant/soil process-based models are used to address the spatial representativity challenge
and quality of biomass data monitoring. These models include the main components of the cropland's carbon budget, plants
photosynthesis, and respiration, emission due to soil organic matter mineralisation. These models can also account for carbon
imports through organic fertilisation and carbon exports of biomass at harvest (Smith et al., 2010).

State of the art agronomic models e.g. DSSAT-CSM (Porter et al., 2010), soil models e.g. DNDC (Gilhespy et al., 2014), and
land surface models e.g. ORCHIDEE-STICS, (Gervois et al., 2008) take into account a large array of environmental conditions
to represent the carbon budget components (Net Ecosystem Exchange - $NEE$, Gross Primary Production - $GPP$, autotrophic
respiration - $Ra$, heterotroph respiration - $Rh$), of the crop biomass, and of the yield variables. However, water and nutri-
ent availability, local topography, pests, and historic factors (e.g. former ditches, roads, field limits) highly influence the soil
and plant processes (Gregory et al., 2009). This can results in high spatio-temporal variability in crop development and soil
processes that can be observed even at intra-field scale (Stevens et al., 2008; de Gruijter et al., 2016). Moreover, to operate
those models, farmer activity data and crop development dynamics are required in order to provide accurate estimates of SOC



stock changes. Getting hold on this information at large scale is still very challenging (Seidel et al., 2018; Wattenbach et al., 2010). Yet it is possible to use time series of biophysical variables such as $GLAI$, derived from remote sensing data, to provide information about development dynamics through data assimilation (Battude et al., 2017; Pique et al., 2020a, b). These assim-

ilated observations allow to provide spatially explicit crop specific estimates of biomass and carbon restituted to the soil using coupled soil-plant models. Assimilation exercises of biophysical variables are usually based on iterative optimization methods such as Simplex, Monte-Carlo Markov Chain, Ensemble Kalman filter, or variational assimilation that are generally applied at moderate resolutions (Kumar et al., 2019; Hararuk et al., 2014) or field scales (Trepos et al., 2020; Upreti et al., 2020). It is often computationally prohibitive to apply those methods at intra-field resolution over large areas. This issue of scalability is

key as solving it is a major stepping stone to assess the spatial variability of the $CO_2$ flux components.

The knowledge of this variability is in it's turn a major asset to define soil sampling strategies, assure spatial coherency of model validation shemes, and modulate precision farming practices. In this paper, we address this challenge by presenting the newly developed AgriCarbon-EO processing-chain for the assimilation of EO data into the SAFYE-CO2 agronomic model (Pique et al., 2020a, b) at large scale (100 km) and intra-field resolution (10 m). These spatial resolutions and scales are achieved

by using the new BASALT (Bayesian Normalised Importance sampling via Look-Up Table generation) algorithm that also provides uncertainty estimates. In AgriCarbon-EO, BASALT is used to inverse the PROSAIL (Baret et al., 1992) radiative transfer model to Obtain $GLAI$ at 10 m resolution, $GLAI$ that is thereafter assimilated into the SAFYE-CO2 crop model. In addition, the paper also aims at: evaluating the accuracy of AgriCarbon-EO outputs through a multi-scale validation and coherency exercise; assessing the benefit of high resolution EO data assimilation through a spatial (pixel vs field) and temporal

(single vs multi-mission) analysis; verifying the coherency of the outputs through intra-field as well as regional analysis.

In the following sections, we first present the AgriCarbon-EO processing-chain including the standard inputs, the models and the BASALT assimilation scheme. We then present the numerical experimental setup and the validation data-sets. Next, we present the validation results, and the spatial analysis results. Finally we conclude on the benefits and the limitations of the presented solution for assessing the cropland carbon budget components and their associated uncertainties at high resolution

over large areas.

## 2 AgriCarbon-EO chain

### 2.1 Overview of the processing chain

AgriCarbon-EO is an end-to-end processing chain that simulates multiple relevant variables of crop development, biomass inputs to the soil, and $CO_2$ fluxes at a daily timescale and over large territories, for the assessment of carbon and water budgets.

It is specifically designed to assimilate optical remote sensing datasets at native high resolution into a simple but generic agronomic model (SAFYE-CO2) over large territories. All the processing steps are conceived in a comprehensive manner (Fig.1). A brief point wise description of the data flow and processing steps is presented here and detailed in the following subsections:





1. A pre-processing "Data ingestion" step allows the updating of existing data sets through automated downloading of satellite images and weather forcing. Optical Bottom Of Atmosphere (BOA) reflectances are downloaded for Sentinel-2 and Landsat-8 (referred to as S2 and L8 below). Satellite data are uncompressed and relevant spectral bands are stacked. The weather data is stored in time series with the associated correspondence matrix to the high resolution grid defined by the user. This is done for the zone defined by the input land cover polygon shapefile.

2. The biophysical variable $GLAI$ is retrieved from the satellite reflectance images by inverting a radiative transfer model (PROSAIL). The retrieval of $GLAI$ is based on an adapted Bayesian importance sampling procedure (*i.e.* BASALT). In this step a spatial application of the retrieval model is done for each satellite image independently.

3. The crop model (SAFYE-CO2) parameters are inverted by assimilating $GLAI$ time series using the same Bayesian importance sampling method (BASALT) as in step two. In this case, LUTs are generated based on the closest known weather record. Only the phenological crop model parameters and Ligth Use Efficiency($LUE$) are inverted in this procedure.

4. A post-processing step allows the construction of the output products based on the a posteriori crop model parameter distribution. Geo-referenced maps of the variables of interest in each model (*i.e.* PROSAIL, SAFYE-CO2) are constructed as well as cumulative variables (e.g. NEP that is the cumulative NEE over one corpping year, number of satellite acquisitions, soil water content, etc.).

AgriCarbon-EO is implemented in Python language. A maximum requirement of 5 GB per process, for the satellite images needs to be considered. These requirements allow mono process tests and development on standard computers over smaller study areas, as well as large scale applications (e.g. $100 \times 100$ km) with HPC resources.

## 2.2 Input dataset

In the following subsections, the spatial datasets needed for AgriCarbon-EO are detailed with the corresponding sources for the current study.

### 2.2.1 Landcover map

The main driver for the data preparation is a Land Cover (LC) map in vector shapefile format. This file should contain the boundaries of each agricultural field for a given cropping year over a selected region of interest (*i.e.* border extents of the LC shapefile). Based on the border extents of the LC shapefile the remote sensing and weather forcing data are downloaded and pre-processed. When the simulations are intended to cover several cash crop cycles a multi run scenario of AgriCarbon-EO is considered for each crop cycle. Additionally, a standard simulation can include a cover crop with each cash crop so that the full cropping year can be taken into account. In this paper, AgriCarbon-EO was applied for the winter wheat crops in south-west of France (on the Sentinel-2 tile referenced as 31TCJ) over 2017, 2018 and 2019. The LC shapefile was obtained from the Registre Parcellaire Graphique (RPG) in France ("RPG," 2021), which is available online in open licence v2.0. This





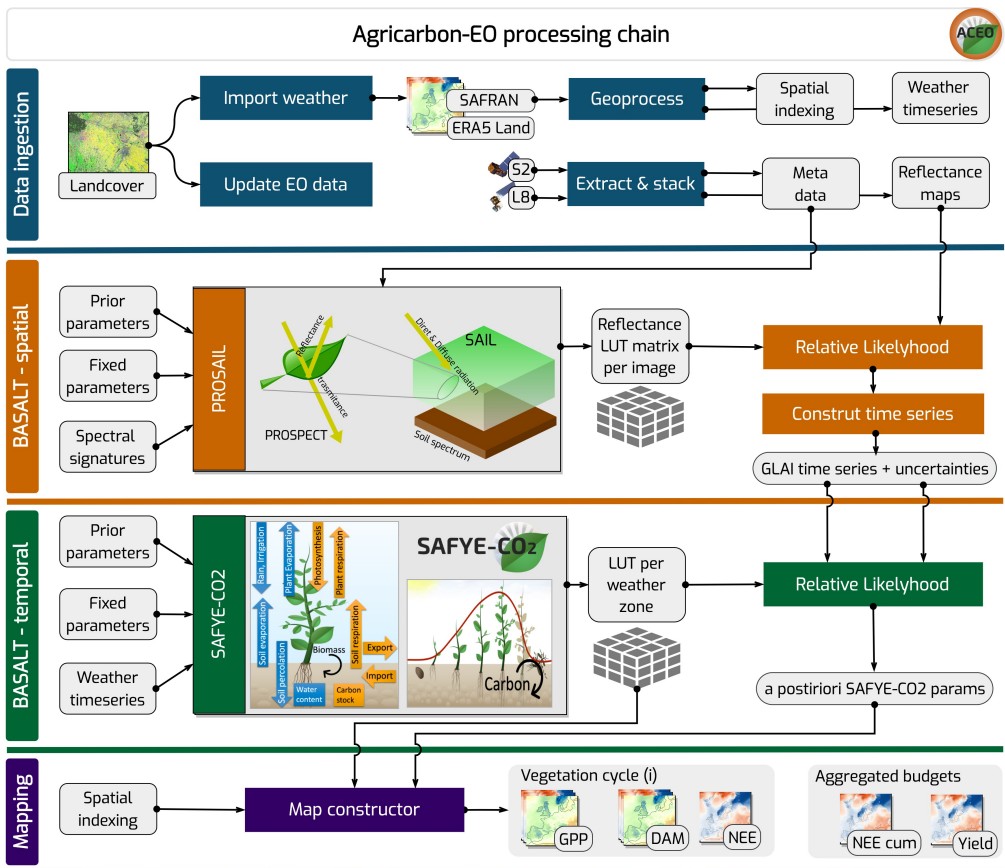

**Figure 1.** Overview of the AgriCarbon-EO data flow and main processing steps that include the data ingestion, BASALT spatial retrieval, BASALT temporal retrieval, and mapping of the variables of interest.

information is produced by the Institut Geographique National (IGN) for the Agence de Service de Paiement (ASP *i.e.* The French Paying Agency) in charge of the implementation, control and payment of the subsidies for the EU Common Agricultural Policy (CAP) in France. The original maps which are in Lambert-93 projection (EPSG:2154 - RGF93) are reprojected to the common grid projection in AgriCarbon-EO, WGS 84/UTM31 in this case.

### 2.2.2 BOA surface reflectances

The assimilated remote sensing data are optical surface reflectances at Bottom Of Atmosphere (BOA), which correspond to reflected energy from the top of the canopy and the soil at a given incidence angle, for a set of observed spectral bands. Currently, AgriCarbon-EO is conceived to use data from EU Copernicus program Sentinel-2 (Drusch et al., 2012) and NASA Landsat-8 (Roy et al., 2014), knowing that the modular interface is compatible with multi-source EO data. The Sentinel-2 data are acquired over 13 optical bands with a resolution of 10 to 60 m depending on the spectral bands with a 5 days revisit from



the constellation. Only the nine visible bands are considered from the Landsat-8 data. Landsat-8 has a revisit of 16 days and a spatial resolution of 30 m in the visible range.

For this study, the data were downloaded from the Thematic centre for continental surfaces (THEIA) that uses a common atmospheric correction and cloud masking algorithm for Sentinel-2 and Landsat-8 through the MAJA processing chain (Hagolle et al., 2021). This enables a harmonised Level-2A database with an efficient cloud masking algorithm (Baetens et al., 2019). The data contains quality indicators including cloud coverage. The datasets are presented as granules (tiles) of $110 \times 110$ km ortho-images in the UTM projection. Prior to the processing, the remote sensing datasets are decompressed, re-sampled at 10 m resolution using nearest neighbour.

### 2.2.3 Weather forcing data

Daily weather data maps covering the simulation period and spatial extents are used to force the crop model. Cumulative daily global incoming solar radiation, $Rg$ in $\mathrm{W\,m^{-2}}$ and daily average air temperature at 2 m, $Ta$ in °C are needed for the vegetation growth module in SAFYE-CO2. Based on previous studies that showed the impact of diffuse radiation on crop development and photosynthesis (Béziat, 2009; Roderick et al., 2001), the diffuse incoming radiation is computed based on De Jong (1980). Furthermore, when the water budget module of SAFYE-CO2 is activated , daily potential evapotranspiration, ET0 in $\mathrm{mm\,d^{-1}}$, and daily cumulative rainfall in $\mathrm{mm\,d^{-1}}$ are extracted from the weather data. AgriCarbon-EO supports two data sources that provide weather data: the Météo-France SAFRAN dataset (Vidal et al., 2010) and the ERA5 Land (Muñoz-Sabater et al., 2021). The extraction of the ERA5 Land data is done via the dedicated API. SAFRAN consists in reanalysis of climate variables at 8 km spatial resolution and hourly timescale over France starting 1958. In this paper, the water module is deactivated and only the weather data ($Rg$ and $Ta$) is extracted from the Météo-France SAFRAN dataset and re-projected over the UTM/31N at 8 km resolution.

## 2.3 Process-based models

### 2.3.1 Radiative transfer modelling using PROSAIL

Maps of geophysical variables (*i.e. GLAI*) are retrieved in AgriCarbon-EO by inverting the PROSAIL radiative transfer model. PROSAIL was extensively used as a radiative transfer model for vegetated areas over a wide range of inversion schemes (Wang et al., 2022). PROSAIL combines the PROSPECT and the SAIL models (Baret et al., 1992). PROSPECT provides leaf spectral properties in the 400 nm to 2500 nm band width (Jacquemoud and Baret, 1990). SAIL (Scattering by Arbitrary Inclined Leaves) is a multidirectional canopy reflectance model (Verhoef, 1984) based on the bidirectional reflectance model (Suits, 1971). PROSAIL and its subsequent versions have been widely used for remote sensing applications (Jacquemoud et al., 2009). The python implementation of PROSAIL is used in AgriCarbon-EO. This version includes the coupled PROSAIL from PROSPECT-5-D (Féret et al., 2017), 4SAIL (Verhoef et al., 2007), and a Simple Lambertian soil reflectance model. PROSAIL parameters are inverted using a Bayesian approach in order to provide $GLAI$ and its corresponding uncertainty as input to the crop model inversion.





### 2.3.2  Crop $CO_2$ fluxes modelling using SAFYE-CO2

SAFYE-CO2 is a parsimonious agronomic model that runs at a daily time-step (Veloso, 2014; Pique et al., 2020a, b). The model stems from the SAFY models (Duchemin et al., 2008; Battude et al., 2017) which computes Dry Above ground bioMass (DAM), based on the Light-Use Efficiency (LUE) theory of Monteith et al. (1977). In Contrast with SAFY, the SAFYE-CO2 model computes the Gross Primary Production (GPP) based on an effective LUE (ELUE) (Pique et al., 2020a).

$$GPP = Rg \cdot \epsilon_c \cdot FAPAR \cdot f_T(Ta) \cdot f_w(WC) \cdot ELUE \cdot SR10 \tag{1}$$

Where $Rg$ is the incoming global radiation ($\mathrm{MJ\,m^{-2}\,d^{-1}}$), $Ec$ is the extinction coefficient, $FAPAR$ is the Fraction of Absorbed Active Radiation, $f_T(Ta)$ is the temperature stress function that depends on $Ta$ the mean air temperature at 2 m (°C), $f_w(WC)$ is the water stress function where WC is the soil water content ($\mathrm{m^{-3}m^{-3}}$). In this study, the water stress function is deactivated (*i.e.* $f_w(WC)$ = 1), $ELUE$ ($\mathrm{gCMJ^{-1}\,m^{-2}}$) is the effective Light Use Efficiency function.

$$ELUE = LUE_a + e^{\left(\frac{Rdiff}{Rg} \cdot LUE_b\right)} \tag{2}$$

Where $LUE_a$ ($\mathrm{gCMJ^{-1}\,m^{-2}}$) is the light use efficiency for direct radiation and $LUE_b$ is a correction coefficient for impact of diffuse radiation on $LUE$.

$SR10$ is a multiplicative factor that takes into account the decrease of photosynthetic efficiency during senescence.

$$SR10 = \begin{cases} \frac{GLA}{GLA\_max \times Cs} & \text{, if } SMT > Sen_a. \\ 1 & \text{, otherwise.} \end{cases} \tag{3}$$

The Net Primary Production $NPP$ ($\mathrm{gCm^{-2}}$) is computed from the $GPP$ ($\mathrm{gCm^{-2}}$) by subtracting the autotrophic respiration $Rauto$ ($\mathrm{gCm^{-2}}$).

$$NPP = GPP - Rauto \tag{4}$$

$Rauto$ is divided into vegetation maintenance respiration $Rmaint$ (Amthor, 2000) and vegetation growth respiration $Rgrow$ (Choudhury, 2000).

$$Rauto = Rmaint + Rgrow \tag{5}$$

$Rmaint$ is computed using a $maint\_coef$ and $SR10$ to represent an increase in relative maintenance cost during senescence.

$$Rmaint = \begin{cases} NPP \cdot maint\_coef \cdot SR10 & \text{if } GPP > NPP \cdot maint\_coef \cdot SR10. \\ GPP & \text{otherwise.} \end{cases} 1 \tag{6}$$

The maintenance coefficient depends on temperature, and two parameters: the basal respiration at 10°C ($R10$) and the soil respiration ($Q10$).

$$maint\_coef = R10 \cdot Q10^{-0.1 \cdot (Temperature - 10)} \tag{7}$$





The growth respiration is computed from the growth conversion efficiency, the $GPP$ and $Rmaint$.

$$Rgrow = (1 - Yg) \cdot (GPP - Rmaint) \tag{8}$$

The $NEE$ $(\mathrm{gC\,m^{-2}})$ is then computed by substracting the heterotrophic respiration $Rh$ from the $NPP$.

$$NEE = NPP - Rh \tag{9}$$

The $Rh$ $(\mathrm{gC\,m^{-2}})$ is computed based on the empirical model in (Delogu et al., 2017) that depends on soil moisture and temperature.

$$Rh = Rh_1 \cdot e^{(Rh_2 \cdot T_s oil)} \cdot H_{water-stress} \tag{10}$$

Where $Rh_1$ is the reference heterotrophic respiration rate, $Rh_2$ expresses the RH sensitivity to temperature and $H_{water\_stress}$ is the effect of soil moisture on soil carbon decomposition.

$$H_{water-stress} = \left(1 + Rh_1 \cdot e^{(Rh_2 \cdot RSM1)}\right)^{-1} \tag{11}$$

Where $Rh\_H1$ and $Rh\_H2$ provide the form of the water stress function and $RSM1$ The relative soil moisture.

The $NPP$ $(\mathrm{gC\,m^{-2}})$ is divided into root and above ground NPP using a root biomass allocation approach (Baret et al., 1992). The root biomass allocation PRT_R is defined using $PRT\_Ra$, $PRT\_Rb$, $PRT\_Rc$, $SMT\_G$ which are respectively the end of cycle biomass allocation to roots, the initial biomass allocation to roots, a coefficient modulating the decrease in Biomass partition to the roots between the initial and end of cycle state, and the sum of temperature at which grain filling starts.

$$PRT\_R = PRT\_Rb + (PRT\_Ra - PRT\_Rb) \cdot e^{\left(-PRT\_Rc \cdot \frac{SMT}{SMT\_G}\right)} \tag{12}$$

The $DAM$ and Dry Below Ground Mass $DBM$ $(\mathrm{g\,m^{-2}})$ are computed from $NPP$

$$\Delta DAM = \frac{NPP}{Cveg} \cdot (PRT\_R) \tag{13}$$

$$\Delta DBM = \frac{NPP}{Cveg} \cdot (1 - PRT\_R) \tag{14}$$

From the change of $DAM$ the green leaf area index change is computed depending on growth or senescence periods, starting from the date of emergence $emerg$.

$$\Delta GLAI^+ = \Delta DAM \cdot PRTL \cdot SLA \tag{15}$$

Where $SLA$ $(\mathrm{m^2\,g^{-1}})$ is the specific leaf area and $PRTL$ is the $DAM$ to leaf biomass partitioning.

$$PRTL = 1 - PRT\_La \cdot e^{PRT\_Lb \cdot SMT} \tag{16}$$



$PRT\_La$ and $PRT\_Lb$ are fitting parameters and $SMT$ is the sum of effective temperature (°C) since emergence.

$$\Delta GLAI^- = GLAI \cdot (SMT - sen_a) \cdot sen_b^{-1} \tag{17}$$

Where $sen_a$ is the sum of temperature (°C) at senescence and $sen_b$ is the rate of senescence. Eq. (15, 17) provide the link between the modeled $GLAI$ and the $GLAI$ retrieved from optical EO and therefore allows to constrain the model's phenological and light use efficiency parameters ($emerg$, $PRT\_La$, $PRT\_Lb$, $SLA$, $sen_a$, $sen_b$, $Harv$, $LUEa$) using EO data
assimilation. Note that, the water fluxes computation in SAFYE-CO2 are based on the Penman-Monteith and FAO-56 methodologies that enables to compute the evapotranspiration and water distribution in the soil based on a bucket model (Allen et al., 1998). The coupling between the carbon and water cycle is two ways. The plant growth impacts the root water uptake and the soil water content impacts the $GPP$ production through a water stress coefficient. Detailed description of the SAFY-CO2 and SAFYE-CO2 model is provided in (Pique et al., 2020a) and (Pique et al., 2020b). In this paper, the water-carbon coupling is
deactivated. The soil stress impact on the vegetation cycle scale is implicitly considered through the assimilation of $GLAI$. The relative parsimony of SAFYE-CO2 compared to models such as STICS (Dumont et al., 2014) or DSSAT (Porter et al., 2010) that have a finer description of the plant, climate and soil processes results in a limited number of model inputs. A new python implementation of SAFYE-CO2 was developed for AgriCarbon-EO. This new version is vectorised in order to provide predictions for multi-runs and build LUTs. It can also handle multiple vegetation cycles for each run (e.g. crop and cover
crop), and has a modular architecture. The physical modules are restructured to regroup soil processes, plant phenology, plant physiology, heterotroph activity and field management.

## 2.4  Bayesian normalised importance SAmpling using Look out Table - BASALT

Answering the need for large-scale high resolution assimilation in AgriCarbon-EO led to the development of a tailored inversion method. The new approach, BASALT, involves the bayesian Normalised Importance Sampling (NIS) approach to answer
the need for uncertainty propagation across the processing chain, and Look-Up Tables (LUT) generation that provides computational gain by reducing the total number of model simulations. In a Bayesian framework, the initial knowledge about the model's parameters is represented by a probability distribution $P(\Theta)$, the so-called prior distribution. The knowledge brought by the observations $x$ is expressed by the conditional probability distribution $P(\Theta|x)$ of the model parameters knowing that the observations $x$; the so-called a posteriori distribution. The goal is to evaluate this a posteriori distribution which can be
evaluated using Bayes theorem that connects $P(\Theta|x)$, $P(\Theta)$, and $P(x|\Theta)$ called the likelihood that is simply the conditional probability of getting an observation given particular value of parameters (Eq.18).

$$P(\Theta, x) = \frac{P(x|\Theta)P(\Theta)}{P(x)} \tag{18}$$

$P(x)$ is the probability of observation (the so-called marginal distribution) that can be expressed as $\int P(x|\Theta)P(\Theta)d$. The Bayesian framework offers the advantage of providing the uncertainty estimates via the a posteriori distribution. It is at the
root of popular inversion algorithms such as MCMC or Dream (Vrugt, 2016). Such algorithms have been applied to agronomic models (Dumont et al., 2014), to ecosystem (Ma et al., 2022) and to radiative transfer models (Zhang et al., 2005).



### 2.4.1 Normalised Importance Sampling and Look-up table

In our case MCMC based approaches are computationally inadequate because they would impose dependent iterative proce-
dures over a very large number of inversion problems. This is a common feature of iterative methods when applied to a large
number of inversions. For example, considering an iterative approach, the total computation time f(e) can be approximated.

$$f(e) = e \cdot N \cdot (t_{sample} + t_{run} + t_{eval}) \tag{19}$$

Where $e$ the number of entities, $N$ is the number of iterations, $t_{sample}$ is the time to make the sampling, $t_{run}$ is the process-
based model run time, and $t_{eval}$ is the time for each objective function evaluation. Given that the process-based models in
AgriCarbon-EO are parsimonious, which entails relatively low complexity inversion problems, a non-iterative Bayesian NIS
method is applied. NIS approximates the a posteriori distribution with a vector of weighted parameters or variables. This
method has been applied to crop model inversions (Hue et al., 2008) but not to the assimilation of satellite imaging. Actually,
when the processed entities are numerous for a similar set of forcing inputs (*i.e.* weather grid), the iterative inversion produces
a considerable amount of common solutions in the explored parameter space (*i.e.* samples). Those common solutions hint at
the possibility to use a LUT for each group of entities that share a common forcing and evaluate the relative likelihood of each
LUT entry. Thus, if we consider that the number of samples in the LUT is $n$, the computation time g(e) of the new solution
changes.

$$g(e) = n \cdot (t_{sample} + t_{run} + e \cdot t_{eval}) \tag{20}$$

When the number of entities to inverse is large ($e.N >> n$) and ($t_{eval} << t_{run}$), this approach requires less simulations and
lower time per entity. In the case of AgriCarbon-EO, the runs are done over tens of millions of entities, the evaluation time
based on log-likelyhood is extremely low, and since the scheme entails statistically independent samples runs are vectorized
which reduces $t_{run}$. The solution presents one drawback in terms of higher memory overhead to store the LUTs compared to
an iterative approach.

### 2.4.2 Log-likelyhoods computations

In practice when evaluating the different LUT entries (i.e weather grid points, images) over a given pixel, we compute log-
likelyhoods to facilitate the computations.

$$logL_{i,j} = \sum (-\frac{1}{2}log(2\pi(\sigma_{o,i,j})^2)) - \frac{(v_{o,i,j} - \mu_{o,i,j}^2)}{2\sigma_{o,i,j}} \tag{21}$$

Where $v$ is the simulation value, $\mu$ and $\sigma$ are the mean and standard deviation of the observation, $j$ is the index for entities,
$o$ is the index of the independent observations, and $i$ is the index for the model run in the LUT. Naive implementation of
this expression leads to the manipulations of depth-3 arrays (number of entities $\times$ number of observations $\times$ number of LUT
samples) that can be rather inefficient in terms of RAM and CPU usage. Eq.(21) is reformulated in a tensor form Eq.(22) that





just involves inner products of depth-2 arrays *i.e.* simple and efficient vanilla matrix product.

$$logL_{i,j} = -\frac{1}{2}(\sum_o (log(2\pi\sigma_{j,o}) + \frac{\mu_{j,o}^2}{\sigma_{j,o}^2} \otimes 1_i) + \sum_o (v_{i,o}^2 \cdot \frac{1}{\sigma_{j,o}^2}) + \sum_o (v_{i,o} \cdot (\frac{\mu_{j,o}}{\sigma_{j,o}^2}))) \tag{22}$$

To represent the likelyhoods using float numbers the log-likelyhoods are re-scaled by their maximum. Eq.(23).

$$SlogL_{i,j} = logL_{i,j} - max_i(logL_{i,j}) \tag{23}$$

Following this step the relative likelyhoods are computed.

$$RL_i = \frac{e^{logL_i}}{\sum_i e^{logL_i}} \tag{24}$$

A final step is used to summarise the results with a normal distribution by computing a weighted mean (Eq. 25) and standard deviation (Eq. 26).

$$\mu_w(v_i, RL_i) = \frac{\sum_i v_i RL_i}{\sum_i RL_i} \tag{25}$$

Where $\mu_w$ is the weighted mean , $v_x$ the vector given by the LUT for a parameter or variable, and $x$ the number of samples.

$$\sigma_w(v_i, RL_i, \mu_w) = \frac{\sum_i (v_i - \mu_w)^2 \cdot RL_i}{\sum_i RL_i} \tag{26}$$

Where $\sigma_w$ is the weighted standard deviation.

In summary the processing steps for BASALT are the following:

1. Random samples are generated for each node of the weather forcing grid based on a probability distribution that rep-
resents the best of our prior knowledge of the model's parameters. The model output variables are calculated for each sampled parameters combination.

2. The log-likelyhood of each model realisation is computed given available observations with uncertainty estimates at the observation grid resolution (*i.e.* 10 m).

3. The relative likelihood (RL) which allows the estimation of the probability distribution of the model's parameters is
computed.

### 2.4.3  Retrieval of $GLAI$ maps from PROSAIL

When inverting PROSAIL, the main objective is to retrieve the $GLAI$ and associated uncertainties that will be assimilated by SAFYE-CO2. This is done by generating a LUT of PROSAIL runs (size = 5000) for each remote sensing image based on a given prior. Equations (22,23 and 24) are then used to evaluate the RL where $j$ is the index of pixels in the simulated image, $i$ the
index of the PROSAIL runs in the LUT and $o$ of the observed reflectances from the Sentinel-2 or Landsat-8 images. The prior used for the LUT generation is shown in Table 1. The solar zenith, observer zenith and relative azimuth angles are provided





by Sentinel-2 and Landsat-8 products used for LUT generation. As PROSAIL provides LAI and not $GLAI$, the chlorophyll content ($cab$) is constrained to a high interval [60,80] $\mathrm{ug\,m}^{-2}$. This forces all considered foliar surfaces to be green and thus allows to inverse $GLAI$. A constraint is also added on the relation between dry Biomass and $GLAI$ to reduce the parameter search space by eliminating solutions with leaves that are too thin or thick. Then, the surface reflectances of the Level2-A BOA products are considered to follow a normal distribution with mean and standard deviation that is considered constant at 0.02. Finally the a posteriori distribution is approximated with a normal distribution, using Eq.(25) and (26) to determine $\mu$ and $\sigma$.

**Table 1.** Priors configuration for PROSAIL parameters used in the Bayesian inversion.

| Name | description | Unit | Prior (uniform [min,max]) |
|---|---|---|---|
| N | Leaf structure parameter | . | [1,2] |
| $cab$ | Chlorophyll a+b concentration | $\mu$g m$^{-2}$ | [60,80] |
| $car$ | Carotenoid concentration | $\mu$g m$^{-2}$ | [5,20] |
| $cm$ | Leaf thickness | g cm$^{-2}$ | [-0.02,0.02] + GLA·0.004 |
| $LAI$ | Leaf area index | m$^2$ m$^{-2}$ | [0,5] |
| $psoil$ | Soil moisture Index | . | [0,1] |

### 2.4.4 Application of BASALT to SAFYE-CO2

The simulated variables, $DAM$, $yield$, $GPP$, ecosysthem respiration $Reco$ and $NEE$ are highly dependent on the duration and intensity of the crop development (Ceschia et al., 2010). The $GLAI$ outputs from PROSAIL are assimilated into SAFYE-CO2 to correct the naive prior vegetation dynamics. This is done by generating a LUT of SAFYE-CO2 runs (size = 10000) for each zone with the same forcing (*i.e.* same prior). In this case, the zoning is defined by the weather forcing data (*i.e.* SAFRAN at 8 km). For each zone, equations Eq.(22),(23) and (24) are applied to evaluate the RL given the $GLAI$ observations, where $j$ is the index of pixels in the simulated area, $i$ the index of for the SAFYE-CO2 runs in the LUT, and $o$ the observed $GLAI$ at different dates. The priors for LUT generation for the SAFYE-CO2 are shown in Table 2. Those priors are used for the SAFYE-CO2 LUT generation and were reassessed in terms of statistical distribution from (Pique et al., 2020a) to account for the high-spatial heterogeneity that can be observed at regional scale. For each parameter a truncated normal distribution is sampled with parameters $\mu$, $\sigma$, $min$ and $max$ independently; the only exception is $PRT\_Lb$ which has an exponential behaviour. For this parameter a logarithmic transformation is applied on the distribution.

$$RL_i = \frac{\sum_j RL_{i,j}}{\sum_i \sum_j RL_{i,j}} https://fr.overleaf.com/project/5fca6d08a7eb14021f9d4b3b \qquad (27)$$

To aggregate the SAFYE-CO2 simulations at field scale, the likelyhoods are summed over all the pixels in the field 27. Eq. (25) and Eq. (26) are used to compute the mean and sigma for a parameter or a variable at a given day for a field or pixel.



**Table 2.** Priors configuration for SAFYE-CO2 parameters used in the Bayesian inversion.

| Name | Description | Unit | Prior $[\mu,\sigma,\text{min,max}]$ |
|------|-------------|------|------------------------------------|
| $emerg$ | Day of year of vegetation emergence | DOY | $[335, 15, 200, 400]$ |
| $harv$ | Day of year of vegetation harvest | DOY | $[200, 0, 160, 200]$ |
| $LUE_a$ | Light use efficiency | g MJ$^{-1}$ | $[1.05, 0.05, 0.8, 1.5]$ |
| $SLA$ | Specific Leaf Area | m$^2$ g$^{-1}$ | $[0.01, 0.002, 0.004, 0.05]$ |
| $PRT\_La$ | Initial fraction of biomass that is not allocated to the Leafs | g g$^{-1}$ | $[0.325, 0.15, 0.01, 0.5]$ |
| $PRT\_Lb$ | decrease rate of the fraction of biomass allocated to the leaves. | g g$^{-1}$ °C$^{-1}$ | $[1.01, 0.005, 1, 1.02]$ |
| $sena$ | Sum of temperature at which senescence starts | °C | $[1350, 200, 1000, 2000]$ |
| $senb$ | Rate of senescence | °C m$^2$ m$^{-2}$ | $[12000, 3000, 0, 20000]$ |

# 3 Application in South-West France over Wheat

In the aim of validating and demonstrating the capabilities of AgriCarbon-EO, a 110×110 km region defined by the 31TCJ
Sentinel-2 tile located in South-West of France has been chosen. In this zone the chain is applied over a for winter wheat in
years 2017, 2018, and 2019 (Figure 2). Several assimilation experiments were conducted to answer the specific objectives of
the paper, they are summarised in Table 3. They alternate the use of S2 alone and the combined use of S2 and L8. Pixel and field
scales are also considered. The ACEO-S2L8-Pixel combines Landsat-8 and Sentinel-2 data at 10 m resolution which represents
about 20 M pixels for our study area. It is used as the main simulation for the validation experiments. The ACEO-S2L8-Field
simulations correspond to averaging the 10 m $GLAI$ from PROSAIL retrieval at field scale. Additionally, an averaging of the
high resolution simulations with Sentinel-2 and Landsat-8 is performed at field scale (ACEO-S2L8-Mean).

**Table 3.** ID, inputs details, and objectives motivating the simulation experiments.

| Name | RS Data | Spatial resolution | years | Objectives |
|------|---------|--------------------|-------|------------|
| ACEO-S2-Pixel | S2 | Pixels (10m) | 2017 | - Determine the impact of revisit |
| ACEO-S2L8-Pixel | S2 & L8 | Pixels (10m) | 2017,18,19 | - Validate the model outputs |
| | | | | - Regional scale spatial analysis |
| ACEO-S2L8-Field | S2 & L8 | Fields* | 2017 | - Quantify the impact of resolution. |
| ACEO-S2L8-Mean | S2 & L8 | Fields** | 2017 | - Quantify spatial and variability . |

\* for ACEO-S2L8-Field the $GLAI$ from PROSAIL inversion is averaged prior to the SAFYE-CO2

\*\* for ACEO-S2L8-Mean the outputs at 10m from SAFYE-CO2 are averaged at field scale.





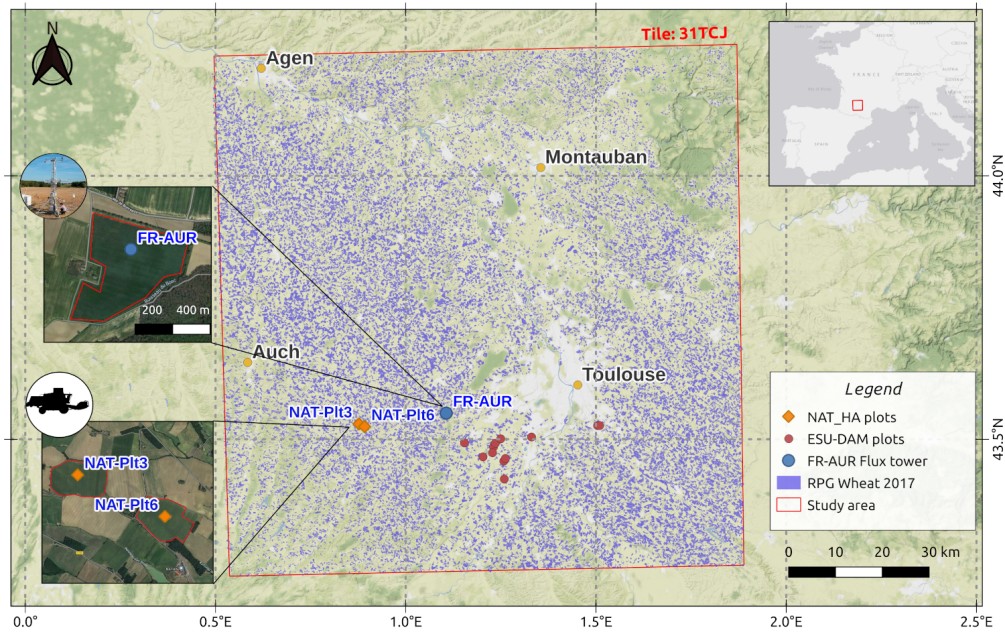

**Figure 2.** Map of the simulation area: in background the terrain map, tile 31TCJ limts (red rectangle), land cover for winter wheat fields for 2017 (blue), location of the FR-AUR flux towers (blue circle), the Dry Above Biomass ($DAM$) measurements for ESU-DAM (red circles), two fields monitored with connected combine harvester (CH) (orange circles). Zoom maps show the FR-AUR field and the combine harvester fields.

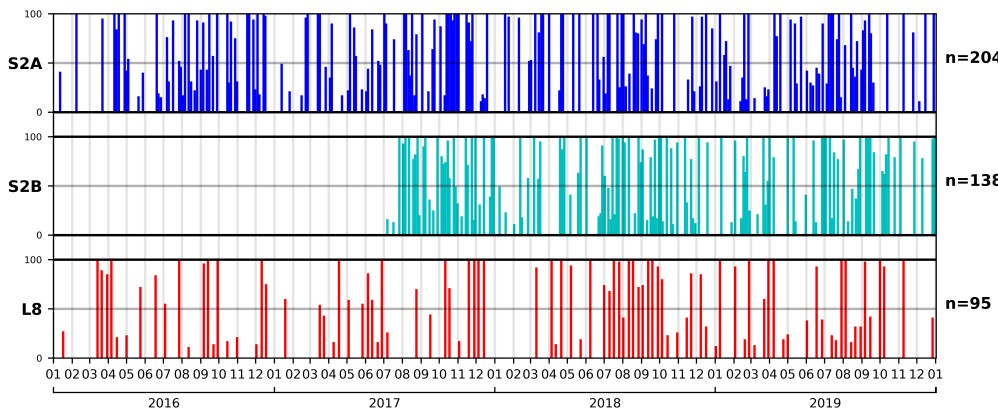

**Figure 3.** Chronogram of the used remote sensing dataset from Sentinel-2A (S2A), Sentinel-2B (S2B) and Landsat-8 (L8), over the 31TCJ tile for years 2016 to 2019. The bars plots represent the percentage of cloud-free pixels for each image.





## 3.1 Study area

The region of interest is covered by tile 31TCJ (Figure 2). It is also part of the Space Regional Observatory that benefits from extensive validation data that are used in this study and presented in Section 3.2. The region has a mean annual precipitation of 655 mm and a mean annual temperature close to 13°C. It is classified as majorly temperate oceanic climate (Cbf) in the plain, and temperate continental climate (Dfb) near the Pyrénnées mountains, based on the Koppen climate classification. In year 2017, winter was exceptionally dry and sunny, and spring was sunny with 10 % deficit in rainfall (Météo-France, 2019), while year 2019, had a mild winter and a sunny spring with 10 % deficit rainfall for the two seasons (Météo-France, 2021). The region has an intermediary cloud coverage that allows for multi-temporal optical remote sensing analysis and analysis of the impact of clouds (Figure 3). It is mainly occupied by agricultural fields that cover about 90 % of the area, among which a majority of seasonal crops. Winter wheat covers around 20 % of the zone and reaches 40 % in some areas. In South-West France, soft-wheat varieties are predominant and they are usually sown in autumn around mid to end October. They represent 75 % of the French exports of soft-wheat. The crop typically develops slowly during the winter and growth accelerates during spring. It is harvested depending on maturation as well as climatic conditions to optimise grain quality from mid June to end of July. The harvest in 2017 was in the norm (6 $\mathrm{t\,ha}^{-1}$ at 15 % humidity), while 2019 was an exceptionally good year with a yield of 11.5 $\mathrm{t\,ha}^{-1}$ at 15 % humidity (ARVALIS, 2019). In terms of pedology, two main soil classes cover the area of study: silt rich soils near the major streams, and clay soils across the hills with a variable density of stones depending on erosion. The topography offers a wide range of expositions. The region also bears the effects of the historical land management, "Remembrement" policy, a political push to merge adjacent fields from 1945 to 1980 in France (Baker, 1961). This leads to a wide range of soil and micro-climatic conditions that cause significant intra-field plant growth variability.

## 3.2 Validation datasets

The validation is based on several datasets covering the main output variables of AgriCarbon-EO: $CO_2$ flux measurements (*i.e.* Net Ecosystem Exchange, $NEE$; Gross Primary Production, $GPP$; Ecosystem Respiration, $Reco$); Dry Aboveground biomass, $DAM$ measurements, and yield maps. A summary of the ID and characteristics of the aforementioned validation datasets is presented in Table 4. The validation datasets are extracted from the database of the Environmental Information

**Table 4.** Description of the validation datasets.

| ID | Source | Type | Sampling | Scale | Frequency |
|---|---|---|---|---|---|
| FR-AUR C-Flux | ICOS | GPP, Reco, NEE | Eddy covariance | Intra-field | Daily |
| FR-AUR DAM | ICOS | DAM | Points in the FR-AUR field | 10 m | End of cycle |
| ESU-DAM | RSO | DAM | 8 Elementary Sampling Units | 10 m | 1-4 Dates |
| NAT-HA | farmer | Yield | 2 CH at two fields fields | 30 cm | End of cycle |

System the laboratory and the Regional Spatial Observatory (RSO).his Information systhem centralises the in-situ datasets



of the RSO including the data from the ICOS flux towers located in the RSO (e.g. FR-AUR) (SIE, 2022). In addition to the validation exercise, the large scale spatial maps from AgriCarbon-EO are analysed with respect to soil texture provided at $250m$ resolution from SOILGRIDS and Digital Elevation Model (DEM) provided by the European Environmental Protection Agency (EPA). The validation strategy covers a wide range of spatial and temporal scales. Figure 4 shows the spatial and temporal representativity of the inputs, the validation datasets, the regional datasets, and AgriCarbon-EO outputs.

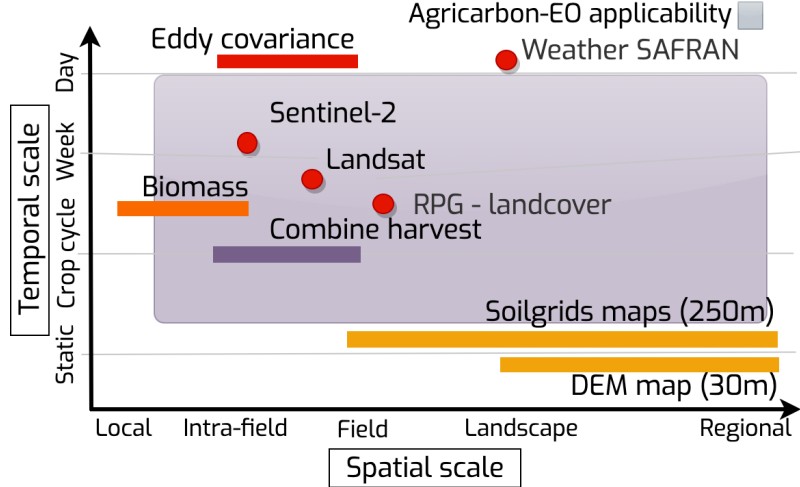

**Figure 4.** Temporal and spatial coverage of the input dataset (S2,L8, SAFRAN, and RPG), the validation dataset (Eddy covariance, Biomass, and Combine harvest), the regional dataset (SOILGRIDS and DEM), and AgriCarbon-EO outputs (blue zone).

### 3.2.1 CO$_2$ fluxes from eddy covariance

The CO$_2$ fluxes components $GPP$, $Reco$ and $NEE$ are provided from the ICOS site at Auradé France (FR-AUR) using the EdiRe software (Clement, 2008) and following the CarboEurope-IP recommendations for data filtering, quality control, and gap filling (Table 4). The computation is based on the Eddy-Covariance (EC) method for CO$_2$ which consists in measuring at 20 hz the 3D wind fluctuations using a high frequency sonic anemometer and the CO$_2$ concentration using an open path analyser. The covariance is then computed between the turbulent component of the vertical wind and the turbulent component of the CO$_2$ concentrations (Baldocchi, 2003). The CO$_2$ flux data was considered representative of the FR-AUR. The NEE corresponds to the sum of the CO$_2$ fluxes and the changes in CO$_2$ storage around the EC devices. The NEE was then partitioned into gross primary production (GPP) and ecosystem respiration (Reco) using a formulation for croplands (Béziat, 2009) adapted from (Reichstein et al., 2005).

### 3.2.2 Vegetation biomass from destructive samples

The dry above ground biomass is obtained from ICOS (FR-AUR-DAM) and from Elementary Sampling Units (ESU-DAM) (Table 4). In the ESU protocol, the above ground vegetation is sampled with five points following a cross pattern inscribed in





a $10 \times 10$ m square; each sample corresponds to a one linear meter of the crop row. The five samples are weighed fresh in the field. In the lab one of the five samples was dried to retrieve the relative humidity, which is then applied to the five fresh weight measurements to obtain dry above ground biomass. The mean and standard deviation are computed to obtain a representative $DAM$ $(\mathrm{g\,m^{-2}})$ for the ESU. Eight fields were also sampled using the ESU protocol in 2018.

### 3.2.3 Yield maps from combine harvester

Yield maps are provided by a farmer for the Gers department. Data from two fields NAT-Plt3 and NAT-Plt6 (Table 4) were produced by a combined harvester (CH) that measures the incoming flow of grain, its humidity as well as its position at a fixed frequency with a GPS. These measurements are integrated between two points of the trajectory taking into account harvesting width to compute the grain production (yield) per surface area. Grain humidity content enables the computation of the dry yield mass $(\mathrm{g\,m^{-2}})$. The point yield data is then converted into a harvest map over the simulation grids by summing the points inside

each pixel. A Gaussian smoothing filter with sigma = 12 m is then applied over these maps to reduce the aliasing effects. The spatial anomaly (*i.e.* $(value - \mu)/\sigma$) maps are also computed.

## 4 Validation against flux towers and fields data

### 4.1 Validation of carbon fluxes with flux towers data

$GPP$, $Reco$, and $NEE$, the three main components of the $CO_2$ fluxes, are compared to the observations from the FR-AUR

flux site. In this exercise, the daily outputs from AgriCarbon-EO at 10 m resolution are spatially averaged over the area of the FR-AUR field (Eq. 27) that is sampled by the EC tower (a.k.a. the target area in the ICOS nomenclature). Figure 5 (a) shows the fitting of predicted $GLAI$ and the assimilated observed $GLAI$. The $GLAI$ fitting statistics computed over the growing season show a good fit ($R^2$ = 0.93) in with a lower fit for the growing season in 2019 ($R^2$ = 0.88). From mid November 2018 until end January 2019, a spontaneous regrowth is observed alongside the wheat and not fitted which is correct as it doesn't correspond to

the wheat vegetation cycle. The $GLAI$ for year 2019 senescence period is under-fitted while the $NEE$ estimates are good. For 2017, the dynamics of $GPP$, $NEE$ and $Reco$ are well represented with most of the observed values in the uncertainty margin of the model with a $R^2$ of 0.88, 0.91, and 0.62 for $NEE$, $GPP$, and $Reco$ respectively. The model's daily flux variations are slightly higher than the observations in 2017. For 2019, however, the modelled $GPP$ values are significantly lower than the observed ones during the growing period (bias = 1.23 $\mathrm{g\,C\,m^{-2}}$) while the differences between the model and observations

are less pronounced at the end of the vegetation cycle. It is also noticeable that the observed $Reco$ dips to zero during the vegetation growth period, which can be related to an anomaly in the partitioning process of NEE into GPP and Reco. The performances of the modelled $NEE$ in 2017 and 2019 are in the range of the scores observed in previous validation exercises with SAFYE-CO2 (RMSE = 1.43-1.90 $\mathrm{g\,C\,m^{-2}}$,Pique et al. (2020b)) despite the poor representation of the beginning of the vegetation cycle in 2019 (RMSE = 3.33 $\mathrm{g\,C\,m^{-2}}$).





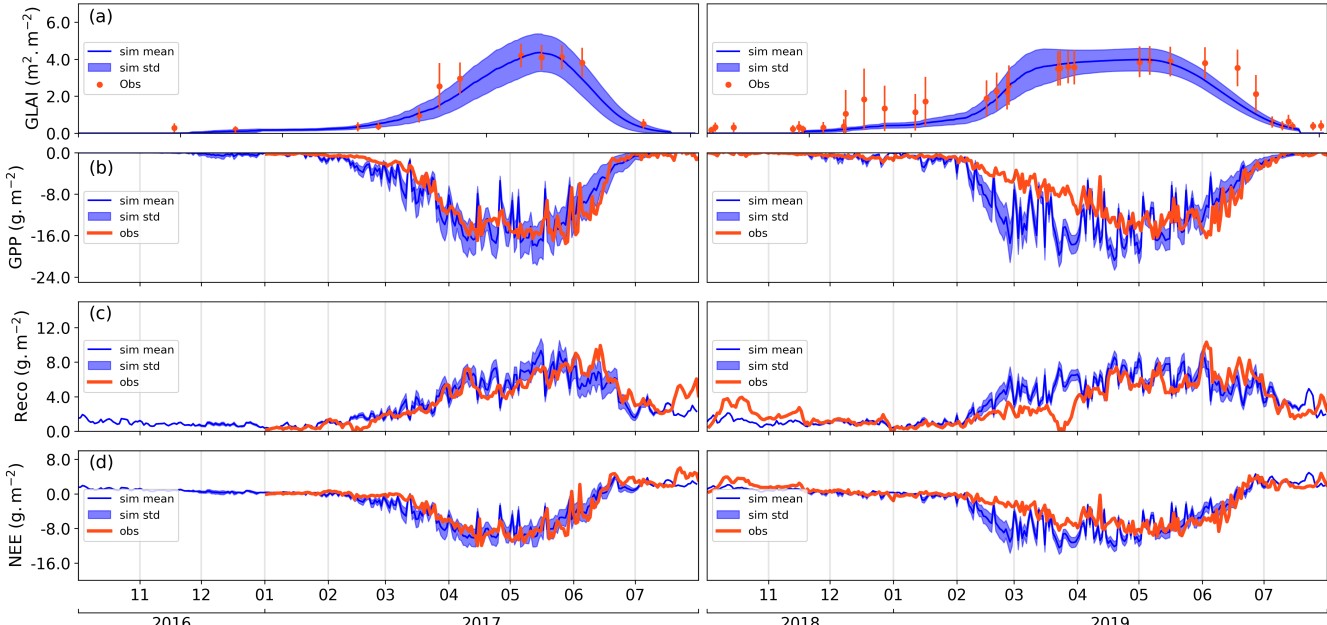

**Figure 5.** Time series of $GLAI$ and $CO_2$ fluxes. In blue the a posteriori distribution and the standard deviation. In red the $GLAI$ derived from the satellite observations and the $NEE$, $Reco$ and $GPP$ at the FR-AUR site for two croping years (2017 and 2019).

## 4.2 Validation of biomass with in situ observations

Figure 6 illustrates the temporal variations of biomass over the FR-AUR field in 2017 and 2019 cropping years, together with the field destructive biomass measurements. The biomass at the end of the cycle is higher for the 2019 cropping year than in 2017 which is consistent with the regional yield statistics (ARVALIS, 2019). Also, the modelled above-ground biomass dynamics are consistent with the observed ones apart from an overestimation of the simulation in the beginning of the vegetation cycle in 2017. This overestimation stays in the combined margin of uncertainty of the model and spread of the observations.

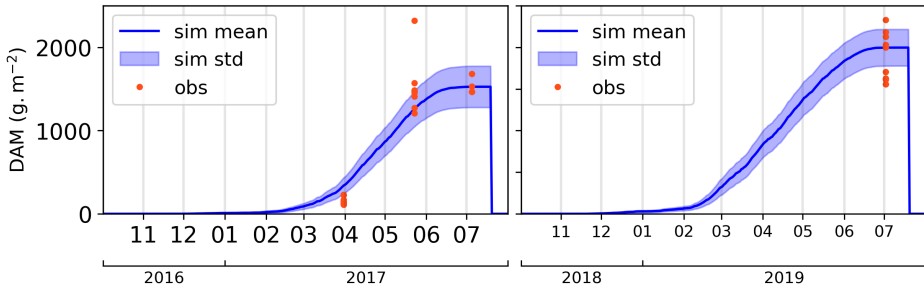

**Figure 6.** Dry above ground biomass ($DAM$) time series with corresponding observations of destructive $DAM$ for winter wheat crop at the FR-AUR site for the 2017 and 2019 cropping years.





**Table 5.** Bias, $R^2$ and RMSE statistics for $GLAI$, $GPP$, $Reco$ and $NEE$ variables in FLX-AUR site over years 2017 and 2019 over key phenological stages.

| Variable | Statistic | 2017 | | | 2019 | | |
|---|---|---|---|---|---|---|---|
| | | Growth | Senescence | All | Growth | Senescence | All |
| $GLAI$ | Bias (m$^2$ m$^{-2}$) | 0.18 | 0.36 | 0.26 | 0.44 | 0.21 | 0.35 |
| | $R^2$ | 0.94 | 0.92 | 0.93 | 0.84 | 0.96 | 0.88 |
| | RMSE (m$^2$ m$^{-2}$) | 0.38 | 0.63 | 0.48 | 0.75 | 0.50 | 0.64 |
| $GPP$ | Bias (g m$^{-2}$ ) | 0.93 | 0.61 | 0.36 | 3.62 | -0.64 | 1.23 |
| | $R^2$ | 0.92 | 0.62 | 0.91 | 0.75 | 0.86 | 0.76 |
| | RMSE (g m$^{-2}$ ) | 1.87 | 1.97 | 1.91 | 5.05 | 2.35 | 3.43 |
| $Reco$ | Bias (g m$^{-2}$) | -0.40 | 0.24 | 0.03 | -1.32 | 0.02 | -0.33 |
| | $R^2$ | 0.78 | 0.87 | 0.62 | 0.60 | 0.49 | 0.60 |
| | RMSE (g m$^{-2}$) | 0.84 | 1.54 | 1.91 | 2.00 | 1.51 | 1.59 |
| $NEE$ | Bias (g m$^{-2}$) | 0.53 | 0.24 | 0.38 | 2.31 | -0.62 | 0.89 |
| | $R^2$ | 0.88 | 0.88 | 0.88 | 0.71 | 0.86 | 0.88 |
| | RMSE (g m$^{-2}$) | 1.43 | 1.97 | 1.69 | 3.33 | 1.90 | 2.40 |

Figure 7 shows the scatter plot between the simulated and observed $DAM$ at the FR-AUR site (2017 and 2019 cropping years) and ESU-DAM. The comparison shows a good fit when considering together all $DAM$ measurements with a $R^2$ of 0.90, a RMSE of 250 g m$^2$ and a slight negative bias 52 g m$^{-2}$. The differences between simulations and observations are most noticeable for the high biomass values of the 2019 cropping years, with observed biomass varying by about 800 g m$^{-2}$, while 415 the simulations are constant at 2000 g m$^{-2}$. Note also that the uncertainties on the 2017 and 2019 in-situ measurements were high ($\sigma = 0.17 \times \mu + 13.09$) while the uncertainty on the ESU-2018 measurements were much lower and the model shows a better fit that year with an $R^2$ = 0.94 (Table 6).

**Table 6.** Statistics in terms of RMSE, MAE, Bias and $R^2$ between the simulated and observed Dry Above ground bioMass ($DAM$).

| Dataset | Bias (g m$^{-2}$) | $R^2$ | RMSE (g m$^{-2}$) |
|---|---|---|---|
| ESU-2018 | -129 | 0.94 | 211 |
| FR-AUR-2017 | -6 | 0.97 | 172 |
| FR-AUR-2019 | 4 | - | 380 |
| All | -52 | 0.90 | 250 |



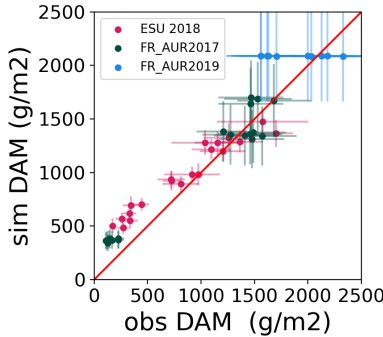

**Figure 7.** Scatter plot of the simulated winter wheat dry above ground biomass ($DAM$) versus the observed ones at FR-AUR in 2017 (green circles) and 2019 (blue circles), and at the ESU fields sampled in 2018 (red circles).

### 4.2.1 Validation of yield maps with connected combine harvester data

The yields simulated with AgriCarbon-EO are compared to the CH yield maps of 2019 over 2 fields. Before presenting the
results it is important to point out that accurate mapping of yield by CH is quite challenging as 1) The mass flow sensor and of the grain moisture content sensor can experience significant sensor drift within the field, 2) CH yield data processing requires a range of parameters such as lag time settings and distance travelled via GPS measurements, header position and cut width, all of which contribute to the uncertainty in the measurements (D. Grisso et al., 2002). The simulated yield maps are obtained from AgriCarbon-EO simulation ID: ACEO-S2L8-Pixel by multiplying the final $DAM$ by the Harvest Index (HI).
We analyse the results in terms of reproduction of the spatial patterns in Figure 8. These maps shows the comparison between the CH yield data and the AgriCarbon-EO yield estimates at pixel level in $\mathrm{t\,ha^{-1}}$ as well as the spatial yield anomaly. The simulated and observed yields for the 2019 cropping year show high values in coherence with the regional yield statistics of that year ($11.5\,\mathrm{t\,ha^{-1}}$) ARVALIS (2019). Overall the observed yields show a larger variability than the simulations and a clear saturation effect is observed in the simulations for the NAT-plt6 field. The AgriCarbon-EO and the CH anomaly maps show
high heterogeneity ($-2\sigma$ to $+2\sigma$ anomaly) with clear spatial patterns. Yet the spatial patterns are more pronounced over the NAT-Plt3 field than over NAT-Plt6. An RMSE, bias, and $\mathrm{R}^2$ of $0.66\,\mathrm{t\,ha^{-1}}$, $-0.38\,\mathrm{t\,ha^{-1}}$, 0.12 and $0.71\,\mathrm{t\,ha^{-1}}$, $-0.40\,\mathrm{t\,ha^{-1}}$, 0.25 are observed for NAT-Plt3 and NAT-Plt6, respectively. The performances of the yield simulations vary strongly between the two fields. A relatively low RMSE and bias indicate a quite good mean representation of the plots. However the correlation coefficient is quite low and indicates that not all the spatial variability in yield can be captured using this approach. The Low
correlation as well as the difficulties in reproducing the range of yield observed variations of yield values may be caused by the simple representation of grain biomass allocation through the use of a HI which does not take into account potential variations of harvest index due to nutrient availability or crop cycle duration (Dai et al., 2016).



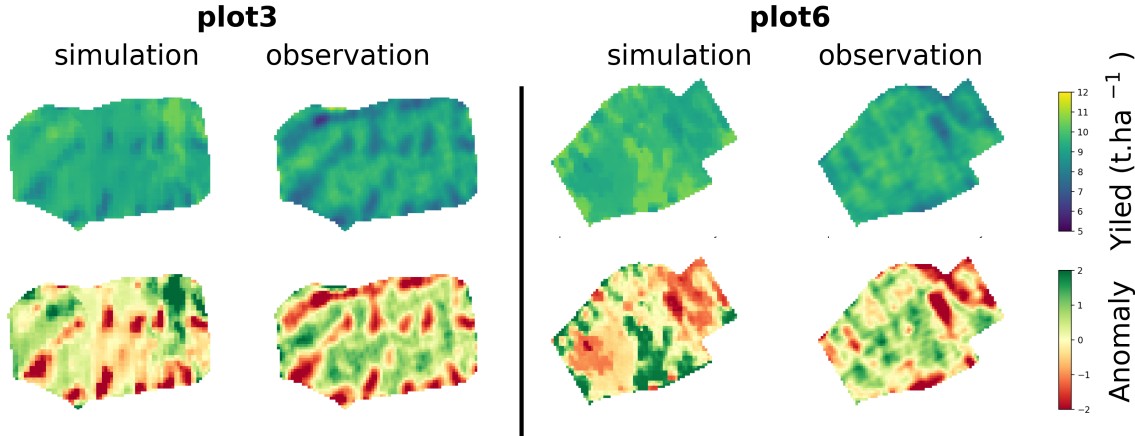

**Figure 8.** Values and spatial anomaly of yield maps simulated by AgriCarbon-EO and observed yield from combine harvester over the Nataïs sites (NAT-Plt3 and NAT-Plt6) for the 2017 and 2019.

# 5 Large scale simulations

## 5.1 Net Ecosystem Exchange, parameter distributions, and singularities

In this section, the results from the ACEO-S2L8-Pixel in 2017 are illustrated and analysed. The RPG shapefile (2017 winter wheat fields), the SAFRAN weather data, the THEIA S2 and L8 EO data were used as input along with the parametrization files for PROSAIL and SAFYE-CO2. The AgriCarbon-EO processing chain was run in parallel computation over a single server rack with 2 computation nodes and with 36 thread max. The memory requirement was the highest for the PROSAIL retrievals reaching 20 Gb per process (image inversion) considering 5000 LUT size. For SAFYE-CO2 the requirements were 5 Gb per process with one process per node of the weather grid considering a 5000 LUT size. A full chain run over the 110 $\times$ 110 km area of study at 10 m resolution requires 4 hours of computation time per year of simulation. The chain produces a considerable amount of variables linked to the carbon (and water) cycles at daily scale. Here, we considered the inverted model parameters ($emerg$, $sena$, $senb$, $PRT\_La$, $PRT\_Lb$, $LUEa$, $SLA$), the $DAM$ at end of the growing season, the NEP (Net Ecosystem Production) which corresponds to the aggregated NEE over the crop cycle time span (from 2016/11/01 to 2017/08/01). The maps in Figure 9 show the NEP (a) with its corresponding uncertainty (b) at native resolution (10 m) as an illustration of typical outputs from the chain. A NEP of +390 to -820 $\mathrm{gCm^{-2}}$ is obtained at 98 % probability interval. Positive values correspond to pixels where no vegetation developed during the cropping year (e.g. no or bad emergence), meaning that heterotophic resiration dominated, while the higest values are in the range of the highest observed NEP in August for winter wheat at European flux sites (Ceschia et al., 2010) When considering heterotrophic respiration during the rest of the cropping year (after harvest), the additional simulated respiration represents 160 $\mathrm{gCm^{-2}}$. This leads to minimum annual NEP values



of -660 $\mathrm{gC\,m^{-2}}$, which is in the range of observed values for winter wheat in Europe (Ceschia et al., 2010) . High levels of heterogeneity are noticed in the retrieved maps while regional patterns are also noticeable. The uncertainty map shows the same regional patterns with values varying between 20 to 280 $\mathrm{gC\,m^{-2}}$ at 98 % confidence interval which corresponds to 3.4 % to 20.8 % relative variation on average. The large scale regional patterns will be analysed in Section 5.3.

The statistical distributions of the underlying parameters that were used to generate the maps in Figure 9 are a presented in

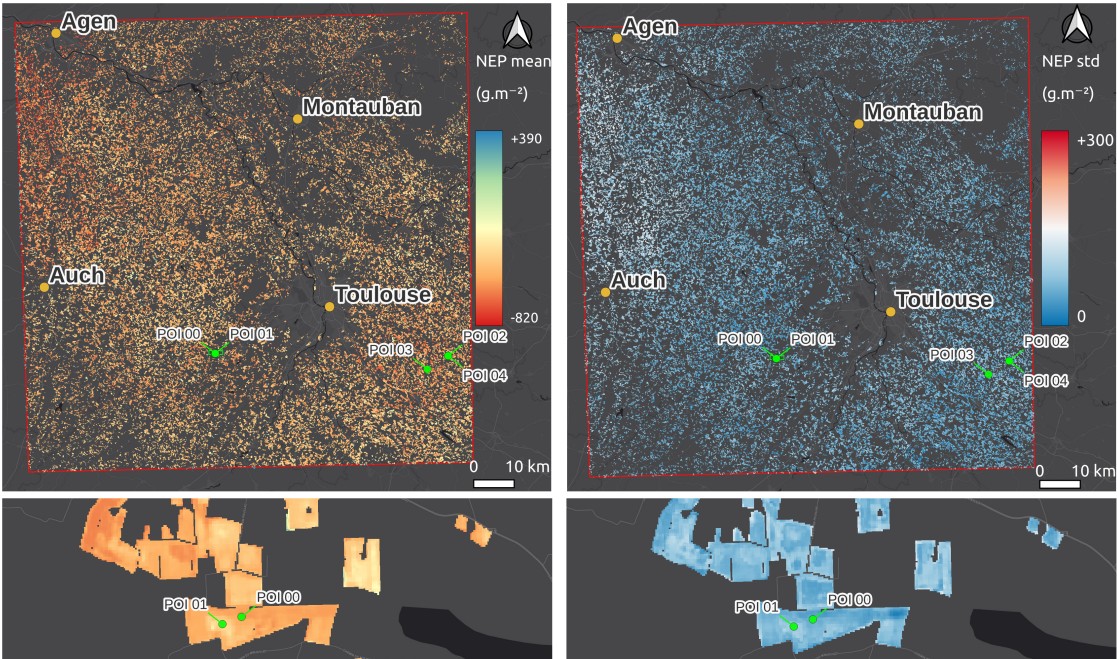

**Figure 9.** On the left, the NEP for the winter wheat fields for the 2016-2017 cropping season and on the right the corresponding uncertainties calculated with the AgriCarbon-EO processing by assimilating Sentinel-2 and Landsat-8 data (ACEO-S2L8-Pixel). The map also shows the position of selected Points Of Interest (POI) presented in Figure 11.


Figure 10 where the a priori and a posteriori parameters distributions are plotted in the diagonal subplots. The diagonal plots show that the a posteriori distribution is much less spread than the a priori distribution. This is mostly observed for LUEa which is consistent with the fact that light use efficiency as a crop parameter is not expected to vary strongly given that it corresponds to a crop/variety specific parameter. The a posteriori distribution of $emerg$ and $sena$, two parameters representing the emergence

and the start of senescence respectively, show a slightly bimodal behaviour that can be linked to the impact or cross impact of pedo-climatic conditions (e.g. warm/cold soil resulting in early/late emergence/senescence), agricultural practices (ex. early vs late sowing), and cultivars (early and late cultivars) across the modelled area. $PRT\_La$ and $PRT\_Lb$ a posteriori distributions are both unaligned with the a priori mean value of those parameters and $PRT\_La$ presents an asymmetry toward high values. Furthermore, the scatter subplots for $PRT\_La$ and $PRT\_Lb$ show a strong relationship between the combinations of the a





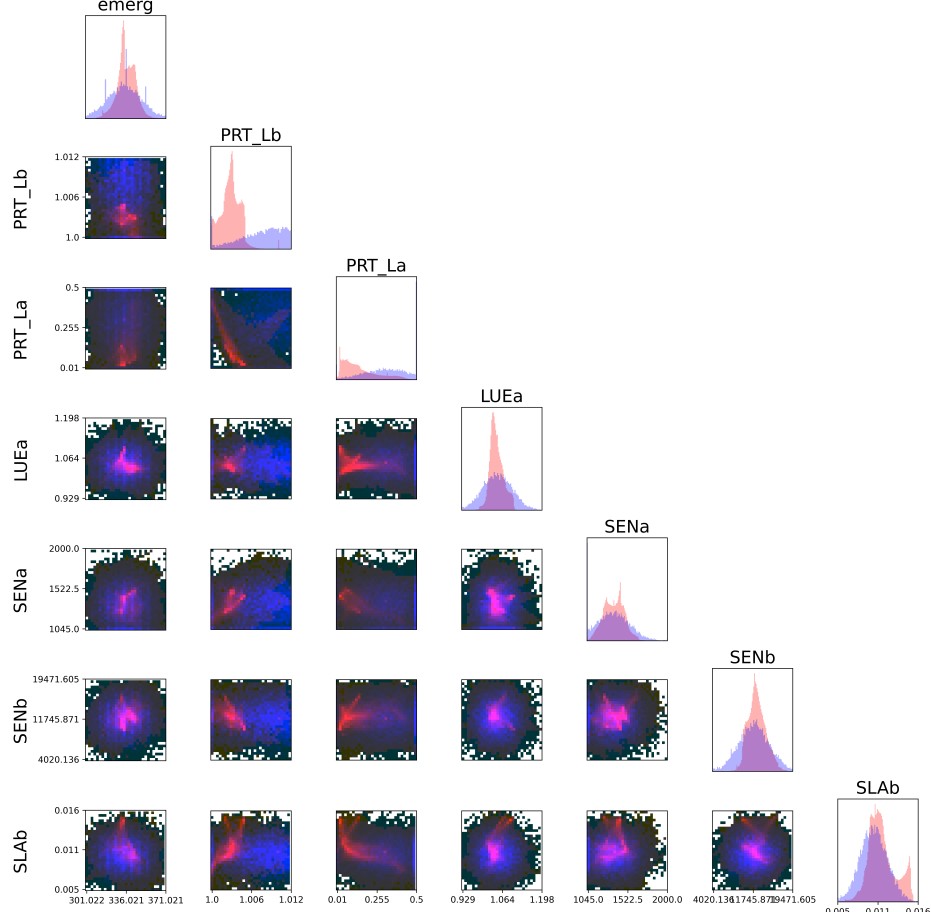

**Figure 10.** Bivariate distributions of the mean values of the sampled SAFYE-CO2 parameters at pixel scale. Blue and red histograms represent the a priori distributions and the a posteriori distribution's mean for each parameter respectively. The blue and red in the images represent the a priori and the a posteriori mean distributions for a pair of parameters.

posteriori mean parameters as already pointed out by Duchemin et al. (2008) and Battude et al. (2017). Milder relationships are also observed between $sena$/ $PRT\_La$, $sena$ / $PRT\_Lb$, $senb$ / $PRT\_La$, and $senb$ / $PRT\_La$.

     To illustrate the impact of parameter combination on the $GLAI$ dynamic and singular anomalies in the simulations that don't stand out in the regional statistics, results over selected sets of POI are presented in Figure 11. The points are selected to illustrate intra-field heterogeneity and specific anomalies. The locations of the POI are shown in Figure 9. Figure 11 (a-e)

shows in green the $GLAI$ inverted using PROSAIL with their respective uncertainties, and the simulated $GLAI$ time series in red with a higher transparency for the solutions with lowest contribution (likelihood). For instance, the results in Figure 11 (a) POI-00 and (b) POI-01 show the fitting of the model over two pixels in the same field (FR-AUR). It is clear from the observed and the retrieved $GLAI$ between the two POIs that the vegetation phenology is quite different with early emergence



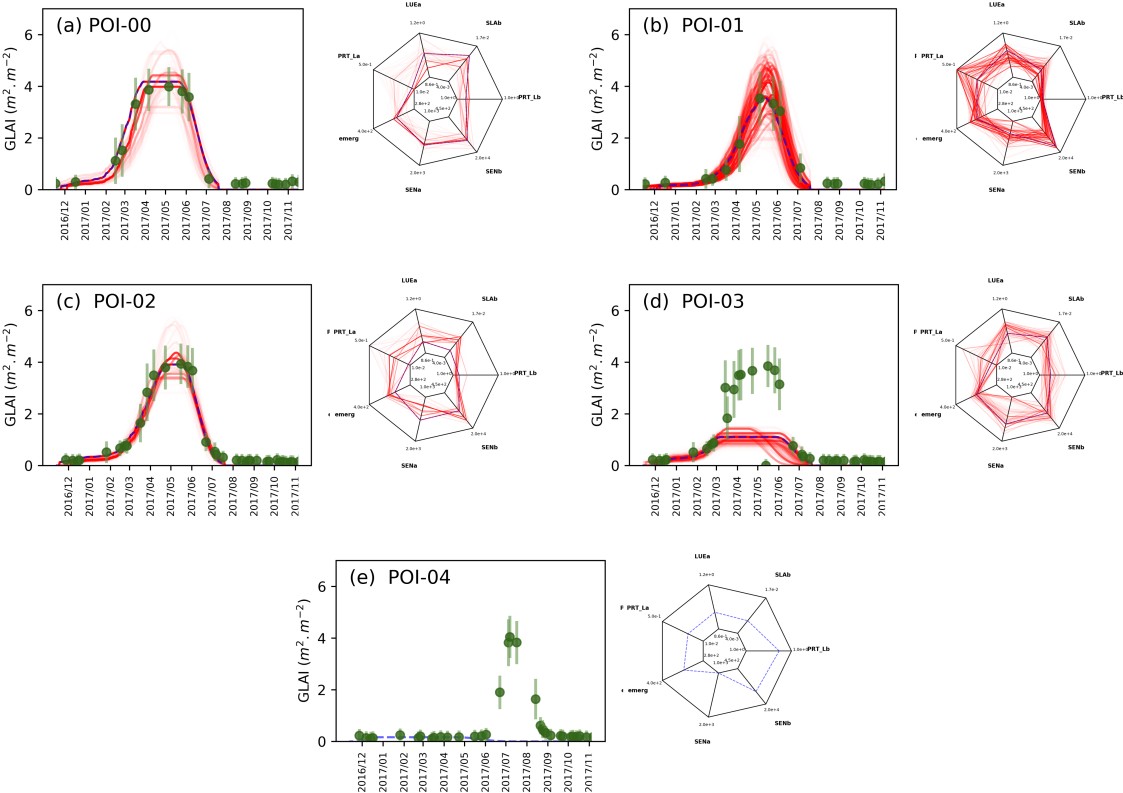

**Figure 11.** Time series of $GLAI$ (in green the PROSAIL predictions and in red the SAFYE-CO2 simulations ), and radar plots for the SAFYE-CO2 sampled parameters for 5 points of interest (POI 00 to 04) selected to represent particular situations observed in the simulated tile. In the radar plots, the blue dashed line represents the maximum likelihood estimation and the red lines represent the ensemble with the relative likelihood expressed through transparency. POI-00 (a) and POI-01 (b) are located at the FR-AUR field. POI-02 (c) and POI-03 (d) are adjacent pixels where a cloud date is not filtered in (d). POI-04 (e) illustrates either an error in the CAP declaration or a failed wheat crop followed by a summer crop (POI 04).

and higher maximum $GLAI$ in the case of POI-00 (a) and later emergence and lower maximum $GLAI$ in the case of POI-01

(b). Also, Figure 11 (c) POI-02 and (d) POI-03 are adjacent pixels over the same field, but each on the limits of the cloud mask in May 2017. In fact, the input $GLAI$ from PROSAIL on this date is associated with a very low uncertainty which impacts the retrieval of the SAFYE-CO2 model. The low uncertainty will result in a high level of information for this date, which negatively impacts the Bayesian inversion and reduces the SAFYE-CO2 model performances pushing the model to better fit this unrealistic inversion of $GLAI$. Finally, Figure 11 (e) POI-04 corresponds to a pixel in a field where the RPG declaration of

winter wheat is not consistent with the observed $GLAI$ that fits better with the $GLAI$ dynamic of a summer crop. The model shows "no fitting" in this case.





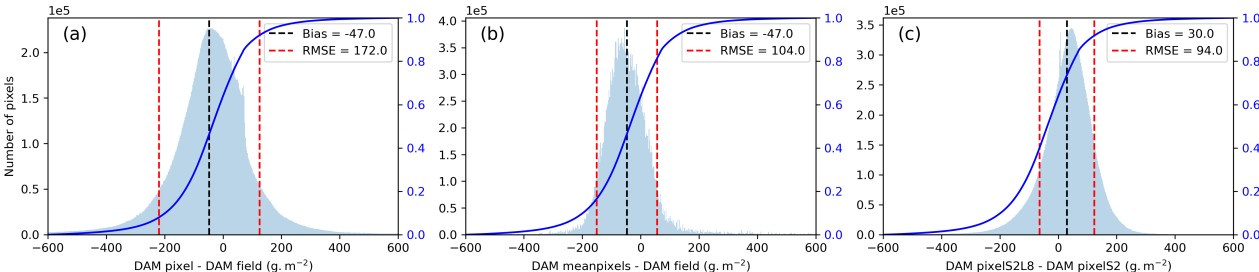

**Figure 12.** Histogram (left y-axis) and cumulative density function (right y-axis) of the bias of biomass at harvest (y-axis). (a) corresponds to ($DAM_{\mathrm{ACEO-S2L8-Pixel}}$ - $DAM_{\mathrm{ACEO-S2L8-Field}}$), (b) ($DAM_{\mathrm{ACEO-S2L8-Mean}}$ - $DAM_{\mathrm{ACEO-S2L8-Field}}$) and (c) ($DAM_{\mathrm{ACEO-S2L8-Pixel}}$ - $DAM_{\mathrm{ACEO-S2-Pixel}}$).

## 5.2 Impact of the spatial resolution and temporal sampling of assimilated GLAI

The AgriCarbon-EO simulations (Table 3) at different scales (*i.e.* pixel vs field) and for different satellite image temporal densities are compared to investigate the benefit of assimilating high resolution multi-mission derived $GLAI$ into SAFYE-
CO2. The impact of the spatial scale of the $GLAI$ assimilation is illustrated by Figure 12 (a) which shows the histogram of ($DAM_{\mathrm{ACEO-S2L8-Pixel}}$ - $DAM_{\mathrm{ACEO-S2L8-Field}}$). An average negative bias of -47 $\mathrm{g\,m^{-2}}$ is observed for the $DAM$ with a spread between -210 $\mathrm{g\,m^{-2}}$ and +120 $\mathrm{g\,m^{-2}}$ for the $[-\sigma, +\sigma]$ interval, when comparing the pixel scale simulation to the field scale simulation. This result is interpreted as the error in bias induced by considering field scale simulations in the crop model which can be avoided by applying a pixel scale assimilation scheme as in AgriCarbon-EO. Note that the same bias value is
obtained for Figure 12 (b) that represents the difference between the averaged pixel at field scale and the field scale simulations: ($DAM_{\mathrm{ACEO-S2L8-Mean}}$ - $DAM_{\mathrm{ACEO-S2L8-Field}}$). This is mathematically expected as $DAM_{\mathrm{ACEO-S2L8-Mean}}$ is obtained by averaging the $DAM_{\mathrm{ACEO-S2L8-Pixel}}$ simulations. However, when comparing the RMSE values between Figure 12 (a) and (b) a noticeable change in RMSE of -68 $\mathrm{g\,m^{-2}}$ is depicted. This result shows that the variability will decrease by 39 % when considering field scale modelling. The variability is directly influenced by the retrieved parameters of the crop model between
the intra-field and field scale for the same crop cycle; resulting in a different a posteriori parameter distribution as shown in the section above. Figure 12 (c) shows the difference between a simulation using only S2 and using S2 + L8. Adding L8 images tends to slightly increase dry biomass with a bias of 30 $\mathrm{g\,m^{-2}}$ and a RMSE of 94 $\mathrm{g\,m^{-2}}$. This difference is caused by the additional samples added at start and end of the vegetation cycle that result in a change in the length of the vegetation cycle. To extend the results at a broader perspective, the $DAM$ outputs from ACEO-S2L8-Pixel are analysed in terms of the number of
images over each pixel. Figure 13 shows the impact of the number of $GLAI$ observations per pixel on $\mu$ and $\sigma$ of DAM. $\sigma$ of $DAM$ decreases by about 66 % with the number of observations (146 $\mathrm{g\,m^{-2}}$ for 11 images to 48 $\mathrm{g\,m^{-2}}$ for 28 images) while $\mu$ $DAM$ values stay stable. This illustrates the stability of $\mu$ values given the range of variation of observed images. However, the decrease of $\sigma$ also illustrates the contribution of the number of images to the constraining of solutions and increased accuracy.



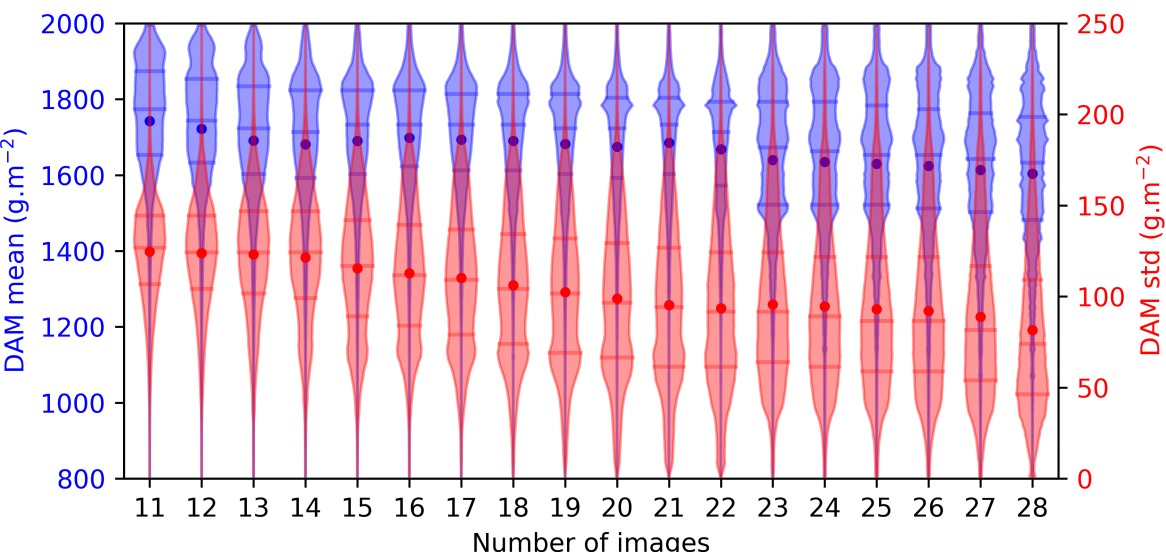

**Figure 13.** Violin-plots of the number of images used for the inversion over each pixel in the x-axis and the mean ($\mu$) $DAM$ on the right y-axis and the standard deviation ($\sigma$) of $DAM$ on the left y axis.

### 5.3 Regional scale analysis

In this section, the results of AgriCarbon-EO are analysed in relation to potential causes of variability at large and small scales to verify the coherency of the outputs from AgriCarbon-EO based on ACEO-S2L8-Pixel simulation in 2017. Maps of the retrieved parameters and final $DAM$ are generated by applying a Gaussian smoothing with a correlation length of 2.5 km to extract large scale patterns. This scale has been chosen to take into account the values of more than 10 fields which tends to average variety and agricultural practice effects and highlight non field dependent local tendencies. The differences of the

raw 10 m resolution ans smoothed images gives us the local anomaly. The anomalies highlight the small scale variations that correspond to intra-field and inter-field variability. Figure 14 (b) is produced from the input land cover maps and shows the density of winter wheat fields over the region where the two main winter wheat regions: they correspond to the hilly areas located South-East of Toulouse and to the Gers department (West of Toulouse). In the valleys, winter wheat is less present as irrigated summer crops (e.g. maize, sunflower) are grown by the farmers. The low winter wheat density around Toulouse

agglomeration is also spotted (Figure 14 - a). Figure 14 c to e are obtained from AgriCarbon-EO outputs and they all show large scale spatial patterns. Figure 14 (c) shows that the winter wheat biomass in the far west part of the Gers region was higher than in the other regions which is consistent with the 2017 yield statistics of the ministry of agriculture in France.

To investigate the large scale paterns in the simulations, $DAM$, $emerg$ and $sena$ are compared to pedological and topographic spatial maps. Figure 15 (a), shows in a ternary plot, the increased $DAM$ and earlier emergence dates with the increased

clay content which can be explained by a higher soil water holding capacity. On the other hand, low $sena$ values can be ob-



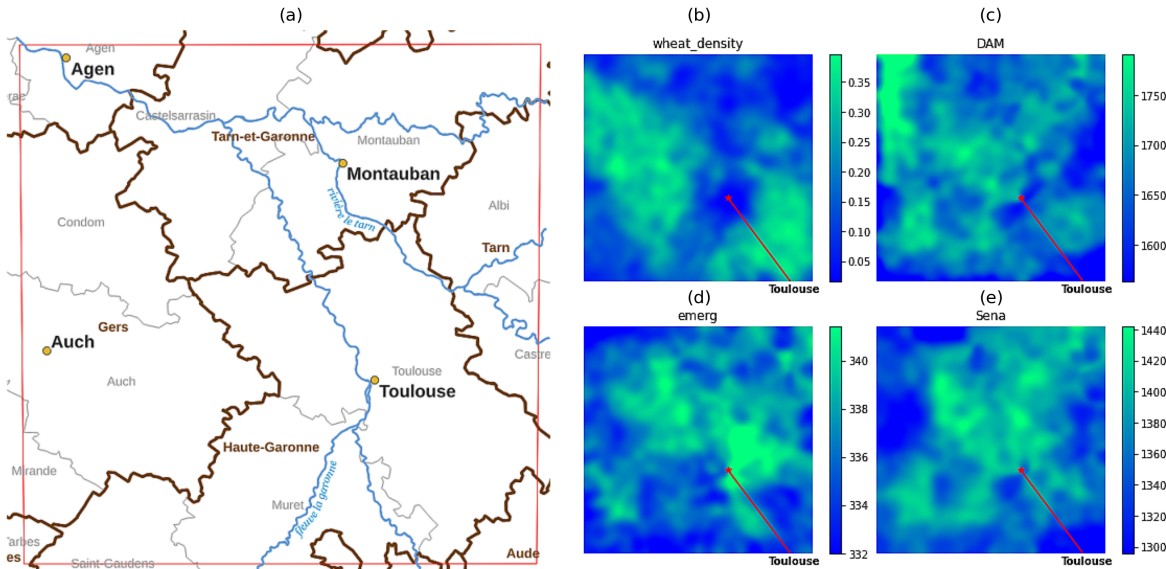

**Figure 14.** Hydrographic and administrative maps (a) next to Gaussian smoothed maps with 2.5 km correlation length for wheat fields density, biomass ($DAM$), emergence date ($emerg$), and start of senescence ($sena$) (b,c,d,e), respectively.

served for soils with low clay content which is consistent with the fact that early senescence can be caused by a lack of water availability. Finally, the emergence date does not seem much affected by the soil texture. Figure 15 (b) shows, in histogram the altitude for positive and negative anomalies of $DAM$, $emerg$ and $sena$. A higher number of winter wheat pixels at high altitude tend to emerge earlier than pixels at lower altitudes. This can be seen intuitively given the altitudinal temperature

gradient, however this difference may be caused by hill shading effects and hills top effects that allow the higher points to receive more sunlight and wind drying the soils which may promote an earlier warming of the soil and thus germination and emergence. The senescence is also slightly delayed with altitude and the biomass seems to be higher at low altitudes. This later observation can be explained by the fact that in those hilly landscapes, soil depth is smaller at the hills top compared to the bottom of the slope because of soil erosion. It results in low soil fertility and water holding capacities on the hill tops that may

reduce biomass production and increase stress which can hasten senescence. Figure 15 (c) shows in radial plots the distribution of expositions (*i.e.* N, W, S, W) given positive and negative anomalies for $DAM$, $emerg$ and $sena$ considering slopes greater than 5%. The radial axis corresponds to the density (*i.e.* a normalised number of pixels). The orange and blue lines correspond to pixels exhibiting positive and negative anomalies respectively. From this figure, it can be observed that the slopes in the region are mainly oriented South-West and North-East. Also, the model's output shows that the north eastern exposed fields

produce less biomass than the regional mean, while the southern fields show higher biomass. This is consistent with the fact that in the northern hemisphere, northerly exposed slopes receive less incoming solar radiation than the southerly exposed ones, which affects crop photosynthesis and temperature stress. Also differences in the dates of emergence and senescence can




be observed between the southerly and northerly exposed areas. The southerly exposed areas tend to emerge earlier and enter
in senescence earlier (at a lower degree days values) because of the higher exposition to incoming solar radiation and higher
temperatures. Soil on the southern slopes tend to be warmer in the winter which allows an earlier germination and also tend to
be drier and hotter during the summer which can also push forward development and senescence.Yet it is also possible that the
sum of temperature at senescence is less variable in reality and that the observed variability translates in part to the difference
between the real local micro-climate and the 8 km Safran products.

## 6  Discussion

In this section, we discuss the outputs and limitations of our approach from the perspective of the challenges for monitoring
the components of the carbon budgets and for broader agronomical diagnostic studies.

### 6.1  Multi-mission data, cloud cover, and limitations

The retrieval of SAFYE-CO2 parameters in AgriCarbon-EO relies on the accuracy and availability of EO data that is hampered
by the errors in image co-location, the atmospheric corrections, the presence of clouds, and the cloud shadow correction. Many
studies show that these effects have an important impact in agricultural remote sensing applications like yield estimation
(Soriano-González et al., 2022), land cover (Song et al., 2021)), or superficial soil carbon content mapping (Vaudour et al.,
2019). In our study, we show that these effects are mitigated through the use of a Bayesian approach in a multi-temporal
context because the uncertainty on the EO derived $GLAI$ are accounted for in the assimilation process. Our approach shows
that increasing the number of observations does not strongly impact the mean $DAM$ values, but increases its uncertainty by
about 66 %. Nevertheless, unfiltered clouds or the lack of images significantly impact the simulations locally (Figure 11 (c)).
The use of additional data from Landsat-8 enhanced simulation quality for our region of interest. Additional optical or even
SAR satellite data could mitigate loss of data due to cloud cover in northern and coastal regions (Veloso et al., 2017; Fieuzal
et al., 2017). The Improvements in cloud detection algorithms will highly benefit our approach (Skakun et al., 2022). The
analysis of $GLAI$ time series to detect anomalous variations (Figure 11 (d)) could also be an option to filter clouds.

### 6.2  Importance of high spatial-resolution

Intra-field heterogeneity is a well established issue in agricultural applications (Weiss et al., 2020; Blackmore et al., 2003;
Grieve et al., 2019; Nowak, 2021), but it has not been thoroughly treated in terms of $CO_2$ fluxes and uncertainties estimates.
In this paper, we argue that reliable and accurate estimates of $DAM$ and $CO_2$ fluxes in support of carbon budget components
monitoring require intra-field scale estimates. Our results show that by assimilating mean field level $GLAI$ products in SAFYE-
CO2 a bias of -47 $\mathrm{g\,m^{-2}}$ and an artificial relative uncertainty decrease of 39 % will be induced compared to assimilating high
resolution $GLAI$ and calculating the mean on the model's output. So high resolution allows more accurate estimates of the
mean $DAM$ values at field scale which enables more accurate field scale estimates of SOC changes by soil models in the
perspective of monitoring SOC stock changes. Still, higher resolutions may need to be investigated to address very small or



## (a) Textures

## (b) Altitude

## (c) Exposition

DAM emerg Sen_a

**Figure 15.** Impact of texture (ternary plots - a), altitude (histograms - b), and exposition (radial positive and negative anomaly plots - c) on biomass ($DAM$), emergence date ($emerg$), and start of senescence ($sena$). In figures (b) and (c) the orange and blue curves correspond to the ensemble of pixels exhibiting positive and negative anomalies, respectively.



elongated fields like for instance in India (Deininger et al., 2017). The other input data products that are driving the spatial
resolution of the AgriCarbon-EO outputs are the land cover and the weather data. While land cover is available at an adequate
resolution (*i.e.* field sale), it is error-prone, either because of erroneous declarations in the RGP (Magnin, 2019) or because of
classification errors when EO based land cover maps are used (Liu et al., 2022). Interestingly, our results show that when a
mismatch occurs, the fields in question exhibit high anomalies in retrieved parameters and are thus detectable. For the weather
forcing, the current application was based on the Météo-France 8 km resolution Safran data which provides reasonable accuracy
over France (Garrigues et al., 2015). Currently, ECMWF provides ERA5-Land at 9 km resolution globally (Muñoz-Sabater
et al., 2021), and NOAA provides HRRR at 3 km over US (Dowell et al., 2022). In the future, coverage and resolutions
of weather forcing data are expected to increase (*i.e.* ERA6 at 2.5 km). Increasing the resolution of the weather forcing in
AgriCarbon-EO would provide better spatial information, but would also increase the computational demand by a factor $\gamma$ as
the LUT for SAFYE-CO2 are generated over the weather grid (Eq. 28).

$$\gamma = \frac{TLUT \times 8^2}{\theta^2} \qquad\qquad (28)$$

Where $TLUT$ is the processing time for the generation of LUT and $\theta$ the weather grid resolution in km. Finally, the use of
high resolution estimates is also of interest for the large scale (regional) trend analysis (Makowski et al., 2014). Ten meter
resolution simulations virtually eliminate mixed pixels encountered in coarse resolution approaches while maintaining the
effects of non-linearity that we depicted in the pixel to field analysis (Figure 12). High resolution simulations also allow the
analysis of variability at different scales through smoothing or aggregation, enabling the assessment of the effect of different
environmental co-variables and an easy inter-comparison of this approach with information at coarser resolution.

### 6.3    Limitations of the Bayesian and physically based approach

While the components of AgriCarbon-EO have been tailored to the requirements mentioned in the introduction (large scale,
high resolution, uncertainty estimates, and biophysical processes), we showed limits for each of them. For instance we show
that the BASALT Bayesian approach can be sensitive to an erroneous observation associated with a low uncertainty (Figure 11
d). A trade-off has to be made between the variability of the generated solutions, and the number of LUT entries to maintain
computational efficiency. A solution could be to consider a joint distribution for prior parameters to propose a better ratio of
appropriate solutions(Figure10),(Wang et al., 2022). On one hand, the radiative modelling is constrained by the spectral library
database (Verhoef et al., 2007), which may not reflect ground conditions like presence of weeds impacting $GLAI$ retrievals.
On the other hand, the crop model predictions will depend on fixed and prior parameters of a given crop, and the a posteriori
parameters distribution. Note that ixed prior parameters need to be chosen through agronomic knowledge and bibliographic
evidence. Alternatively, we could have reverted to machine learning approaches that have gained in popularity recently for
precision agriculture and soil carbon farming application (Sharma et al., 2021). But, while they are powerful tools to extract
the most of existing training data, they are also constrained by the extent of the observation dataset. Also, Machine Learning
(ML) approaches predict a limited number of variables at once and may need regular retraining to take into account changing
climatic conditions and management practices. Therefore physically based approaches are still needed (Vereecken et al., 2016).



In the future, if the confidence in this approach increases, surrogate ML models could be a good option to consider in order to reduce computational needs for large scale carbon budget components monitoring (Wolanin et al., 2019). In the current state, it is reasonable to consider that an MRV platform for SOC carbon stock changes shall include ensemble approaches (direct measurements, ML, statistical, physical models) with varying levels of complexity and involving a diverse array of stakeholders (Nevalainen et al., 2022) (e.g. Tier 1,2 and 3), similar to what has been implemented in the IPCC approaches (Parker, 2013). AgriCarbon-EO is designed with the objective to implement some of these solutions.

### 6.4 From AgriCarbon-EO to carbon budget

The present approach provides high resolution estimates of key carbon budget components using a soil respiration module, in the SAFYE-CO2 crop model, that is decoupled from the soil carbon reservoirs. This methodology is adapted for short term ,large scale assessment of the carbon budgets (typically one year) (Pique et al., 2020a). In fact, soil processes that affect soil organic matter mineralisation at longer time scales, and stock dependent processes, such as the priming effect, are not accounted for here. The inclusion of a soil carbon decomposition module as in Guenet et al. (2016), that describes the different active and stable carbon pools as well as mineralisation and humification of the carbon imports is needed to diagnose long term soil carbon trends. Such exercise, requires in addition to the new soil module parameters, input dataset on initial soil carbon content and organic organic amendments which is challenging for large scale applications. One way of achieving this is to take advantage of the rapidly developing Farm's Management Information Systems (FMIS) and enhanced soil properties maps through Digital Soil Mapping (DSM). Even though FMIS data are not easily accessible, it is expected that this limitation will be reduced with the development of soil carbon farming policies (like the Label Bas Carbon) and the auditing schemes (de Gruijter et al., 2016). Such data exchange will have a dual positive effect providing that adequate soil sampling protocols are applied. The SOC data will augment the data collection available for validation and verification of tools like AgriCarbon-EO, and at the same time, approaches such as AgriCarbon-EO may provide optimal sampling strategies for the estimation of SOC stock changes for carbon auditing.

### 7 Conclusion and outlook

This paper presents the AgriCarbon-EO processing chain that assimilates remote sensing data into the PROSAIL radiative transfer model and the SAFYE-CO2 crop model to estimate some of the key carbon budget components of crop fields at high resolution and regional scale. AgriCarbon-EO was designed to cover essential features to comply with the monitoring component of the MRV systems for cropland carbon budget:

1. Scalability, which is of major specification in the design of AgriCarbon-EO. The proposed assimilation scheme has been constructed to prevent the time related drawbacks of iterative methods.

2. Uncertainty of the model's variables, is estimated using an innovative Bayesian approach labelled BASALT.



3. Native high resolution modelling by assimilation of EO data at 10 m resolution enabling a more precise evaluation of field caracteristics, a coherent resolution with verification data, and the means to enhance in-situ soil and vegetation sampling.

4. Readily operational tool, by the use of accessible remote sensing, weather and ancillary data in an end-to-end processing chain.

After detailing the processing chain algorithm and formulation, we presented validation and analysis results, in a multi-scale temporal and spatial framework over several numerical experiments. When validating the outputs against flux tower measurements, we find that the new inversion approach (BASALT) produces reliable estimates of $Co_2$ fluxes, $NEE$, $GPP$ and $Reco$ and with similar performances compared to previous studies while providing their associated uncertainties. Our estimates for dry aboveground biomass $DAM$ were close to the observations while the validation exercise for yield was less conclusive due to the uncertainty of the combine harvester's data and processing, and/or the use of a Harvest Index to estimate yield that may not allow the account of essential drivers of yield. When applying AgriCarbon-EO at large scale, we show its ability to reproduce the high variability of the outputs at local scale while showing coherent patterns at regional scale when considering texture, altitude and exposition. Our analysis of the impact of the number of remote sensing acquisitions show a reduction in uncertainty of 66 % when full S2 and L8 data is available while the median retrieved $NEE$ and $DAM$ remained the same. Also, we assessed the importance of the temporal sampling of remote sensing images not only in increasing the robustness of the retrievals by reducing the impact of errors from cloud cover, but also in decreasing the retrieval uncertainty in the Bayesian framework. Also we find that the assimilation of field scale $GLAI$ products induces a bias on the $DAM$ of -120 to 210 $\mathrm{g\,m^{-2}}$ and a reduction of the $DAM$ inter-field variability of about 39 % compared to pixel scale assimilation. Based on this we argue that an intra-field scale quantification of the $DAM$ and of the biomass that returns to the soil is needed for accurate monitoring of soil organic carbon stock changes. Assessing the carbon budget components and the SOC stock changes at this resolution provides 1) a coherent spatial information with soil samples. 2) the means to provide better sampling strategies forsoil and plants in MRV approaches. In the future, we aim to provide further constraints to the assimilation scheme through the inclusion of additional remote sensing datasets from SAR and other variables derived from optical sensing. Such information will reduce the impact of cloud coverage and the equifinality in parameter retrievals. It is notable that AgriCarbon-EO can also provide variables related to the water cycle such as evaporation, transpiration, drainage as well as soil water content. In summary further studies will thus aim to use AgriCarbone-EO as a coherent and multi-criteria full crop cycle agronomic diagnosis tool for production, carbon, phenology and water use.

*Author contributions.* TW and AA proposed the methodology. TW, AA and LA developed the chain code. TW and AA conducted the simulations and the visualisations. TW, AA and EC conducted the analysis. EC and AA provided funding acquisition and supervision. TW and AA prepared the manuscript. EC, LA and RF provided comments on the manuscript. All Authors agreed on the proposed paper.



*Competing interests.* The contact authors declare that neither they nor their co-authors have any competing interests.

*Acknowledgements.* Data acquisition at FR-Lam and FR-Aur were mainly funded by the Institut National des Sciences de l'Univers of
the Centre National de la Recherche Scientifique (CNRS-INSU) through the ICOS and OSR SW observatories. We thank Jean-François
Dejoux, Tiphaine Tallec, Franck Granouillac, Nicole Claverie, for their technical support. We extend special thanks to Mr. Andréoni (farmer)
for accommodating measurement devices in their fields at FR-AUR. T. Wijmer thesis was financed by the NIVA project from ASP and
"Naturellement Popcorn". Financial support was also obtained from the ERANET ANR SMARTIES project, the Horizon Europe ClieNfarm
(n° 101036822) and ORCaSa (n° 101059863) projects and the Bag'ages (Agence de l'eau Adour Garonne) projects. This work was granted
access to the HPC resources of CALMIP super-computing centre under the allocation 2022-P20013.

*Code and data availability.* Source of datasets and codes is given hereafter.

Datasets:

1. Remote sensing data for Sentinel-2 and Landsat8 using the maja processing are downloaded from THEIA : https://www.theia-land.fr/en/product/sentinel-2-surface-reflectance/. The Sentinel-2 level 2A and Landsat8 L2A data are distributed under the ETALAB V2.0 open license.

2. Land Cover datasets are available at : https://geoservices.ign.fr/rpg

3. Validation datasets are available from the SIE website : https://sie.cesbio.omp.eu/

4. Soil texture are available at: https://www.isric.org/explore/soilgrids : https://doi.org/10.17027/isric-wdcsoils.20190901

5. DEM dataset are available at: https://www.eea.europa.eu/data-and-maps/data/eu-dem Outputs:

6. Full dataset of all simulations is about 5T of memory, selected outputs can be made available upon request to the authors.

7. Output maps for Wheat 2017 are available at : 10.5281/zenodo.7534280

Code availability:

AgriCarbon-EO is implemented in python3. Agricarbon-EO requires the PROSAILv5 python package and the SAFYE-CO2 v2.0.5 python implementation. AgriCarbon-EO v1.0.1 is available free of charge for research and evaluation purposes (non-commercial) upon signature of a license agreement with Toulouse Technology Transfer (TTT) office of Université Toulouse 3.

For this, the user contacts the TTT at "contact@toulouse-tech-transfer.com" providing his contact information, affiliation, and objective of use. Upon validation of the license, the code is provided by the team at CESBIO. SAFYE-CO2 v2.0.5 is provided with AgriCarbon-EO v1.0.1 in this same procedure. Note that for this paper, and in compliance with the journal requirements, an anonymous procedure was put in place to grant access for the reviewers. PROSAIL: python bindings v2.0.3 for PROSAIL5 is hosted at https://github.com/jgomezdans/prosail and archived under https://zenodo.org/record/2574925#.Y-lIVK3MI2w by Dr.José Gómez-Dans.



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
