# Peer review of "AgriCarbon-EO v1.0.1: Large Scale and High-Resolution Simulation of Carbon Fluxes by Assimilation of Sentinel-2 and Landsat-8 Reflectances using a Bayesian approach"

_EGUsphere, 2023_

## Referee Comment (RC1)

**General Comments**

The five authors have submitted a rather long manuscript (approx. 13.500 words excluding references), in which they advertise an approach for improved carbon balance mapping in agricultural soils. Long-term storage of carbon in agriculturally used soils is a very hotly discussed topic in the frame of carbon farming, which again is part of the EU green deal. Finding new ways to quantify soil carbon fluxes in agricultural systems therefore is a very interesting and promising research topic!

If we take a closer look on the manuscript, however, it turns out that the proposed approach is more a method for producing yield maps for winter wheat from correlations with green LAI observations from Earth Observation time-series. As, by using a simple LUE-based growth model and neglecting water-stress effects, the interrelations of the carbon and water cycle are explicitly not considered in this study, so that mass and energy balance may not necessarily be closed and no direct link between atmospheric carbon dioxide concentration and carbon sequestration in agriculturally used soils is established, I have the feeling that what we as a modelling community can learn from the presented study for carbon-farming related questions unfortunately is limited.

The paper consists of at least five major parts/questions, whereby each of the topics potentially would provide substance for individual articles.

First, the model and the assimilation approach are introduced and the system is applied to field scale simulations, which are validated against destructively measured biomass and yield data. Thereby, ESU samples and combine harvester data are used. In figure 6 it can be seen that the ESUs were sampled on different dates throughout the growing period of 2017, while in 2019 only one date was sampled. Unfortunately, this is not clearly described in the text (e.g. sampling dates are not explicitly mentioned/listed) and confuses the validation of the temporal and the spatial performance of the algorithm. For example, in Figure 7 it remains largely unclear if the good performance of the year 2018 can be traced to the fact that the algorithm well follows the temporal dynamics of biomass accumulation (as biomass is suspected to continuously increase over the growing season, $R^2$ correlation will necessarily be high…), or if the good performance is due to a good spatial mapping of heterogeneities of yield. Especially for 2019, where only one in-situ sampling date was available (as I deduce from Figure 6), the correlation is extremely poor, as the algorithm returns constant values of 2000 g/m² , while the in-situ data show a wide range of values. The way that I read the validation figures, this leads to the conclusion that the spatial heterogeneities of yield cannot be reasonably mapped with the proposed approach. This is confirmed then by Figure 8 also for the intra-field scale, where very poor correlations between the spatial combine harvester measurements and the satellite-based yield product become evident. For a satellite-based approach that explicitly targets the monitoring of intra-field variability in the context of precision farming measures, as claimed in the introduction, this is rather poor.

Following the field-scale validation, the paper takes a sudden turn towards large-scale simulations and shows the application of the method for retrieving Net Ecosystem Production for a 110 x 110 km large scene for the growing season of 2017. Obviously, only the winter wheat pixels were investigated, although this is not clearly stated in the text. In this section, Figure 11, although suffering from some stylistic errors, makes the assimilation procedure and potential pitfalls of the algorithm transparent, so that the readers get a clear picture of how the Bayesian approach works. However, as the maps shown in Figure 9 due to the color stretch do not allow for the detection of large-scale patterns and also because these patterns are not discussed in the text, the readers wonder why this had not already been explained as part of the intra-field scale validation and why the jump to the large-scale actually was required for the purpose of this paper.

The next section opens two further side-questions about the impacts of spatial resolution and temporal sampling frequency (from my perspective these are questions three and four of the paper).

The impact of spatial resolution is assessed by determining the bias between field averages and pixel-based values. I think that we no longer need to prove that intra-field variability indeed plays a major role. From my perspective it would have been more interesting to analyze, in a proper quantitative way, how much detail is lost when going from high-resolution of 10 m to a more modest resolution of 30 m, which widens the possibility for multi-mission observations. I think that in view of future Sentinel-Missions, which potentially will even provide higher resolutions of up to 5 m, this discussion is relevant. A similar drawback from my point of view is that the analysis of observation frequency impact on the DAM simulations is limited to the absolute number of satellite scenes and does not account for the differences in spatial resolution between Landsat and Sentinel-2. Also, the analysis does not consider the impact of observations that happen to cover specific growth stages. Satellite observations at specific growth stages might benefit the retrieval accuracy of certain parameters. By simply correlating the satellite images to the in-situ-sampled yield maps, it can be found that the pronunciation of intra-field patterns should high during BBCH 70-89 and reduced during the bolting phase. The occurrence of cloud cover during these phases may be highly specific for the region and may impact the application of the approach differently in different parts of the world.

The fifth part of the paper then widens the scale even further up to the regional level by filtering the spatial results to correlation lengths of 2,5 km, thus entering a spatial scale beyond individual fields. The found large-scale patterns are explained by the soil characteristics in terms of water holding capacity and by the terrain situation (elevation, slope, aspect). While this undoubtedly explains the found spatial patterns, it remains unclear what the contribution of this section to the overarching subject of the study actually is.

The last part of the paper is dedicated to the discussion, which, in comparison to the size of the rest of the paper, is rather short (three pages). Here, the authors - among other recommendations - suggest the use of SAR-data with their approach. As the proposed retrieval scheme which is based on the PROSAIL model does not apply to microwave data, it remains unclear how SAR data could successfully be integrated into the system. In the second part and again in the fourth part of the discussion, the authors try to link their study, which is on winter wheat yield n South-West France, to soil carbon processes in general. As the connection is vague and indirect, because the relevant soil processes were neglected here, these attempts come across as rather endeavored. The further discussion treats well-known basic facts about remote sensing, e.g. the respective tradeoffs and advantages of physical modelling and machine learning. In general, I think that the Bayesian approach with the associated uncertainties makes a lot of sense and I would have been very curious about the explanations why the system performed so poorly with respect to spatial patterns. The discussion, however, only traces this major drawback to uncertainties in the in-situ data (e.g. in the combine harvester measurements), which I do not find very convincing.

A short conclusions and outlook section that mainly summarizes the main findings again closes the manuscript. As the authors state correctly at the end of the paper, the approach could potentially be used as *"a coherent and multi-criteria full crop cycle agronomic diagnosis tool for production, carbon, phenology and water use".* It is a pity that the presented paper does not advertise this potential of integrated remote sensing supported modeling approaches, but focuses on only one variable (aboveground biomass alias yield) and from my point of view misses to ask the relevant questions for time-series from EO data (e.g. What is the additional value of 10 m resolution compared to 30m? What can be expected from future 5 m resolution data? What is the additional value of the spectral bands of Sentinel-2 compared to Landsat? What will be the impact of the new SWIR bands in the Sentinel-2 next generation for such integrated approaches? What does the interpretation of top of canopy spectral signals actually reveal about processes happening in the soil? etc.)

Overall, I think the fact that field scale temporal patterns, field scale spatial patterns, large-scale and regional scale modelling together with questions about spatial and temporal sampling density are all mixed in the paper, blurs the structure of the presentation and makes the manuscript a rather demanding and rather exhausting read. I would highly recommend focusing on fewer aspects, e.g.

maintaining sections 1-4 plus Figure 11 from section 5 and removing the rest. The link to the CarbonFarming buzzword seems artificial and should be removed or mitigated. Also, the large amount of typing, language and format errors is quite surprising, given the autocorrection capabilities of state-of-the-art text processing software (see specific comments below).

Before the potential further processing of the manuscript I recommend:

(i) to resolve the very large number of formal and spelling errors that prevent the readers from focusing on the content.
(ii) to decide, if indeed all the scales, the intra-field scale, the large scale and the regional scale, should be treated in a single paper.
(iii) to rethink the connection between the introduction that focusses on carbon farming and the actual content of the paper, which is more on yield modelling than on carbon sequestration.
(iv) to think about, if comparing different pixel resolutions would be more relevant than comparing field scale to pixel scale resolution.
(v) to analyze the impact of observations in specific phenological stages instead of taking only the absolute number of observations into account.
(vi) to rewrite the discussion so that not only very general aspects of remote sensing are discussed, but the approach is referenced against other studies/approaches in the same field and especially the poor performance with respect to spatial patterns at the intra-field scale is adequately explained.
(vii) to rephrase the rather general title so that the limitation of the study to certain crops (winter wheat), certain variables (biomass, yield) and the region (South-West of France) is reflected in the headline.

Yours sincerely

**Specific Comments**

**Abstract**

Line 1: Phrasing. I am not sure if "*mitigation solution*" is the right word here and would prefer "*mitigation strategy*" instead.

Line 2: Phrasing. I don't think that in-situ sampling is "*prohibitive*". It surely is extremely labor intensive and thus not feasible. But the main drawback in my opinion is that it will never be spatially continuous.

Line 5: Phrasing. Which kind of resolution is referred to in this sentence (spatial, temporal, radiometric…)?

Line 7: Phrasing. Please be consistent. Either use "*assimilate in*" or "*assimilate into*". I'd prefer the latter.

Line 8: Typo. "*transfert*" → "*transfer*"

Lines 10/11: Grammar. "*The chain considers as input a land cover maps, multi-spectral reflectance maps from the Sentinel-2 and Landsat-8 satellites, and daily weather forcing.*" → "*The chain considers land cover maps, multi-spectral reflectance images from the Sentinel-2 and Landsat-8 satellites, and daily weather forcing as input.*"
Line 11: Terminology. "*inversed*" → "*inverted*"
Line 14: Typo. "*agaisnt*" → "*against*"
Line 15: Question. I fear I don't understand what "*R² = 0.88 - 0.88*" means.
Line 16: Phrasing. "*We quantify the difference between pixel and field and pixel scale simulations…*" → "*We quantify the difference between pixel and field scale simulations…*"

**1 Introduction**

Line 24: Grammar. "*Agriculture and land use changes accounts for 15%...*" → "*Agriculture and land use changes account for 15%...*"

Line 39/40: Grammar. "*The need to monitor soil carbon at Farm level to inform individual farmers guide policies and development of carbon markets led…*" → "*The need to monitor soil carbon at farm level to inform individual farmers together with the development of guide policies and carbon markets led…*"

Line 42: Grammar. "*They rely on a soil centered quantification approaches which has limitations in terms of accuracy and reliability of the soil and biomass input data and a field scale resolution that often does not match the spatial resolution of in-situ soil and plant growth variability*" → "*They rely on soil centered quantification approaches which have limitations in terms of accuracy and reliability of the soil and biomass input data. Also, their spatial aggregation level on the field scale often does not match the spatial resolution of the soil and plant growth variability observed in-situ*"

Lines 46/47: Phrasing. "*These models include the main components of the cropland's carbon budget, plants photosynthesis, and respiration, emission due to soil organic matter mineralisation. These models can also account…*" → "*These models include the main components of the cropland carbon budget, photosynthesis and respiration, and emission due to soil organic matter mineralization. They can also account…*"

Line 52: Grammar. "*autotrophic respiration - Ra, heterotroph respiration - Rh*" → "*autotrophic respiration - Ra, heterotrophic respiration - Rh*"

Line 54: Grammar. "*This can results in high…*" → "*This can result in high…*"

Line 57: Phrasing. "*Getting hold on this information…*" → "*Getting hold of this information…*"

Line 59: Comment. *"…information about development dynamics…"* I think it would be important to highlight here that GLAI incorporates both, information on environmental growth conditions as well as information on human (management) behavior.

Line 61: Phrasing. The term *"restituted to the soil"* is somewhat misleading in my opinion, because the main issue here is the long-term storage of atmospheric carbon in the topsoil, which to the largest part comes from geological reservoirs and not only from agricultural soils and thus is not strictly all given back to the soils. I'd thus phrase it a little more neutral and just speak of *"biomass and carbon storage"* in the soil.

Line 64: Phrasing. Again, I am not so happy with the term *"prohibitive"*. I'd recommend changing *"often computationally prohibitive"* to „*computationally demanding"*.

Line 67: Typo. *"shemes"* → "*schemes*"

Line 72: Grammar. *"…radiative transfer model to Obtain GLAI at 10 m resolution, GLAI that is thereafter assimilated…"* → *"..radiative transfer model to obtain GLAI at 10 m resolution that is thereafter assimilated…"*

Line 79: Question. Again, as there are temporal (frequency of observation) as well as geometric questions (intra-field resolution) targeted in your study, what kind of resolution is focused here?

**2 Methods**

Line 84: Question. Is a daily time-scale adequately suited to model crop growth and potential stressors?

Line 85: Question. Wouldn't the assimilation scheme not also work for microwave data?

Line 96: Question. Is analyzing each image independently making full use of time-series of satellite data?

Line 99: Typo. *"Ligth Use Efficiency(LUE)"* → *"Light Use Efficiency (LUE)"*

Line 103: Typo. *"corpping year"* → *"cropping year"*

Line 105: Interpunctuation. *"…requirement of 5 GB per process, for the satellite images…"* → *"…requirement of 5 GB per process for the satellite images…"*

Line 119: Comment. I think it would be worth mentioning that parceled land use data are not available for many parts of the world. The requirement of parceled land use inputs limits the applicability of the AgriCarbon-EO approach to those areas where parcel information is available.

Figure 1: Typo. *"construt"* → *"construct"*

Line 123: Question. I understand that a UTM map projection corresponds to the Sentinel-2 imagery that is used in the approach. I just wonder, if staying with an equal-area projection would make sense here to facilitate the quantification of fluxes per area.

Line 137: Layout. Please avoid line-breaks between numbers and corresponding physical units.

Section 2.2.3: Question. I am surprised that neither wind velocity nor atmospheric carbon dioxide concentration are required as meteorological input. For a model that is targeting carbon farming applications, I would have expected a direct link between the water and carbon cycle to be present in the algorithm.

Line 149: Question. How does 8 km resolution weather data correspond to the intra-field geometric detail that is targeted in this study?

Line 152: Terminology. LAI is widely classified as a biophysical rather than a geophysical variable throughout the remote sensing community.

Line 155: Structure. Abbreviations such as SAIL should be explained when first mentioned in the text.

Line 159: Question. One is wondering why in a study about carbon storage instead of Prospect-5d not the most recent version PROSPECT-PRO is used, which specifically includes absorption coefficients for the carbon-based constituents of aboveground biomass.

Eq.1: Comment. I think it's a pity that such a "simple" LUE-model is used to describe carbon fixation at the land surface. There are gas-exchange models available that create a direct link between carbon and water cycles. For a study in carbon farming, I would expect a more complex approach.

Eqs. general: Structure. I am missing direct references from the text to the Equations. You have included these cross—references for some, but not for all Equations. I suggest going through the entire manuscript and adding direct references for every single Equation in the paper to avoid potential misunderstandings. Thereby, you should aim for a uniform layout and decide whether you'd like to write "Eq. (x)" or "(Eq. x)" (I'd prefer the latter).

Line 170/171: Question. What was the reason for ignoring water stress effects in the simulation? I think that very interesting findings can be made for example when the modelled biomass according to the natural water budget does not correspond to the biomass accumulation observed from satellites. Ignoring the water stress also means ignoring uncertainties in the soil parameterization. How does your model explain the differences found between simulation and satellite observation, if soil processes are ignored? If the model does not even try to explain them and simply accepts the observation and carries on, what can we learn about natural processes from such a model?

Line 175: Question. Is a multiplicative factor well suited to describe the temporal dynamics of senescence? A multiplicative factor will result in a rapid decrease of "greenness" at the onset of senescence, while the increments of the greenness-decrease will become smaller as senescence progresses. From my experience, S-shaped sigmoid functions better correspond to the dynamics of senescence that are observed in the field. The model results shown in Fig. 5 also do not look like as if a constant senescence factor was applied. Could you please explain?

Eqs. 6 and 7: Comment. From my perspective, the maintenance respiration should be connected to the tissue that already has been accumulated which requires "maintenance energy". I don't see a link to the accumulated biomass here.

Eq. 8: Question. What is $Yg$? Growth conversion efficiency? This could be included in line 189.

Eq. 10: Typo. "$T_soil$" → "$T_{soil}$"

Eq. 11. Question. If the water stress response of the vegetation in the model indeed has been deactivated as stated in line 170, no realistic simulation of the soil moisture status is possible. Does it then make sense to use soil moisture as a proxy for Rh?

Line 199. Typo. "…function and RSM1 The relative soil moisture…" → "…function and RSM1 the relative soil moisture…"

Line 203. Typo. "Biomass" → "biomass"

Eq. 15.: Question. I'm curious. SLA obviously is the key parameter for GLAI development in most models. I understand that in you study SLA (Cm) is constrained in the PROSPECT model inversions to the ranges given in Table 2. However, does the growth model, if running without satellite data to assimilate, consider changes of SLA over the course of the growing cycle?

Lines 224/225: Comment. I understand that decoupling the water and carbon cycle is convenient, because it alleviates the necessity to explain discrepancies between vegetation growth simulated according to the natural conditions and growth observed by the satellite. However, I see potential gaps evolving from that. E.g., your approach allows you to force the model into the reproduction of GLAI values that may be found in the satellite data, but cannot be explained by the meteorological budget (water, temperature) or the natural conditions (soil structure, nutrient

supply etc.), as it might for example be the case for irrigated areas. In this case, the mass and energy balance of your approach would not be maintained.

Line 231: Typo. *"…physiology,heterotroph activity…"* → *"…physiology, heterotrophic activity…"*

Line 232: Question. What is a *"Look out table"*?

Line 238: Grammar. *"…of the model parameters knowing that the observations x…"* → *"…of the model parameters knowing the observations x…"*

Line 257: Phrasing. *"…but not to the assimilation of satellite imaging."* → *"…but not to the assimilation of satellite images."* or *"but not to the assimilation of satellite imagery."*

Line 263: Typo. *"(e.N >> n)"* → *"(e >> n)"*

Line 269: Typo. *"i.e"* → *"i.e."*

Line 274: Grammar. *"…this expression leads to the manipulations of…"* → *"…this expression leads to manipulations of…"*

Line 276: Grammar. *"…vanilla matrix product."* → *"…vanilla matrix products."*

Line 278: Format. *"…re-scaled by their maximum. Eq.(23)."* → *"…re-scaled by their maximum (Eq. 23)."*

Line 299: Typo. *"…Equations (22,23 and 24)…"* → *"…Equations (22, 23 and 24)…"*

Line 303: Typo. *"ug m$^{-2}$"* → *"μg m$^{-2}$"*

Line 307: Comment. Please always include physical units. *"…that is considered constant at 0.02"* → *"…that is considered constant at 0.02 g cm$^{-2}$"*

Line 313: Typo. *"…Equations (22,23 and 24)…"* → *"…Equations (22, 23 and 24)…"*

Eq. 27: Format. I think it's nice that you are using overleaf as it provides a good platform for manuscript editing by many authors. However, the reference to Eq. 27 in line 321 seems to be messed up…

Table 2: Grammar. *"…is not allocated to the Leafs"* → *"…is not allocated to the leaves"*

**3. Application**

Line 325: Phrasing. I think that simply referring to *"the chain"* might not sound straightforward to many readers. I'd suggest referring to *"the model chain"* or *"the assimilation chain"* instead.

Line 325. Grammar. *"…is applied over a for winter wheat in years 2017, 2018, and 2019"* → *"…is applied for winter wheat in the years 2017, 2018, and 2019"*

Line 327. Phrasing. *"Several assimilation experiments were conducted to answer the specific objectives of the paper, they are summarised in Table 3. They alternate the use of…"* → *"Several assimilation experiments were conducted to answer the specific objectives of the paper. They are summarized in Table 3 and alternate the use of…"*

Table 3: Grammar. The *Objectives* are not complete sentences. The full stop-symbols therefore should be removed.

Figure 2: Question. What is the data source of the DEM in the background?

Figure 2 caption: Typo. *"…tile 31TCJ limts…"* → *"…tile 31TCJ limits…"*

Figure 3 caption: Phrasing. *"The bars plots represent the percentage…"* → *"The length of the bars represents the percentage…"*

Line 335. Style. Please decide to either use spaces between numbers and their respective physical units or not and apply uniformly throughout the manuscript (also see line 359).

Line 338. Interpunctuation. "…while year 2019, had a mild winter…" → "…while year 2019 had a mild winter…"

Line 340/341: Phrasing. *"It is mainly occupied by agricultural fields that cover about 90 % of the area, among which a majority of seasonal crops."* → *"It is mainly occupied by agricultural fields that cover about 90 % of the area and are predominantly cultivated with seasonal crops."*

Line 356. Typo. *"…from the database of the Environmental Information System the laboratory and the Regional Spatial Observatory (RSO).his Information systhem centralizes…"* → *"…from the database of the Environmental Information System, from the laboratory and from the Regional Spatial Observatory (RSO). This Information system centralizes…"*

Line 375. Grammar. "…each sample corresponds to a one linear meter of the crop row." → "…each sample corresponds to one linear meter of the crop row."

Line 376. Question. I think the term *"relative humidity"* should be reserved for the meteorological variable. What was the reason for avoiding the commonly agreed term *"canopy water content"*?

Line 378. Question. It is unclear what you mean by *"Eight fields were also sampled using the ESU protocol in 2018"*? As you didn't mention how many fields were sampled in 2017 and 2019. As your analysis focusses on the growing seasons of 2016/2017 and 2018/19, including the 2018 fields is somewhat confusing. Please provide a clear overview of how many fields were sampled according to which protocol in which year. This ideally should correspond to the points displayed in Figure 6.

Section 3.2.2. Question. It is known that yield data from combine harvesters is well-suited for describing relative spatial heterogeneities of yields, but may suffer from large errors concerning the absolute yield values. Were the CH measurements corrected, e.g. by determining the absolute weight of the harvest of the fields on a scale and applying the bias?

**4. Validation**

Line 391: Question. I don't understand the reference to Equation 27, please explain.

Line 393: Phrasing. *"…fitting statistics computed over the growing season show a good fit (R² = 0.93) in with a lower fit for the growing season in 2019"* → *"…fitting statistics computed over the growing season show a good fit in 2017 (R² = 0.93) with a lower fit for the growing season in 2019"*

Line 395: Phrasing. *"The GLAI for year 2019 senescence period is under-fitted while the…"* → *"The GLAI for the senescence period of 2019 is under-fitted while the…"*

Line 397: Grammar. "…with a R² of 0.88, 0.91, and 0.62…" → "…with R² 0.88, 0.91, and 0.62…"

Line 397: Structure. A reference to Table 5, where all the error indicators are listed, is missing here.

Line 403: Typo. "(RMSE = 1.43-1.90 gCm⁻²,Pique et al. (2020b))" → "(RMSE = 1.43-1.90 gCm⁻², Pique et al. 2020b)"

Figure 5: Question. It appears that growth activity in terms of GPP was overestimated in the model compared to the observations in the months February to May 2019. The simulated GLAI development in March 2019, however, is underestimated compared to the observations. Could you please elaborate on that? Compared to 2017/18, the deviations between modelled and observed variables indeed are higher for 2018/19. Would you think that the neglection of water stress dynamics contributed to these deviations?

Figures 5 and 6: Typo. In the y-axis label, please either write (g * m⁻²) or better (g m⁻²), but avoid (g. m⁻²).

Lines 412/413: Grammar. *"The comparison shows a good fit when considering together all DAM measurements with a $R^2$ of 0.90, a RMSE of 250 $gm^2$ and a slight negative bias 52 $gm^{-2}$."* → *"The comparison shows a good fit when considering all DAM measurements together with $R^2$ 0.90, RMSE 250 $gm^{-2}$ and slight negative bias of 52 $gm^{-2}$."*

Line 417: Grammar. *"…better fit that year with an $R^2 = 0.94$…"* → *"…better fit that year with $R^2 = 0.94$…"*

Table 6: Comment. I think it is important to highlight somewhere that the statistics given in Table 6 and in the text for the FR-AUR-Fields correspond to the agreement of the temporal biomass development and not to the agreement of the spatial yield patterns within or between the fields.

Figure 7: Style. Please aim for uniform labels throughput the manuscript. The text and all other figures print the units as "g $m^{-2}$". In Figure 7 it is "g/m2" [sic].

Figure 7: Comment. I understand that you cannot show a comparison for simulated and observed growth for the ESU-fields 2018, as there are not flux towers installed at these fields. However, first reading about the detailed model results for 2017 and 2019 and then seeing a validation including lots of points for 2018, is somewhat confusing. I think a well-structured overview about all the samples that are used is missing. To me, it is not clear to which ground samples the different data pairs in the scatter plot actually correspond. Obviously, the model returned constant values of 2000 g $m^{-2}$ for 2019, while the in-situ data showed large variations. Are these data from different fields? Or are they from different ESUs in the same field? Or are they from different ESUs in the two combine harvester fields? Sorry, if I'm sounding confused here, but I think this must be made more clear.

Line 423: Style. The reference to Grisso et al. 2022 is not according to format. This also accounts for the respective entry in the list of references.

Line 424: Phrasing. Incomplete sentence? *"ACEO-S2L8-Pixel by multiplying the final DAM by the Harvest Index (HI)."*

Line 425: Grammar. *"These maps shows the comparison…"* → *""These maps show the comparison…"*

Line 431: Comment. The listing of the different performance measures is confusing. I suggest including all these numbers into Figure 8.

Figure 8: Comments/Questions. What do "plot3" and "plot6" mean? Why are the names of the fields as given in the caption not displayed here? Why are the respective harvest years not printed? Why are there no scale bar, no North-arrows and no coordinates? The units should be t $ha^{-1}$ and not t.$ha^{-1}$. The variable is *"Yield"* and not *"Yiled"*. In the right part of the figure, there are small black dots between the fields. What do they represent? The agreement of the spatial patterns is surprisingly poor, given that the assimilation of GLAI should above all enable the simulation of intra-field heterogeneities.

Figure 8 Caption: *"…for the 2017 and 2019."* → *"…for the 2017 and 2019 growing period."*

**5. Large Scale**

Line 444: Phrasing. *"…considering 5000 LUT size."* → *"…considering a LUT size of 5000."*

Line 445/446: Question. While the scene is 110 x 100 km, the number of pixels with wheat fields is much lower as it can be seen in Figure 9. For which number of pixels do the given computing performances apply?

Line 460: Typo. *"…the maps in Figure 9 are a presented in…"* → *"…the maps in Figure 9 are presented in…"*

Figure 9: Style. Coordinates are missing. The units should be g $m^{-2}$ and not g.$m^{-2}$. In the overview map, the extent indicators of the zoomed maps at the bottom are too tiny to be discernible. The zoom maps are missing a scale bar. The color bars seem to have an inadequate color stretch. The map

for NEP-Mean is scaled to show high positive values in blueish colors. However, only very few fields with a blue hue are visible in the map when zooming in to a maximum. Maybe the color stretch should be applied more aggressively to reveal the spatial patterns in the negative (red) value range. This would also benefit the zoom maps. The same applies to the NEP-std map, where I was not able to find a single red pixel, even when zooming in to the maximum.

Line 466. Typo. *"…agricultural practices (ex. early vs…"* → *"…agricultural practices (early vs…"*

Line 468. Phrasing. I'm not sure if *"presents"* is the right word here. Maybe better *"shows"* or *"reveals"*?

Figure 10. Style. A Legend explaining the colors is missing. The labels partly overlap (e.g. SENb vs. SLAb). The cropping of the decimals places of the labels seems arbitrary (Do three decimal places make sense in this case: 19471,605?).

Line 470. Phrasing. I'd suggest changing *"milder"* for *"less pronounced"*.

Figure 11: Style. The units should be m² m$^{-2}$ and not m².m$^{-2}$. A legend explaining the colors of the lines is missing. Interpretation is not very intuitive, if the readers have to look up all the color codes in the caption. The date labels in figure (d) and (e) are cropped. The numbers indicating the ranges of the parameters in the radar plots are too tiny and partly overlap. There seem to be errors in the name labels of the parameters with exception of figure (a).

Figure 11: Caption. *"…in red the SAFYE-CO2 simulations ), and"* → *"…in red the SAFYE-CO2 simulations), and"*

Figure 11: Caption. *"…where a cloud date is not filtered in (d)."* → *"…where a cloud date is not filtered."*

Figure 12: Style. The units should be g m$^{-2}$ and not g. m$^{-2}$

Figure 13: Style. The units should be g m$^{-2}$ and not g.m$^{-2}$

Line 515: Typo. *"…10 m resolution ans smoothed images…"* → *"…10 m resolution and smoothed images…"*

Line 516/517: Phrasing. *"Figure 14 (b) is produced from the input land cover maps and shows the density of winter wheat fields over the region where the two main winter wheat regions: they correspond to the hilly areas located South-East of Toulouse and to the Gers department (West of Toulouse)."* → *"Figure 14 (b) is produced from the input land cover maps. The density of winter wheat fields indicates the two main wheat cultivation regions in the hilly areas South-East of Toulouse and in the Gers department (West of Toulouse)."*

Line 523: Typo. *"paterns"* → „*patterns"*

Figure 14: Style. Coordinates, North arrows and scale bars are missing for all five maps. The legends should show the respective physical units.

Line 528/529: Phrasing. *"A higher number of winter wheat pixels at high altitude tend to emerge earlier than pixels at lower altitudes. This can be seen intuitively given the altitudinal temperature gradient, however this difference 530 may be caused by hill shading effects…"* → *"Winter wheat pixels at high altitudes tend to emerge earlier than pixels at lower altitudes according to the altitudinal temperature gradient. However, this difference may also be caused by hill shading effects…*

Line 532/533: Phrasing. *"This later observation can be explained…"* → *"The latter can be explained…"*

Line 536: Typo. The major axes are N, W, S, E. BTW, the correct technical term is *"aspect"* and not *"exposition"*.

Line 544: Phrasing: *"(at a lower degree days values)"* → *"(at a lower degree days threshold)"*

Figure 15: Style. The physical units should be provided for all parameters next to the color bars and diagram axes. The frames of the legends in Figure (c) are cropped at the top. *"Exposition"* should be replaced by *"aspect"* in the plots and in the figure caption.

**6. Discussion**

Line 558: Grammar. *"…because the uncertainty on the EO derived GLAI are accounted for…"* → *"…because the uncertainties in the EO derived GLAI are accounted for…"*

Line 598: Typo. *"…appropriate solutions(Figure10),(Wang et al., 2022)."* → *"…appropriate solutions (Figure10; Wang et al., 2022)."*

Line 601: Typo. *"Note that ixed prior parameters…"* → *"Note that fixed prior parameters…"*

Line 604: Structure. Please explain abbreviations when first mentioned in the text and only use the abbreviation afterwards (*"ML"*).

Line 616: Typo. *"term ,large scale assessment…"* → *"term ,large scale assessment…"*

Line 620: Grammar. *"…requires in addition to the new soil module parameters, input dataset on initial soil…"* → *"…requires in addition to the new soil module parameters, input datasets on initial soil…"*

Line 644: Typo. *"$Co_2$"* → *"$CO_2$"*

Line 658: Typo. *"forsoil"* → *"for soil"*

---

## Author Comment (AC1)

Discussion reply on Anonymous Referee #1 comments on submitted paper:

**AgriCarbon-EO: v1.0.1: Large Scale and High Resolution Simulation of Carbon Fluxes by Assimilation of Sentinel-2 and Landsat-8 Reflectances using a Bayesian approach**

Taeken Wijmer, Ahmad Al Bitar, Ludovic Arnaud, Rémy Fieuzal, and Eric Ceschia

**General Comments:**

**Comment:**
The five authors have submitted a rather long manuscript (approx. 13.500 words excluding references), in which they advertise an approach for improved carbon balance mapping in agricultural soils. Long-term storage of carbon in agriculturally used soils is a very hotly discussed topic in the frame of carbon farming, which again is part of the EU green deal. Finding new ways to quantify soil carbon fluxes in agricultural systems therefore is a very interesting and promising research topic!

**Answer:**
The referee mentions that the paper is long. We agree with him about reducing the size of the manuscript while maintaining the needed sections to answer the aim of the paper which is to provide the community with a methodology that enables realistic estimation of crop carbon fluxes and production at large scale (regional) and high resolution (intra-plots).

**Comment:**
If we take a closer look at the manuscript, however, it turns out that the proposed approach is more a method for producing yield maps for winter wheat from correlations with green LAI observations from Earth Observation time-series. As, by using a simple LUE-based growth model and neglecting water-stress effects, the interrelations of the carbon and water cycle are explicitly not considered in this study, so that mass and energy balance may not necessarily be closed and no direct link between atmospheric carbon dioxide concentration and carbon sequestration in agriculturally used soils is established, I have the feeling that what we as a modelling community can learn from the presented study for carbon-farming related questions unfortunately is limited.

**Answer:**
We don't agree with the statement above and we bring the following precisions:

- The main outputs of the approach are the carbon fluxes: Net Ecosystem Exchange (Equation 9), Gross Primary Production (Equation 1), heterotrophic respiration (Rh) (Equation 10), autotrophic respiration (Equation 5), Ecosystem respiration (Reco) as well as biomass (Equation 13), and yield. NEE, Reco, GPP are validated in figure, NEE is showcased in figure (9). We also propose to add the sum of NEE over the growing season to figures (15). Probably some confusions originate from the differences between the SAFY (Duchemin et al. 2008) and the SAFYE-CO2 (Pique et al., 2020a,b ) models that share a limited number of equations.

- The input remote sensing information of the modelling chain is top of canopy (TOC) reflectances and not green leaf area index (GLAI). GLAI is an intermediate biophysical variable that is first retrieved with its associated uncertainty via the PROSAIL model and then assimilated into the SAFYE-CO2 crop model. As described in section (2.1). This ensures a propagation of uncertainties across the modelling chain.

- SAFYE-CO2 doesn't rely on correlation with GLAI but on a parsimonious modelling approach (Pique et al. 2020b) unless the referee means here "relation" and not "correlation". The model formalisms that are presented have been widely used in the community. They are less detailed than other radiative transfer and

agronomic models but offer the advantage of limited amount of data inputs which enables large scale high resolution spatialisation.

- The impact of water stress is considered indirectly for the case of Gross primary production (GPP) through the impact of in situ water stress on the observed GLAI that is assimilated into SAFYE-CO2. On the other hand, the impact of water-stress on soil respiration and evapotranspiration is considered via the available water content in the soil. The question of irrigation remains to be answered, we provide some elements later in this comment.

Overall, the very rich comments and open questions raised by the referee are of interest for the modelling community. From our point of view while AgriCarbon-EO includes a parsimonious modelling approach it provides answers to questions related to the crop carbon fluxes at never applied levels of high spatial resolutions and regional scales with a crop model.

**Comment:**
The paper consists of at least five major parts/questions, whereby each of the topics potentially would provide substance for individual articles.

**Answer**
Concerning the paper structure: Again, the main question behind the aim of the paper is:
" Is it possible to Provide reliable estimates of CO2 fluxes (e.g. NEE) and other key carbon budget components (biomass, yield) at intra field resolution and large scale".

In order to answer this question we need to provide a adapted methodology (presented in section 2) as well as to provide answers to the following specific questions (section 4 and 5):
 - What is the accuracy and limits of the approach based on existing validation datasets (carbon fluxes, biomass and yield) ?
- What is the impact of using Sentinel-2 and Landsat8 data at intra-field compared to the more commonly applied spatial unit (field scale)?
- Are the aggregated regional scale estimates coherent ?

To answer the main question we firstly present Agricarbon EO in detail for a generic application (section 2) e.g. methodology for inversion, modelling tools and input datasets). We describe the choices of input data for the paper, and then we present the application over South-West France (section 3). In this case study, we provide a multiscale validation/verification that covers the different scales covered by the AgriCarbon-EO tool (see figure R1 / Figure 4 in the manuscript), we also provide in Table 3, the name of the different simulation and which sub-objective they answer.

We agree that we can better explain the sub objectives and the structure at the end of the introduction, by providing a more concise paper and enhanced transitions we will make it clearer for the referee and the reader.

[Figure]

Figure R1: Temporal and spatial coverage of the input dataset (S2,L8, SAFRAN, and RPG), the validation dataset (Eddy covariance, Biomass, and Combine harvest), the regional dataset (SOILGRIDS and DEM), and AgriCarbon-EO outputs (blue zone).

**Comment:**

First, the model and the assimilation approach are introduced and the system is applied to field scale simulations, which are validated against destructively measured biomass and yield data. Thereby, ESU samples and combine harvester data are used. In figure 6 it can be seen that the ESUs were sampled on different dates throughout the growing period of 2017, while in 2019 only one date was sampled. Unfortunately, this is not clearly described in the text (e.g. sampling dates are not explicitly mentioned/listed) and confuses the validation of the temporal and the spatial performance of the algorithm. For example, in Figure 7 it remains largely unclear if the good performance of the year 2018 can be traced to the fact that the algorithm well follows the temporal dynamics of biomass accumulation (as biomass is suspected to continuously increase over the growing season, R2 correlation will necessarily be high...), or if the good performance is due to a good spatial mapping of heterogeneities of yield. Especially for 2019, where only one in-situ sampling date was available (as I deduce from Figure 6), the correlation is extremely poor, as the algorithm returns constant values of 2000 g/m2, while the in-situ data show a wide range of values. The way that I read the validation figures, this leads to the conclusion that the spatial heterogeneities of yield cannot be reasonably mapped with the proposed approach.

**Answer:**

First, it is incorrect to state that we deal with biomass and yield only. Indeed, before the section relative to the spatial biomass validation, we provide a validation of carbon fluxes (NEE, GPP, Reco) over the ICOS FR-Aur site (Section 4.1, Figure 5) at different periods (Table 5).

Second, concerning the biomass validation (figure 6 and figure 7). We mentioned in Table 4 of the manuscript, that ESU are sampled on 1 to 4 dates without providing the exact dates. **Table R1** shows the date and the field location of each of the measurements (it will be also added to the supplementary materials).

Table R1 : List of biomass records used in the validation for figures 6 and 7 of the manuscript

| Data set | Lat | Lon | Name | Date |
|---|---|---|---|---|
| ESU_2018 | 1,2507 | 43,4999 | Plot 1 | 20180702 |
| ESU_2018 | 1,2507 | 43,4999 | Plot 1 | 20180528 |
| ESU_2018 | 1,2507 | 43,4999 | Plot 1 | 20180504 |
| ESU_2018 | 1,2507 | 43,4999 | Plot 1 | 20180416 |
| ESU_2018 | 1,2507 | 43,4999 | Plot 1 | 20180406 |
| ESU_2018 | 1,5095 | 43,5261 | Plot 2 | 20180703 |
| ESU_2018 | 1,5095 | 43,5261 | Plot 2 | 20180524 |
| ESU_2018 | 1,5095 | 43,5261 | Plot 2 | 20180502 |
| ESU_2018 | 1,5095 | 43,5261 | Plot 2 | 20180416 |
| ESU_2018 | 1,5095 | 43,5261 | Plot 2 | 20180403 |
| ESU_2018 | 1,5038 | 43,5259 | Plot 3 | 20180703 |
| ESU_2018 | 1,5038 | 43,5259 | Plot 3 | 20180524 |
| ESU_2018 | 1,5038 | 43,5259 | Plot 3 | 20180502 |
| ESU_2018 | 1,5038 | 43,5259 | Plot 3 | 20180403 |
| ESU_2018 | 1,2029 | 43,4663 | Plot 4 | 20180528 |
| ESU_2018 | 1,2029 | 43,4663 | Plot 4 | 20180502 |
| ESU_2018 | 1,2029 | 43,4663 | Plot 4 | 20180416 |
| ESU_2018 | 1,2029 | 43,4663 | Plot 4 | 20180406 |
| ESU_2018 | 1,2355 | 43,4916 | Plot 5 | 20180702 |
| ESU_2018 | 1,2355 | 43,4916 | Plot 5 | 20180502 |
| ESU_2018 | 1,2355 | 43,4916 | Plot 5 | 20180416 |
| ESU_2018 | 1,2355 | 43,4916 | Plot 5 | 20180406 |
| ESU_2018 | 1,2290 | 43,4870 | plot 6 | 20180702 |
| ESU_2018 | 1,2290 | 43,4870 | plot 6 | 20180528 |
| ESU_2018 | 1,2290 | 43,4870 | plot 6 | 20180502 |
| ESU_2018 | 1,2290 | 43,4870 | plot 6 | 20180406 |
| ESU_2018 | 1,1559 | 43,4919 | plot 7 | 20180528 |
| ESU_2018 | 1,1559 | 43,4919 | plot 7 | 20180502 |
| ESU_2018 | 1,1559 | 43,4919 | plot 8 | 20180702 |

| AUR_2019 | 1,1052 | 43,5497 | AUR | 20190701 |
|----------|--------|---------|-----|----------|
| AUR_2019 | 1,1053 | 43,5499 | AUR | 20190701 |
| AUR_2019 | 1,1066 | 43,5494 | AUR | 20190701 |
| AUR_2019 | 1,1065 | 43,5493 | AUR | 20190701 |
| AUR_2019 | 1,1068 | 43,5494 | AUR | 20190701 |
| AUR_2019 | 1,1065 | 43,5496 | AUR | 20190701 |
| AUR_2019 | 1,1062 | 43,5496 | AUR | 20190701 |
| AUR_2019 | 1,1057 | 43,5497 | AUR | 20190701 |
| AUR_2019 | 1,1060 | 43,5498 | AUR | 20190701 |
| AUR_2017 | 1,1069 | 43,5497 | AUR | 20170704 |
| AUR_2017 | 1,1058 | 43,5497 | AUR | 20170704 |
| AUR_2017 | 1,1065 | 43,5497 | AUR | 20170704 |
| AUR_2017 | 1,1054 | 43,5496 | AUR | 20170704 |
| AUR_2017 | 1,1070 | 43,5496 | AUR | 20170522 |
| AUR_2017 | 1,1065 | 43,5496 | AUR | 20170522 |
| AUR_2017 | 1,1067 | 43,5498 | AUR | 20170522 |
| AUR_2017 | 1,1059 | 43,5496 | AUR | 20170522 |
| AUR_2017 | 1,1063 | 43,5497 | AUR | 20170522 |
| AUR_2017 | 1,1054 | 43,5496 | AUR | 20170522 |
| AUR_2017 | 1,1057 | 43,5497 | AUR | 20170522 |
| AUR_2017 | 1,1052 | 43,5497 | AUR | 20170522 |
| AUR_2017 | 1,1054 | 43,5496 | AUR | 20170330 |
| AUR_2017 | 1,1057 | 43,5497 | AUR | 20170330 |
| AUR_2017 | 1,1070 | 43,5496 | AUR | 20170330 |

Here, we will detail information about each dataset, to bring very precise elements about the results.

**ESU 2018 dataset:** The data is sampled from 8 plots in 2018 (1 to 4 ESU measurements per plot ). We extracted the time series for the ESU-2018 samples locations. Figure R2 shows the agreement between the observed ESU 2018 data and simulated biomass from AgriCarbon-EO at different stages of growth. *Figure R2* shows that the estimates are in good agreement across all dates and over all plots.

[Figure]

*Figure R2: LAI and biomass timeseries for the 8 ESU sites. Simulated ensembles are in black with transparency proportional to their relative likelihood, field observations are in red, and satellite observations are in green.*

**FR-AUR 2017 and 2019 :** The sampled locations for FR-AUR (Auradé site) are shown in Figure R3. The plot as a whole can exhibit strong heterogeneities. However the biomass measurements are done on a homogeneous part of the plot, close to the ICOS Eddy covariance flux tower. In fact, the flux tower location was intentionally installed in a relatively homogeneous and flat area according to ICOS site protocol. This is confirmed by the relatively homogeneous GLAI value next to the tower and in the sampling area (see Figure R3). This is why we associate the variability in the measurements to measurement uncertainties (std =300-400 g/m2 ). Year 2019, was a highly

productive year, the model shows a very high estimate within the observation uncertainty (Figure 6) but also a flat estimate of biomass that the referee has commented on. We associate it to the saturation effect of optical images, but we still confirm that the measurement uncertainty was high considering the low heterogeneity of the sampled region.

[Figure]

**Figure R3:** Location of the FR-AUR biomass measurement on 320170330(A), 20170522(B), 20170704(C) and 20190701 (D). The mean GLAI maps show the LAI on the 16th of may for both 2017 and 2019 years. This date corresponds roughly to the maximum GLAI.

In order to better represent the results, Figure 7 in the manuscript will be modified to exhibit the differences between end of cycle measurements and measurements during growth.

**Comment:**
This is confirmed then by Figure 8 also for the intra-field scale, where very poor correlations between the spatial combine harvester measurements and the satellite-based yield product become evident. For a satellite-based approach that explicitly targets the monitoring of intra-field variability in the context of precision farming measures, as claimed in the introduction, this is rather poor.

**Answer:**
Regarding the Spatial correlation of the yield maps, we agree that the R² is low compared to other studies focussing on yield predictions such as (i.e Hao 2021). In Hao (2021) we can see that state of the art agronomic models such as Apsim expect an RMSE of 1/t/ha which is higher than the RMSE obtained in our study. Furthermore it is important to keep in mind that this type of model is crop specific and most of the cases in the meta-analysis benefit from site specific calibrations in opposition to Agricarbon EO that only relies on weather data and satellite acquisitions.

In this context of application the small $R^2$ can however be explained by the range of variation of wheat yield that is smaller at intra field scale than regional or worldwide scale. As an illustration of this effect we realised a simple synthetic experiment. In this analysis we took the yield maps produced by the combine harvester as ground truth, added a normal simulated measurement noise with sigma = 1t/ha. The combine-harvester maps are compared to the maps with simulated observation noise to compute $R^2$. This exercise was repeated 1000 times to be able to represent the expected distributions of $R^2$ given this measurement noise (**Fig R4**:). This figure we can see that agricarbon EO the obtained $R^2$ is not out of line with the expected R2 for applications of state of the art Agronomic model for plot 3.

A.  B.

[Figure]

**Fig R4**: Expected R2 given simulated measurement error with sigma = 1t/ha for plots 6 (A) 3 (B). This value corresponds to the RMSE of SAFYE_CO2 at field scale and the expected RMSE for Apsim. The vertical line represents the value returned by Agricarbon-EO.

Furthermore when considering the correspondence between high and low Yield anomalies (i.e. |anomaly|>0.5) we got a good correspondence. This means that we can reliably identify high and low productivity areas in the field when there is significant variability in the yield to begin with as can be seen in **figure R5**.

A.  B.

[Figure]

**Figure R5:** Correspondence between the sign of extreme anomalies inside the plot 6 (A) and plot 3 (B). PP represents true positives, NN true negatives, PN false positives and NP false negatives.

Finally, we wish to highlight that if we compare these simulations to standard field wise simulations (that explains no variability i.e.$R^2$=0) the explained spatial variance illustrated here is a net gain.

To clarify our position and the context in which the statistics are obtained we will amend the discussion with these elements and also strengthen the discussion about the uncertainties.

**Comment:**
Following the field-scale validation, the paper takes a sudden turn towards large-scale simulations and shows the application of the method for retrieving Net Ecosystem Production for a 110 x 110 km large scene for the growing season of 2017. Obviously, only the winter wheat pixels were investigated, although this is not clearly stated in the text.

**Answer:**
Large-scale is mentioned in the title of the paper. Section (5) content was announced in the abstract, at the end of introduction and in the application section (3). Yes, We use winter wheat as an application and transitioning at this

stage of the paper to other available crop parameters like maize, sunflower would have been more confusing. Winter wheat is a major crop in South-west France as mentioned in the study area section 3.1, therefore this specific focus makes sense. However, because AEO is not "Winter wheat specific", the focus is not mentioned in the title.

Comment :
In this section, Figure 11, although suffering from some stylistic errors, makes the assimilation procedure and potential pitfalls of the algorithm transparent, so that the readers get a clear picture of how the Bayesian approach works. However, as the maps shown in Figure 9 due to the color stretch do not allow for the detection of large-scale patterns and also because these patterns are not discussed in the text, the readers wonder why this had not already been explained as part of the intra-field scale validation and why the jump to the large-scale actually was required for the purpose of this paper.

**Answer:**
We thank the reviewer for his positif feedback, we would add that it is important to note also that these represent a small percentage of the simulation. ("singular anomalies in the simulations that don't stand out in the regional statistics (line 473)".

Concerning Figure 9, we present here an updated formatting of the figure (**Figure R6**). It will also be updated in the manuscript. Note that in order to reduce the stretch positive values are omitted in this figure. These values correspond to abnormally low wheat development as well as places where no wheat grew contrary to the information provided by the RPG. We changed the zoomed maps to the limits of the NT-Plot6 field (for 2017). The regional patterns are more visible, but still the figure shows intra-field, inter-fields and regional heterogeneities at the same time.
We suggest moving Figure 10 to the supplementary material, and Figure 11 will be improved stylistically.
As explained above, large scale applicability is the core of the paper. It is important to address this scale if we aim at demonstrating the capabilities of the modelling chain. It suggests specific developments in order to address such scales. Removing the analysis at large scale, reduces the methodology to an application of the model to the field scale. As mentioned in section (2.4), demonstrating the capability of producing large scale and high spatial resolution estimates is one of the aims of Agri Carbon EO and this manuscript .

[Figure]

**Figure R6:** On the left, the NEP for winter wheat for the 2016-2017 cropping season. On the right, the corresponding uncertainties calculated with the AgriCarbon-EO processing by assimilating Sentinel-2 and Landsat-8 data (ACEO-S2L8-Pixel). The map also shows the position of selected Points Of Interest (POI) presented in Figure 11. The zoomed maps are on the NAT-Plot3.

**Comment:**

The next section opens two further side-questions about the impacts of spatial resolution and temporal sampling frequency (from my perspective these are questions three and four of the paper). The impact of spatial resolution is assessed by determining the bias between field averages and pixel-based values. I think that we no longer need to prove that intra-field variability indeed plays a major role. From my perspective it would have been more interesting to analyze, in a proper quantitative way, how much detail is lost when going from high-resolution of 10 m to a more modest resolution of 30 m, which widens the possibility for multi-mission observations. I think that in view of future Sentinel-Missions, which potentially will even provide higher resolutions of up to 5 m, this discussion is relevant. A similar drawback from my point of view is that the analysis of observation frequency impact on the DAM simulations is limited to the absolute number of satellite scenes and does not account for the differences in spatial resolution between Landsat and Sentinel-2.

**Answer:**

We explain hereby those choices:

- Concerning intra_field heterogeneities, in the discussion we mention the following: *"Intra-field heterogeneity is a well established issue in agricultural applications (Weiss et al., 2020; Blackmore et al., 2003; Grieve et al., 2019; Nowak, 2021), but it has not been thoroughly treated in terms of CO2 fluxes and uncertainties estimates. "* lines 566-567. So we address this point because if we are advocating for large scale intra-field estimation of carbon fluxes with its corresponding uncertainties, we should quantify the impact. Otherwise it is difficult to justify for data intensive methodologies that require higher computational needs.

- Concerning the choice of spatial resolution for the comparisons : From an agronomic modelling perspective which is central to this paper, the field scale and intra-field scale are well established concepts (Pasquel et al., 2022). Again, as we are advocating for intra-field scale we compare the highest resolution in our system (10m) to the field scale. It is important to bring the field scale to the picture. The only motivation to use 30m data is that it is the resolution of Landsat8 extracted bands, but actually the temporal frequency is 16 days which is not enough to constrain the crop model and anyway the Sentinel-2 is available at 10m. Still, we make a comparison in the paper between Sentinel-2 and Landsat8, when we compare the Sentinel-2 data against Sentinel-2 and Landsat8 combined which is most logical because Sentinel-2 is an operational mission with data available globally at 10m - 20m resolutions (figure 12 C).

- Concerning the context of Sentinel-2 Next Generation: Sentinel-2 NG at 5m (Löscher et al. 2020) is in study for an expected launch in 2034. To provide concrete elements in the discussion, we should consider existing similar or higher resolution data from drones, Planet Scope (Aragon et al., 2021), VENμS VM5 (Dick et al. 2022) (on daily revisit cycle). Such data is not operational or freely available globally. Also this will also have specific complexities (revisit, resolution, quality, spectral sampling) and is clearly out of scope of this paper's objectives, but we agree on the potential interest in the future.

- Finally on the DAM comparison in figure 13, the objective here is to show that increasing the absolute number of images reduces the uncertainties in the estimates but doesn't impact strongly the mean values which is important. Producing this figure for Landsat8 data only would not be feasible because the frequency of the Landsat8 data alone is not high enough to constrain SAFYE_CO2. As mentioned above the impact of adding Landsat8 specifically is shown in Figure 12 C.

**Comment:**

Also, the analysis does not consider the impact of observations that happen to cover specific growth stages. Satellite observations at specific growth stages might benefit the retrieval accuracy of certain parameters. By simply correlating the satellite images to the in-situ-sampled yield maps, it can be found that the pronunciation of intra-field patterns should high during BBCH 70-89 and reduced during the bolting phase. The occurrence of cloud cover during these phases may be highly specific for the region and may impact the application of the approach differently in different parts of the world.

**Answer:**

We agree with the Referee on this point. However doing such an analysis is not that straightforward for several reasons:

- First, in its current version our model does not estimate the timing of specific phenological phases as some classical agronomic models (e.g. STICS) apart from emergence, beginning and end of senescence. It makes difficult for instance to define when BBCH 70-89 will occur precisely on a given plot/pixel.
- Second, over our area of study, winter wheat varieties exist that have different sensitivities to degree days and we have no information on which winter wheat variety at 100x100 km. Also ,considering a fixed temporal window over the area of study in the attempt to catch accurately a precise phenological status is not an option as winter wheat pixels over our area of study may be characterised on a given day by rather different phenological status given the variability in sowing dates, climatic gradients, the differences in altitude and slope and aspects (even at intra field level). This is illustrated by the bivariate distribution of the phenological parameters in Figure 10 and by the spatial variability in the "*emerg*" and "*Sena*" parameters in Figure 14 (d) and (e).
- Third, the cloud coverage is not statistically stationary across the phenological stages over our area of study, therefore considering a fixed temporal window would bias the analysis.

Considering the above elements all together, it seems out of reach for this paper to do a meaningful analysis of the effect of the temporality of the gaps in EO observations on the performance of the method. Yet, we consider that a dedicated analysis on this topic would be very valuable.

**Comment:**

The fifth part of the paper then widens the scale even further up to the regional level by filtering the spatial results to correlation lengths of 2,5 km, thus entering a spatial scale beyond individual fields. The found large-scale patterns are explained by the soil characteristics in terms of water holding capacity and by the terrain situation (elevation, slope, aspect). While this undoubtedly explains the found spatial patterns, it remains unclear what the contribution of this section to the overarching subject of the study actually is.

Answer:

At this point in the manuscript we have provided performance metrics for several cases at pixel and plot scale applications for different variables. However this verification is operated on fractions of a percent of the total domain. It is thus possible to have doubt on the representativity of the simulations in other parts of the region with different pedoclimatic and growth conditions. In the absence of more validation data we deem important to verify if the simulations comply with the hypothesis about the effect of the pedoclimatic conditions in the region on crop growth. To perform this coherency analysis we compared the aspect, slope and soil texture to the phenological and growth outputs of Agricarbon-EO. However as the referee mentioned for Figure 9 regional tendencies can not be depicted clearly from the map. He is right on this and it is mainly because the map contains a mix of different variation scales. Those scales are the intra-field (soil texture, fertility depth, waterflow, slope, aspect), inter-field (soil texture, fertility, impact of choice of variety, farming practices, slope, aspect ), and landscape (impact of, altitude, local weather conditions).These facts motivated the application of a gaussian smoothing at 2.5 km to retrieve landscape scale trends. By subtracting this landscape trend from the raw signal we can retrieve the anomaly of the landscape signal containing the information about plots and intraplot variability. After unmixing these scales of variation we can compare the phenological and growth to the environmental variables. As the reviewer noticed "The found large-scale patterns are explained by the soil characteristics in terms of water holding capacity and by the terrain situation" as expected if we respect our hypothesis regarding the effects of the environment on crop growth. This allows to improve credibility to the spatial variabilities at different scales that are retrieved by Agricarbon-EO from weather and EO data alone.

Concerning Figure 14: We suggest moving it to supplementary material as it is just an intermediate result that shows the smoothed maps and confuses the message of the section.

Concerning Figure 15: The emergence (*emerg*) and sum of temperature at senescence (*sena*) are in this regard interesting because they answer part of the comment raised by the referee about the analysis with regards to the phenological stages. Dry above ground biomass is also important as it determines the quantities of exported biomass and thus biomass incorporated in the soil.

**Comment**:

The last part of the paper is dedicated to the discussion, which, in comparison to the size of the rest of the paper, is rather short (three pages). Here, the authors - among other recommendations-suggest the use of SAR-data with their

approach. As the proposed retrieval scheme which is based on the PROSAIL model does not apply to microwave data, it remains unclear how SAR data could successfully be integrated into the system.

**Answer:**

Naturally and considering the team's previous experiences with SAR data (Tomer et al. 2015, Tomer et al. 2016, Zribi et al. 2019, Fieuzal et al. 2017, Valero et al. 2021, Velozo et al. 2017, Baup et al. 2019), we consider assimilating SAR but we do not intend to assimilate SAR data into PROSAIL or to adapt PROSAIL for microwave data. As answered above in a previous question raised by the referee, the assimilation scheme which is based on a Bayesian approach can integrate additional information, but we would need to replace the PROSAIL model by an observation operator for SAR using a different model (Water cloud model WCM or other approaches). Two references are mentioned in the text (line 566), Veloso et al. ( 2017) who showed relation between SAR data GLAI and biomass, and Fieuzal et al. (2017) who derived GLAI from SAR and assimilated it into the SAFY model.

**Comment:**

In the second part and again in the fourth part of the discussion, the authors try to link their study, which is on winter wheat yield n South-West France, to soil carbon processes in general. As the connection is vague and indirect, because the relevant soil processes were neglected here, these attempts come across as rather endeavored. The further discussion treats well-known basic facts about remote sensing, e.g. the respective tradeoffs and advantages of physical modelling and machine learning.

**Answer:**

Again as explained above we re-emphasis: Our paper is not limited to winter wheat yield in South-West France. Actually only one figure over 15 concerns the yield ! We think that if we presented in the paper the methodology only without an application, it would have been impossible to show the advantages, drawbacks, and limitations of the approach. So alternatively producing simulations over a given area of interest of 100x100km shouldn't be an argument to reduce the paper to just this application and moreover reduce the application to one single variable.

**Comment:**

In general, I think that the Bayesian approach with the associated uncertainties makes a lot of sense and I would have been very curious about the explanations why the system performed so poorly with respect to spatial patterns. The discussion, however, only traces this major drawback to uncertainties in the in-situ data (e.g. in the combine harvester measurements), which I do not find very convincing.

**Answer:**

We agree that the discussion on the spatial patterns can be strengthened. We provided many elements in this comments that show that the results are not actually poor while we are aware of the limitations, including for a paragraph on the challenge to estimate very high aboveground biomass values (see previous answer relative to the 2019 biomass data) and more elements relative to the uncertainty on the yield in-situ data as mentioned. We didn't trace the results to the uncertainties in in-situ data only, here are excerpts from the text mentioning the model limitations:

- Line 429 : *"a clear saturation effect is observed in the simulations for the NAT-plt6 field"*
- Line 435 : *"The Low correlation as well as the difficulties in reproducing the range of yield observed variations of yield values may be caused by the simple representation of grain biomass allocation through the use of a HI which does not take into account potential variation of harvest index due to nutrient availability or crop cycle duration (Dai et al., 2016). "*

We also brought many elements in the previous responses that help clarify these points.

**Comment:**

A short conclusions and outlook section that mainly summarizes the main findings again closes the manuscript. As the authors state correctly at the end of the paper, the approach could potentially be used as "a coherent and multi-criteria full crop cycle agronomic diagnosis tool for production, carbon, phenology and water use". It is a pity that the presented paper does not advertise this potential of integrated remote sensing supported modeling approaches, but focuses on only one variable (aboveground biomass alias yield) and from my point of view misses to ask the relevant questions for time-seriesfrom EO data (e.g. What is the additional value of 10 m resolution compared to 30m? What

can be expected from future 5 m resolution data? What is the additional value of the spectral bands of Sentinel-2 compared to Landsat? What will be the impact of the new SWIR bands in the Sentinel-2 next generation for such integrated approaches? What does the interpretation of top of canopy spectral signals actually reveal about processes happening in the soil? etc.)

**Answer:**
First, we are really sorry to have to repeat the answer to the comment that we adresse "one variable" but we find it important to recall: we strongly disagree with the statement that the paper only focuses on yield and on the reasons why we limited our analysis to the carbon budget components (please see answers above).

Concerning the unraised questions that the referee would have liked the authors to ask:
- What is the additional value of 10 m resolution compared to 30m? We provided the reason why the 10 m to 30 m resolution analysis is not the most relevant (see answers above). We summarise this here again. 1) the only reason to go for specifically an arbitrary resolution of 30m is that the Landsat data is at 30m but this data is at 16 days revisit and is not sufficient to properly drive the SAFYE-CO2 parsimonious model. 2) The vast majority of crop modelling application at regional scale uses the field scale resolution, so we computed the impact of the use of field scale against 10m resolution 3) we also provided in the analysis the comparison to using S2 or S2+L8, as it has no sense to make an analysis by removing the S2 data which is freely available at 10m resolution globally.

- What can be expected from future 5 m resolution data? This was also answered above. So in summary it is of interest to answer this question, but clearly out of scope of this paper. It can be partially investigated before the launch of Sentinel-2 Next Generation (2034) by using Planet Scope, Venus VM5. We have a research program on that, we hope studies from other groups could answer this question with AgriCarbon-EO.

-What is the additional value of the spectral bands of Sentinel-2 compared to Landsat? We didn't make specific analysis by removing spectral bands for the PROSAIL retrievals, this has been already investigated in the literature (Dong et al. 2023).

- What will be the impact of the new SWIR bands in the Sentinel-2 next generation for such integrated approaches? It is an interesting question but again was the referee expecting us to add a section on the impact of potentially new SWIR bands in Sentinel-2 next generation expected to be launched not before 2034 ? Clearly this would require a specific analysis. One way to proceed would be to use a physically based RT model like DART model (Gastellu-Etchegorry et al. 2017) to generate synthetic scenes at different stages of development of the crops and to assimilate it into AgriCarbon-EO while updating for PROSAIL-Pro while coding PROSAIL-Pro in python. Referee can understand that this is out of scope of this paper.

- What does the interpretation of top of canopy spectral signals actually reveal about processes happening in the soil?  Top of canopy/soil spectral signal can give access to different variables with varying levels of uncertainty. In a general case, optical remote sensing can provide information about the structure and chemical composition of the plant as well as the composition and moisture of the soil. The processes inside the soil are thus not observed directly but the variables mentioned before condition de biomass that is returned to the soil, the top soil composition and moisture that are all relevant elements that are needed to model biological activity in the soil. The assimilation of this information can thus help to constrain soil processes indirectly.

- We didn't discuss this point indeed. The model produces above ground biomass but also below ground biomass, and soil respiration constrained by the soil moisture, so the assimilation of optical remote sensing of TOC reflectances is enabling better estimation of the soil processes.

 In summary we find it quite positive that the current paper opens so many questions. Clearly launching research programs in the community to answer them is of interest. We hope that AgriCarbon-EO could contribute to answering them.

**Comment:**
Overall, I think the fact that field scale temporal patterns, field scale spatial patterns, large-scale and regional scale modelling together with questions about spatial and temporal sampling density are all mixed in the paper, blurs the

structure of the presentation and makes the manuscript a rather demanding and rather exhausting read. I would highly recommend focusing on fewer aspects, e.g. maintaining sections 1-4 plus Figure 11 from section 5 and removing the rest. The link to the CarbonFarming buzzword seems artificial and should be removed or mitigated.

**Answer:**

Clarifications and argumentations about spatial and temporal scales and the need for the regional study have been provided in previous answers. We agree though that streamlining the paper by synthesising, and recontextualizing some details and removing some elements is needed to enhance the readability. We provide more details on this at the end of the general comments section.

Concerning the last comment on Carbon Farming, to make the link with carbon farming more explicitly, we propose to add at the end of line 72 the following sentence "One of the main motivation for developing AgriCarbon-EO is to answer the growing demand for a MRV tools allowing the production of high resolution maps of carbon budget components estimates and their uncertainties for different context of applications (e.g. carbon farming projects for the voluntary carbon market, Common Agricultural Policy, National Determined Contributions) compliant with the frameworks proposed by Smith et al. (2020) and Paustian et al. (2019). Those are relying among other things on the combined use of process based models, remote sensing data and a range of in-situ data (e.g. flux towers) for validation as in our approach." As a matter of fact, one of the main purpose for developing AgriCarbon-EO is to answer the demand from the scientific community working on soil carbon sequestration in Agriculture that span from several initiative like the CIRCASA project recommendations (https://www.circasa-project.eu/). The objective was thus to contribute in the development of tools for Monitoring Reporting and Verification (MRV) of soil carbon stock changes that would meet a set of criteria. Among other things it involves the use of process based models, assimilation of high spatial resolution remote sensing data, flux tower sites for validation of the models…to estimate the carbon budget components with associated uncertainties. This demand and context have also given rise to scientific projects that financed the development of AgriCarbon-EO mentioned in the acknowledgment (H2020 NIVA project, "Naturellement Popcorn", the Bag'ages (Agence de l'eau Adour Garonne) projects, the Horizon Europe ClieNfarm (n° 101036822) and ORCaSa (n° 101059863) projects, all of which are addressing carbon farming. Of course, to finalise the tools meeting all the criteria and comply with the frameworks proposed by Paustian et al. (2019) and Smith et al. (2020) some developments are still needed in AgriCarbon-EO (e.g. to validate the coupling of SAFYE-$CO_2$ with several soil models) and some validations of the whole approach against in-situ data of soil organic carbon stock changes are foreseen, as mentioned in the last section of the discussion (section 6.4). More generally, our approach is meant to provide a solution for MRV of soil organic carbon stock changes following carbon farming practices compliant with the framework of Smith et al. (2020 ) and applicable to different contexts of MRV (voluntary carbon market, agrifood sector's insetting, national inventories, carbon indicators for the common agricultural policy as during the NIVA project…).

**Comment:**
Also, the large amount of typing, language and format errors is quite surprising, given the autocorrection capabilities of state-of-the-art text processing software (see specific comments below).

**Answer:**
We sincerely thank the referee for his extensive editing work ! We will approve all the suggested modifications for the upcoming review process. We will also have the manuscript verified by an editing service.

**Comment:**
Before the potential further processing of the manuscript I recommend:
(i) to resolve the very large number of formal and spelling errors that prevent the readers from focusing on the content.

**Answer:**
All the formal and spelling errors listed by the referee will be corrected and the paper will be processed by an editing service.

(ii) to decide, if indeed all the scales, the intra-field scale, the large scale and the regional scale, should be treated in a single paper.

**Answer:**

We commented on this point above, explaining the structure and the modifications we suggest to streamline the reading of the paper. Clearly, several modifications should be made to enhance the readability. We suggest to put the sub objectives at the end of the introduction, to move SAFYE-CO2 equations to supplementary materials; to reduce section 2.4.1 on LUT generation ; to enhance the objective for each results section, to enhance figure 6, 7 and 9 ; to remove figure 10 and figure 14 in the results; to enrich the discussion section based on the exchanges in this document. We would maintain the multiscale validation as augmented in our previous answers.

(iii) to rethink the connection between the introduction that focuses on carbon farming and the actual content of the paper, which is more on yield modeling than on carbon sequestration.

**Answer:**
We reiterate the objective of the paper is to present the carbon fluxes and biomass (yield is presented in one figure of the 15 figures). As mentioned above we will better explain how AgriCarbon-EO fits in a soil carbon sequestration approach.

(iv) to think about, comparing different pixel resolutions would be more relevant than comparing field scale to pixel scale resolution.

**Answer:**
We invite the referee to consider the elements in our previous replies. To summarise, common applications of crop models at regional scale are done over field scale, we are advocating for regional scale intra-field resolution, so considering the community the paper addresses it is most important to compare the field to intra-field scale in the context of carbon fluxes and biomass.

(v) to analyze the impact of observations in specific phenological stages instead of taking only the absolute number of observations into account.

**Answer:**
We agree that from an agronomic point of view this would be interesting, but to make such a proper analysis is really not straightforward as explained above. We consider that a dedicated study is needed on this.more areas should be considered, because cloud coverage is not statistically stationary across the phenological stages and would bias the analysis.

(vi) to rewrite the discussion so that not only very general aspects of remote sensing are discussed, but the approach is referenced against other studies/approaches in the same field and especially the poor performance with respect to spatial patterns at the intra-field scale is adequately explained.

**Answer:**
Many elements in this discussion will enrich the discussion.

(vii) to rephrase the rather general title so that the limitation of the study to certain crops (winter wheat), certain variables (biomass, yield) and the region (South-West of France) is reflected in the headline.

**Answer:**
As this is the first paper that presents the methodology behind AgriCarbon-EO. We proposed to replace the current title by :
"AgriCarbon-EO: v1.0.1: Large Scale and High Resolution Simulation of Crop Carbon Fluxes and production by Assimilation of Sentinel-2 and Landsat-8 Reflectances using a Bayesian approach".
As explained above we do not wish to focus on winter wheat in the title as the methodology can apply to different crops and locations, winter wheat was chosen as an example for the application of the method.

**Specific Comments**

**Note:**

All typo, grammatical and rephrasing comments will be answered (validated) in the discussion stage. They were not included in this clarification reply.

**Abstract**

**Comment:**
Line 2: Phrasing. I don't think that in-situ sampling is "prohibitive". It surely is extremely labor intensive and thus not feasible. But the main drawback in my opinion is that it will never be spatially continuous.

**Answer:**
We agree on the message to relay to the reader. We will replace the word "prohibitive" as it has a legal sense to it that is not intended here. We mention that making soil samples every 3-5 years at intra-field resolutions and at national scale is unrealizable.

**1 Introduction**

**Comment:**
Line 59: Comment. "...information about development dynamics..." I think it would be important to highlight here that GLAI incorporates both, information on environmental growth conditions as well as information on human (management) behavior.

**Answer:**
We agree on this, we will clarify this in section 2.3.2. For instance, by replacing the following sentence "The soil stress impact on the vegetation cycle scale is implicitly considered through the assimilation of GLAI." by "The effects on the vegetation cycle of environmental stress impacts (e.g. nutrient or soil water content availability) and management are implicitly considered through the assimilation of GLAI".

**Comment:**
Line 61: Phrasing. The term *"restituted to the soil"* is somewhat misleading in my opinion, because the main issue here is the long-term storage of atmospheric carbon in the topsoil, which to the largest part comes from geological reservoirs and not only from agricultural soils and thus is not strictly all given back to the soils. I'd thus phrase it a little more neutral and just speak of *"biomass and carbon storage"* in the soil.

**Answer:**
This is a misunderstanding. Here we are talking about the carbon contained in the biomass that is returned to the soil. The word "restituted" will be replaced by "returned" which is commonly used by the soil carbon scientific community.

**Comment:**
Line 79: Question. Again, as there are temporal (frequency of observation) as well as geometric questions (intra-field resolution) targeted in your study, what kind of resolution is focused here?

**Answer:**
Spatial resolution as mentioned above. Our study, target the production of outputs at daily timescale and 10m spatial resolution over large areas (100 x 100 km).

**2 Methods**
**Comment:**
Line 84: Question. Is a daily time-scale adequately suited to model crop growth and potential stressors?

**Answer:**
Yes for crop growth and carbon fluxes at crop cycle scale, provided that the model trajectory is corrected via assimilation of frequent observations, not surely for the instantaneous impact of stressors. Clearly the time-scale and spatial scale depends on the modelled phenomenons and processes. The vast majority of crop models have daily time steps (CropSyst (Stöckle, 2003), STICS (Brisson, et al. 2003), DNDC Gilhespy et al. 2014 , SAFY- Duchemin et

al. 2008, SAFYE-CO2 Pique et al. 2020a, Sunflo (Casadebaig et al. 2011). We consider that in the light of the announced objectives the daily time step is a good trade-off between precision and efficiency knowing that the objective of the paper is not to present a new crop model but the assimilation scheme and the overall performance of the approach to estimate carbon budget components for cropland.

Still, for more detailed modelling a higher time-scale would be needed (DSSAT runs at hourly time step, SCOPE also). If one wan't to aim at modelling for example the unsaturated flow and solute transport in the soil (Al Bitar, 2007), or turbulent flow above the canopy for energy balance, for specific stressors, a sub-second time step may be required not only to represent the process but also for numerical stability. In these cases a regional modelling at very high resolution would be extremely demanding, and an extensive amount of approximations need to be considered for the input parameters.

**Comment:**
Line 85: Question. Wouldn't the assimilation scheme not also work for microwave data?

**Answer:**
For passive microwave: for example if we are addressing soil moisture or vegetation optical depth (Al Bitar et al., 2017), a spatial downscaling would be needed considering the high spatial resolution gap with the model spatial resolution, also the representative depth should be considered (Lievens et al., 2016).

For active microwave: SAR data exist at 10m but it should be processed for speckle effect so resolution change should be considered. In summary specific processing should be done.

Nevertheless algorithmically, the Bayesian approach is a good framework for multi-sensor data assimilation and would work as long as an observation operator is provided to generate a variable that is modelled by SAFYE-CO2.

**Comment:**
Line 96: Question. Is analyzing each image independently making full use of time-series of satellite data?

**Answer:**
Yes, in the context of the defined objectives and trade-off between computational needs and precision. The other two options to consider spatio-temporal information would have been:

1) make an integrated retrieval of the PROSAIL (observation operator) and SAFYE-CO2 model in a spatio-temporal manner by assimilating the reflectances in the integrated system. For local application this would be possible but at large scale and 10 m resolutions it would require a large amount of computational resources which limits the scalability to 100 by 100 km scales.

2) We could get more information from the time series as there is temporal correlation inside of the LAI time series that could be used to constrain each image further. However, this correlation is variable in time and space depending on the different growth stages and thus difficult to characterise and implement.

**Comment:**
Line 119: Comment. I think it would be worth mentioning that parceled land use data is not available for many parts of the world. The requirement of parceled land use inputs limits the applicability of the AgriCarbon-EO approach to those areas where parcel information is available.

**Answer:**
Actually, AgriCarbon-EO can be applied if a pixel-based classification map is available without parcel's limits. The limiting factor would be a classification map which is much more easily accessible than parcel data. We can clarify this issue in the manuscript.

**Comment:**

Line 123: Question. I understand that a UTM map projection corresponds to the Sentinel-2 imagery that is used in the approach. I just wonder if staying with an equal-area projection would make sense here to facilitate the quantification of fluxes per area.

**Answer:**

A concept of abstract grid exists in AgriCarbon-EO. Actually, all data sets are projected to a common Discret Global Gridding (DGG) System. Users can define the DGG as the EASE Cylindrical grid (Brodzik et al., 2012) which is equal-area, but at 10m resolution, the scale of this exercice and considering the associated uncertainties we don't see the specific interest of using an equal-area projection .

**Comment:**

Section 2.2.3: Question. I am surprised that neither wind velocity nor atmospheric carbon dioxide concentration are required as meteorological input. For a model that is targeting carbon farming applications, I would have expected a direct link between the water and carbon cycle to be present in the algorithm.

**Answer:**

The effect $CO_2$ concentration is implicitly accounted for through the LUE calibration approach. Concerning wind speed, previous work on SAFY, SAFYE or SAFYE-CO2 showed that biomass, yield or $CO_2$ fluxes could be estimated with good accuracy without accounting for wind speed. Also, When the user activates water balance in SAFYE-CO2, the wind velocity is implicitly considered in the Potential evapotranspiration computation.

**Comment:**

Line 149: Question. How does 8 km resolution weather data correspond to the intra-field geometric detail that is targeted in this study?

**Answer:**

The 8 km weather data corresponds to a mean value for the zone. However, we agree that the climatic variables environmental stressors (temperature radiation and rain, wind, air humidity etc..) that constrain the plant development and soil processes are forced by the microclimate (especially in hilly landscapes such as in the application section). In our approach, this effect is compensated by the value of the variable parameters that adapt to local conditions through the assimilation of GLAI remote sensing datasets. The processing chain has been developed to take into account the state-of-the art global and open weather dataset (e.g. ERA5-land) which is an important criteria to provide future carbon farming MRV tools. Providing weather data with better resolution will surely have an impact on accuracy and computational performances. In the manuscript, we discussed this in Section 6.2 while providing the impact on performances (Equation 28) of future weather data.

**Comment:**

Line 159: Question. One is wondering why in a study about carbon storage instead of Prospect-5d not the most recent version PROSPECT-PRO is used, which specifically includes absorption coefficients for the carbon-based constituents of aboveground biomass.

**Answer:**

Prospect-Pro (Férét et al. 2021) is a recent evolution of the Prospect model. Férét et al. (2021) mentions the following: "Our results indicate the importance of narrow SWIR domains, which will remain to be important also at the canopy level. Current multispectral spaceborne data (e.g., Landsat 8/9 and Sentinel-2 images) do not comply with the narrowband SWIR spectral requirements that we identified, and further investigations are necessary to conclude on feasibility and limitations of its potential use for N mapping using PROSPECT-PRO." If this limitation remains applicable, we would potentially integrate Prospect-Pro when narrow SWIR is available (launch of Sentinel-2 NG in 2034).

**Comment:**

Eq.1: Comment. I think it's a pity that such a "simple" LUE-model is used to describe carbon fixation at the land surface. There are gas-exchange models available that create a direct link between carbon and water cycles. For a study in carbon farming, I would expect a more complex approach.

**Answer:**

The scope of this paper is to analyse carbon budget components only. Also, we chose a parsimonious approach because our objective is to simulate in a diagnostic mode only those components at large scale and high resolution benefiting from the assimilation of GLAI derived from remote sensing which implicitly accounts for some environmental stress including atmospheric $CO_2$ fertilisation effect. Many other agronomic models rely on LUE approaches (e.g. STICS). Using more complex photosynthesis modelling approaches (e.g. based on the Farquahr model) may be justified for analysing infra daily photosynthesis process or for forecasting (e.g. in future climate). In our case it would require more parameters, more calibration processes, more computation time and it might result in larger uncertainties in the key outputs. Therefore, we consider that given our objectives using a "simplified LUE approach" is justified.

**Comment:**

Line 170/171: Question. What was the reason for ignoring water stress effects in the simulation? I think that very interesting findings can be made for example when the modelled biomass according to the natural water budget does not correspond to the biomass accumulation observed from satellites. Ignoring the water stress also means ignoring uncertainties in the soil parameterization. How does your model explain the differences found between simulation and satellite observation, if soil processes are ignored? If the model does not even try to explain them and simply accepts the observation and carries on, what can we learn about natural processes from such a model?

**Answer:**

- Question. What was the reason for ignoring water stress effects in the simulation?

We recall that the impact of water stress on production of biomass is implicitly accounted for in the assimilation of frequent remote sensing observations which are a representation of the in situ plant developpement. Impact of water availability on soil respiration and percolation is actually taken into account.

- I think that very interesting findings can be made for example when the modelled biomass according to the natural water budget does not correspond to the biomass accumulation observed from satellites. Ignoring the water stress also means ignoring uncertainties in the soil parameterization. How does your model explain the differences found between simulation and satellite observation, if soil processes are ignored?

It is not clear what the referee means with biomass accumulation from satellites. We are assimilating GLAI which is not a direct proxy of biomass. Still, we find the comment on discrepancies of interest. Actually, in such cases to identify the discrepancies, a crop model would be run without assimilation of remote sensing GLAI and forced by weather inputs only. The discrepancies between modelled GLAI and observed GLAI (or modelled soil moisture and remote sensing based soil moisture) can then be interpreted in an inversion scheme as the impact of unrepresented processes ie. soil properties, irrigation, water flow or pathogens depending on the model. These effects can all be present at the same time which makes them difficult to discriminate. These are all alternative research objectives in which water stress should be taken into account explicitly.

- If the model does not even try to explain them and simply accepts the observation and carries on, what can we learn about natural processes from such a model?

Again, concerning soil processes, they are not ignored as mentioned above. Also the model doesn't just accept the observations and "carries on" as we are not doing a direct insertion. As we are using a Bayesian approach for the assimilation, the assimilation scheme takes the model uncertainties and the observation uncertainties into account, which are also the basis for obtaining the output variables uncertainties. The assimilation of GLAI in the agronomical model is to rely on the observations which has many benefits such implicitly accounting for not only stress but also for pest and pathogens effects on the plant development that cannot be reliably simulated by most agrometeorological models at large-scale, while simulating several processes (e.g. photosynthesis, plant respiration, biomass allocation…). Additionally the model provides useful information on the effects of climatic variable on the carbon budget components that allow for instance to analyse the effect of straw export or return on the annual C budget (Pique et al. 2020a), the effect of cover crops or spontaneous regrowth on the carbon budget components (Al bitar et al. 2021), the water use efficiency (Pique et al. 2020b). But again, As mentioned above the objective of the paper is not to present the SAFYE-CO2 model, but to present AgriCarbon-EO and to evaluate its potential for estimating carbon budget components at high resolution and large scale, and we argue that the community can learn from AgriCarbon-EO on questions linked to the uncertainty of the estimations of the carbon budget components.

**Comment:**

Line 175: Question. Is a multiplicative factor well suited to describe the temporal dynamics of senescence? A multiplicative factor will result in a rapid decrease of "greenness" at the onset of senescence, while the increments of

the greenness-decrease will become smaller as senescence progresses. From my experience, S-shaped sigmoid functions better correspond to the dynamics of senescence that are observed in the field. The model results shown in Fig. 5 also do not look like as if a constant senescence factor was applied. Could you please explain?

**Answer :**

The senescence is modelled in the study, as a function of the sum of temperature (SMT) so it can adapt to changes in weather conditions and changes in GLAI as implemented in SAFY (Duchemin et al. 2008), SAFYE (Battude et al. 2017), and SAFYE-CO2 (Pique et al. 2020b) models. Actually, the function used here is a discrete form of a sïgmoid function in SMT. It depends on sena: the sum of temperature at which senescence begins, and senb the parameters that control the slope of LAI decrease.

**Comment:**

Eqs. 6 and 7: Comment. From my perspective, the maintenance respiration should be connected to the tissue that already has been accumulated which requires "maintenance energy". I don't see a link to the accumulated biomass here.

**Answer:**

The maintenance respiration is linked to the NPP that integrates all the carbon that has been accumulated by the plant. It is equivalent to (DAM+DBM)/C content.

**Comment:**

Eq. 8: Question. What is Yg? Growth conversion efficiency? This could be included in line 189.

**Answer:**

Yes it is growth conversion efficiency. We will clarify this in the text.

**Comment:**

Eq. 11. Question. If the water stress response of the vegetation in the model indeed has been deactivated as stated in line 170, no realistic simulation of the soil moisture status is possible. Does it then make sense to use soil moisture as a proxy for Rh?

**Answer:**

The water stress corresponds only to the impact of water availability on the production of the vegetation (GPP and DAM), which is continuously updated by the assimilation of the remote sensing data. The available water on the other hand impacts the transpiration and soil respiration so there is a realistic modelling of the water budget. Actually, Pique et al. (2020b) showed for contrasted climatic years that water stress effect was already accounted for though GLAI assimilation in SAFYE-CO2 when estimating GPP and DAM for winter wheat (no improvement of GPP estimates when the stress function was activated) (again only for GPP).

**Comment:**

Eq. 15.: Question. I'm curious. SLA obviously is the key parameter for GLAI development in most models. I understand that in you study SLA (Cm) is constrained in the PROSPECT model inversions to the ranges given in Table 2. However, does the growth model, if running without satellite data to assimilate, consider changes of SLA over the course of the growing cycle?

**Answer:**

Yes, *SLA* can be configured to be either constant (Pique et al. 2020 a) or dynamic (Battude et al. 2017) over the course of the growing cycle. So yes theoretically, it is possible to consider changes of *SLA*. It is important to mention that SAFYE-CO2 has been developed specifically for spatialised simulations through GLAI assimilation. It has not been applied without remote sensing assimilation.

**Comment:**

Lines 224/225: Comment. I understand that decoupling the water and carbon cycle is convenient, because it alleviates the necessity to explain discrepancies between vegetation growth simulated according to the natural conditions and growth observed by the satellite. However, I see potential gaps evolving from that. E.g., your approach allows you to force the model into the reproduction of GLAI values that may be found in the satellite data, but cannot be explained by the meteorological budget (water, temperature) or the natural conditions (soil structure, nutrient supply etc.), as it

might for example be the case for irrigated areas. In this case, the mass and energy balance of your approach would not be maintained.

**Answer:**

As mentioned above the decoupling is only applied for the computation of the GPP by removing the impact of water stress on the photosynthesis. The water budget that is expressed via the percolation in the soil and the evapotranspiration is still dependent on the available soil moisture. For the natural conditions, it is also used like in the case of the soil respiration which depends on the soil moisture. So the water balance is maintained. For the energy balance we are only using the FOA56 method to compute the evapotranspiration based on the potential evapotranspiration provided by the weather dataset.

The case of irrigation doesn't concern this winter wheat application as winter wheat is not irrigated in our area of interest. On other crops such as corn the user will activate an automatic irrigation module that ensures a consistency between evaporation demand, and plant growth (Battude et al. 2017). Nevertheless, applying irrigation schemes in crop and land surface models is a hot subject and requires specific attention (Druel et al. 2022).

Referee has raised this point several times and we answered it. We find it is important to add this information to the manuscript to clarify it also for the readers.

**3. Application**

**Comment:**

Figure 2: Question. What is the data source of the DEM in the background?

**Answer:**

The map is ESRI World Topo Map

The map sources are: Esri, HERE, Garmin, Intermap, increment P Corp., GEBCO, USGS, FAO, NPS, NRCAN, GeoBase, IGN, Kadaster NL, Ordnance Survey, Esri Japan, METI, Esri China (Hong Kong), (c) OpenStreetMap contributors, and the GIS User Community.

**Question:**

Line 376. I think the term "relative humidity" should be reserved for the meteorological variable. What was the reason for avoiding the commonly agreed term "canopy water content"?

**Answer:**

We agree that the term "canopy water content" is in fact more accurate. The term was taken from the sampling protocol technical documents.

**Comment:**

Line 378. Question. It is unclear what you mean by "Eight fields were also sampled using the ESU protocol in 2018"? As you didn't mention how many fields were sampled in 2017 and 2019. As your analysis focuses on the growing seasons of 2016/2017 and 2018/19, including the 2018 fields is somewhat confusing. Please provide a clear overview of how many fields were sampled according to which protocol in which year. This ideally should correspond to the points displayed in Figure 6.

**Answer:**

This was thoroughly answered earlier. We also provided the locations and dates of the 53 measurements used. We also provided the time series of simulated vs observed DAM for the year 2018 showing the fitting. For 2017-2018 over field FR_AUR the crop was not winter wheat. Table 4. in the manuscript shows all the list of simulations and the way they are used.

**Comment:**
Section 3.2.2. Question. It is known that yield data from combine harvesters is well-suited for describing relative spatial heterogeneities of yields, but may suffer from large errors concerning the absolute yield values. Were the CH measurements corrected, e.g. by determining the absolute weight of the harvest of the fields on a scale and applying the bias?

**Answer:**
Thanks for pointing this out. No, this correction wasn't applied. It may explain some mismatch between the results and the in-situ data and may be of interest. We will mention it in the discussion.

**4. Validation**

**Comment:**
Line 391: Question. I don't understand the reference to Equation 27, please explain.

**Answer:**
We referenced equation 27 to indicate the method used to obtain the plot scale time series from the original 10 m pixel scale simulation.

**Comment:**
Figure 5: Question. It appears that growth activity in terms of GPP was overestimated in the model compared to the observations in the months February to May 2019. The simulated GLAI development in March 2019, however, is underestimated compared to the observations. Could you please elaborate on that? Compared to 2017/18, the deviations between modeled and observed variables indeed are higher for 2018/19. Would you think that the neglection of water stress dynamics contributed to these deviations?

**Answer:**
It is true that modelled GPP is higher than the measured GPP during the growing phase in 2019, and the GLAI in March seems to be retrieved correctly but not at the end of the cycle. The performances in 2018-2019 are less accurate than in 2016-2017. We attribute this discrepency to three factors. First The quality of flux data seems to be better in 2016-2017 than in 2008-2019, we illustrate this in **Figure R7**. In this figure we overlay 2 flag information from the flux tower data over Figure 5 from the manuscript. The grey color corresponds to date where more than 50% of the eddy measurements - that are initially provided at 30 min intervals - are gap filled. This reveals that the majority of the information given during the month of february is produced by the gap-filling procedure. It is however known that gap-filling is less performant for large gaps (Moffat et al., 2007). The second flag in red represents days where there is doubts on the partitioning of Net Ecosystem exchange in Green primary production and Ecosystem respiration as the respiration values are incoherent for the end of march in the absence of freezing. It is notable that this period also presents a high number of gap filled information over a shorter time span. Finally 2018-2019 was an exceptional cropping season for wheat in or region of interest in terms of production. This is due to climatic as well as management factors through a high Nitrogen fertilisation on this specific plot. It is a well known fact that nitrogen fertilisation can influence Chlorophyll concentrations (Hinzman et al., 1986). This can lead to slight overestimations of LAI. This is due to the similar effect of LAI and chlorophyll content increase have on the reflectances simulated by PROSAIL. This can be mentioned in the manuscript.

[Figure]

**Figure R7: Time series of CO2 fluxes. In blue the a posteriori distribution and the standard deviation. In red the GLAI derived from the satellite observations and the NEE, Reco and GPP at the FR-AUR site for two cropping years (2017 and 2019).**

**Comment:**
Table 6: I think it is important to highlight somewhere that the statistics given in Table 6 and in the text for the FR-AUR-Fields correspond to the agreement of the temporal biomass development and not to the agreement of the spatial yield patterns within or between the fields.

**Answer:**
It is of interest as pointed out to specify spatial and temporal variability and keep this figure to illustrate the representation of the temporal component.

**Comment:**
Figure 7: Comment. I understand that you cannot show a comparison for simulated and observed growth for the ESU-fields 2018, as there are no flux towers installed at these fields. However, first reading about the detailed model results for 2017 and 2019 and then seeing a validation including lots of points for 2018, is somewhat confusing. I think a well-structured overview about all the samples that are used is missing. To me, it is not clear to which ground samples the different data pairs in the scatter plot actually correspond. Obviously, the model returned constant values of 2000 g m-2 for 2019, while the in-situ data showed large variations. Are these data from different fields? Or are they from different ESUs in the same field? Or are they from different ESUs in the two combine harvester fields? Sorry, if I'm sounding confused here, but I think this must be made more clear.

**Answer:**
The questions around figure7 were answered above. And all the raised questions were answered. As the referee noticed (Figure R1) we did provide the observed and simulated growth for the ESU-2018 fields. They show a good agreement. What we don't have is flux towers on all these fields to show the carbon fluxes. As mentioned above figure 7 will be enhanced to show better these information.

**Comment:**
Figure 8: Comments/Questions. What do "plot3" and "plot6" mean? Why are the names of the fields as given in the caption not displayed here? Why are the respective harvest years not printed? Why are there no scale bar, no North-arrows and no coordinates? The units should be t ha-1 and not t.ha-1. The variable is "Yield" and not "Yiled". In the right part of the figure, there are small black dots between the fields. What do they represent? The agreement of the spatial patterns is surprisingly poor, given that the assimilation of GLAI should above all enable the simulation of intra-field heterogeneities.

**Answer:**

The figures will be enhanced but concerning the representation of intra-field heterogeneity we showed an analysis earlier in the answers that show that the outputs have a reasonable agreement regarding the range of variability seen in the field.

**5. Large Scale**

**Comment:**
Line 445/446: Question. While the scene is 110 x 100 km, the number of pixels with wheat fields is much lower as it can be seen in Figure 9. For which number of pixels do the given computing performances apply?

**Answer:**
In fact, the number of wheat pixels (20 M pixels line 329) which corresponds to about 20 000 000 /(11000*11000)*100=16.5% of the scene. To be clear we simulate all the 20 000 000 wheat pixels given by the RPG in the 110*110 km window.

**References :**

Al Bitar, Modélisation des écoulements en milieu poreux hétérogènes 2D/3D, avec couplages surface/souterrain et densitaires, Thése de Doctorat, SDUS2E, 2007.

Al Bitar, A., Mialon, A., Kerr, Y. H., Cabot, F., Richaume, P., Jacquette, E., ... & Wigneron, J. P. (2017). The global SMOS Level 3 daily soil moisture and brightness temperature maps. *Earth System Science Data*, *9*(1), 293-315.

Al Bitar, A., Wijmer, T., Arnaud, L., Fieuzal, R., Soussana, J. F., Gibrin, H., ... & Ceschia, E. (2022, July). Quantification of the impact of cover crops on Net Ecosystem Exchange using AgriCarbon-EOv0. 1. In *IGARSS 2022-2022 IEEE International Geoscience and Remote Sensing Symposium* (pp. 5781-5784). IEEE.

Aragon, B., Ziliani, M. G., Houborg, R., Franz, T. E., & McCabe, M. F. (2021). CubeSats deliver new insights into agricultural water use at daily and 3 m resolutions. Scientific reports, 11(1), 1-12.

Broberg, M. C., Högy, P., Feng, Z., & Pleijel, H. (2019). Effects of elevated CO2 on wheat yield: non-linear response and relation to site productivity. *Agronomy*, *9*(5), 243.

Brodzik, M. J., Billingsley, B., Haran, T., Raup, B., & Savoie, M. H. (2012). EASE-Grid 2.0: Incremental but significant improvements for Earth-gridded data sets. *ISPRS International Journal of Geo-Information*, *1*(1), 32-45.

Brisson, N., Gary, C., Justes, E., Roche, R., Mary, B., Ripoche, D., Zimmer, D., Sierra, J., Bertuzzi, P., Burger, P. and Bussière, F., 2003. An overview of the crop model STICS. *European Journal of agronomy*, *18*(3-4), pp.309-332.

Casadebaig, Pierre, et al. "SUNFLO, a model to simulate genotype-specific performance of the sunflower crop in contrasting environments." *Agricultural and forest meteorology* 151.2 (2011): 163-178.

Deininger, K., Monchuk, D., Nagarajan, H. K., and Singh, S. K.: Does Land Fragmentation Increase the Cost of Cultivation? Evidence from India, The Journal of Development Studies, 53, 82–98, https://doi.org/10.1080/00220388.2016.1166210, https://doi.org/10.108000220388.2016.1166210, publisher: Routledge _eprint: https://doi.org/10.1080/00220388.2016.1166210, 2017.

Dick, A., Dedieu, G., Hagolle, O., Raynaud, J. L., Pelou, S., Farges, M., & Peschoud, C. VENµS Mission Evolutions and Radiometric Performances During VM5 In-orbit Test Phase. Conference on Characterization and Radiometric Calibration for Remote Sensing (CALCON) (2022)..

Dong, T., Liu, J., Liu, J., He, L., Wang, R., Qian, B., ... & Shang, J. (2023). Assessing the consistency of crop leaf area index derived from seasonal Sentinel-2 and Landsat 8 imagery over Manitoba, Canada. *Agricultural and Forest Meteorology*, *332*, 109357

Druel, A., Munier, S., Mucia, A., Albergel, C., & Calvet, J. C. (2022). Implementation of a new crop phenology and irrigation scheme in the ISBA land surface model using SURFEX_v8. 1. *Geoscientific Model Development*, *15*(22), 8453-8471.

Duchemin, B., Maisongrande, P., Boulet, G., and Benhadj, I.: A simple algorithm for yield estimates: Evaluation for semi-arid irrigated winter wheat monitored with green leaf area index, Environmental Modelling & Software, 23, 876–892, https://doi.org/10.1016/j.envsoft.2007.10.003, https://www.sciencedirect.com/science/article/pii/S1364815207002010, 2008.

Féret, J. B., Berger, K., De Boissieu, F., & Malenovský, Z. (2021). PROSPECT-PRO for estimating content of nitrogen-containing leaf proteins and other carbon-based constituents. *Remote Sensing of Environment*, *252*, 112173.

Gastellu-Etchegorry, J. P., Lauret, N., Yin, T., Landier, L., Kallel, A., Malenovský, Z., ... & Mitraka, Z. (2017). DART: recent advances in remote sensing data modeling with atmosphere, polarization, and chlorophyll fluorescence. *IEEE Journal of Selected Topics in Applied Earth Observations and Remote Sensing*, *10*(6), 2640-2649.

Gilhespy, S. L., Anthony, S., Cardenas, L., Chadwick, D., del Prado, A., Li, C., ... & Yeluripati, J. B. (2014). First 20 years of DNDC (DeNitrification DeComposition): model evolution. *Ecological modelling*, *292*, 51-62.

Hao, S., Ryu, D., Western, A., Perry, E., Bogena, H., & Franssen, H. J. H. (2021). Performance of a wheat yield prediction model and factors influencing the performance: A review and meta-analysis. *Agricultural Systems*, *194*, 103278.

Hinzman, L. D., Bauer, M. E., & Daughtry, C. S. T. (1986). Effects of nitrogen fertilization on growth and reflectance characteristics of winter wheat. *Remote sensing of environment*, *19*(1), 47-61.

Pasquel, D., Roux, S., Richetti, J. *et al.* A review of methods to evaluate crop model performance at multiple and changing spatial scales. *Precision Agric* 23, 1489–1513 (2022). https://doi-org.insu.bib.cnrs.fr/10.1007/s11119-022-09885-4

Paustian, K., Larson, E., Kent, J., Marx, E., and Swan, A.: Soil C Sequestration as a Biological Negative Emission Strategy, Frontiers in Climate, 0, https://doi.org/10.3389/fclim.2019.00008, https://www.frontiersin.org/articles/10.3389/fclim.2019.00008/full, publisher: Frontiers, 2019.

Pique, G., Fieuzal, R., Al Bitar, A., Veloso, A., Tallec, T., Brut, A., Ferlicoq, M., Zawilski, B., Dejoux, J.-F, Gibrin, H., and Ceschia, E.: Estimation of daily CO2 fluxes and of the components of the carbon budget for winter wheat by the assimilation of Sentinel 2-like remote sensing data into a crop model, Geoderma, 376, 114 428, https://doi.org/10.1016/j.geoderma.2020.114428, https://linkinghub.elsevier. com/retrieve/pii/S0016706119321998, 2020a.

Pique, G., Fieuzal, R., Debaeke, P., Al Bitar, A., Tallec, T., and Ceschia, E.: Combining High-Resolution Remote Sensing Prod- ucts with a Crop Model to Estimate Carbon and Water Budget Components: Application to Sunflower, Remote Sensing, 12, 2967, https://doi.org/10.3390/rs12182967, https://www.mdpi.com/2072-4292/12/18/2967, number: 18 Publisher: Multidisciplinary Digital Pub- lishing Institute, 2020b.

Löscher, A., Martimort, P., Jutz, S., Gascon, F., Donlon, C., Manolis, I., and Del Bello, U.: The ESA Sentinel Next-Generation Land & Ocean Optical Imaging Architectural Study, an Overview, EGU General Assembly 2020, Online, 4–8 May 2020, EGU2020-6675, https://doi.org/10.5194/egusphere-egu2020-6675, 2020

Lievens, H., De Lannoy, G. J. M., Al Bitar, A., Drusch, M., Dumedah, G., Franssen, H. J. H., ... & Pauwels, V. R. N. (2016). Assimilation of SMOS soil moisture and brightness temperature products into a land surface model. Remote sensing of environment, 180, 292-304.

Moffat, A. M., Papale, D., Reichstein, M., Hollinger, D. Y., Richardson, A. D., Barr, A. G., ... & Stauch, V. J. (2007). Comprehensive comparison of gap-filling techniques for eddy covariance net carbon fluxes. *Agricultural and Forest Meteorology*, *147*(3-4), 209-232.

Smith, P., Soussana, J.-F., Angers, D., Schipper, L., Chenu, C., Rasse, D. P., Batjes, N. H., van Egmond, F., McNeill, S., Kuhnert, M., Arias-Navarro, C., Olesen, J. E., Chirinda, N., Fornara, D., Wollenberg, E., Álvaro Fuentes, J., Sanz-Cobena, A., and Klumpp, K.: How 920 to measure, report and verify soil carbon change to realize the potential of soil carbon sequestration for atmospheric greenhouse gas removal, Global Change Biology, 26, 219–241, https://doi.org/10.1111/gcb.14815, https://onlinelibrary.wiley.com/doi/abs/10.1111/gcb. 14815, _eprint: https://onlinelibrary.wiley.com/doi/pdf/10.1111/gcb.14815, 2020.

Stöckle, C. O., Donatelli, M., & Nelson, R. (2003). CropSyst, a cropping systems simulation model. *European journal of agronomy*, *18*(3-4), 289-307.

Tomer, S. K., Al Bitar, A., Sekhar, M., Zribi, M., Bandyopadhyay, S., Sreelash, K., ... & Kerr, Y. (2015). Retrieval and multi-scale validation of soil moisture from multi-temporal SAR data in a semi-arid tropical region. Remote Sensing, 7(6), 8128-8153.

Tomer, S. K., Al Bitar, A., Sekhar, M., Zribi, M., Bandyopadhyay, S., & Kerr, Y. (2016). MAPSM: A spatio-temporal algorithm for merging soil moisture from active and passive microwave remote sensing. Remote Sensing, 8(12), 990.

Valero, S.; Arnaud, L.; Planells, M.; Ceschia, E. Synergy of Sentinel-1 and Sentinel-2 Imagery for Early Seasonal Agricultural Crop Mapping. *Remote Sens.* **2021**, *13*, 4891. https://doi.org/10.3390/rs13234891

Zribi, M., Muddu, S., Bousbih, S., Al Bitar, A., Tomer, S. K., Baghdadi, N., & Bandyopadhyay, S. (2019). Analysis of L-band SAR data for soil moisture estimations over agricultural areas in the tropics. *Remote Sensing*, *11*(9), 1122.

---

## Author Comment (AC2)

**Discussion reply on Anonymous Referee #2 comments on submitted paper :**

**AgriCarbon-EO: v1.0.1: Large Scale and High Resolution Simulation of Carbon Fluxes by Assimilation of Sentinel-2 and Landsat-8 Reflectances using a Bayesian approach**

Taeken Wijmer, Ahmad Al Bitar, Ludovic Arnaud, Rémy Fieuzal, and Eric Ceschia

**General comments**

**Perception of the manuscript**

This preprint describes a newly developed method of estimating the soil carbon budget of major crops at high spatial resolution. It addresses a "hot" and very relevant topic in climate research, specifically the field of estimating soil carbon fluxes. The method embodies a novel combination of both high spatial resolution (the intra-field scale) with large spatial coverage (up to 100x100 km) of carbon flux-related variables and their variability. It assimilates high-resolution remote sensing data into an agronomic model and accounts for uncertainties across the processing chain. The paper deals with evaluating the model accuracy at multiple scales, assessing the impact of spatial and temporal scale in the remote sensing-based input data, and integration of multiple models/dataset for a specific crop type and study site (winter wheat in SW France).Although the topic of quantifying soil carbon fluxes holds high significance, the way the proposed methodology is presented in the manuscript offers only little knowledge gain and shows only limited potential to be used for further studies of the soil carbon budget.

We agree with the Referee #2 on the overall view of the scope that we wanted to give to the manuscript. Note however that our objective is not only to assess soil carbon fluxes (as written in the last sentence above), but also to determin the main carbon budget components (biomass, C exported at harvest, GPP, plant respiration and soil respiration). We agree that a better link between the assessment of carbon fluxes (vegetation + soil) and soil carbon budget needs to be presented. We addressed this issue in the following answers by presenting the Net Ecosystem Carbon Budget (NECB) equation and producing additional results that will be added to the manuscript.

**Comment I**:

First, although the paper starts with the need for better quantifications of soil organic carbon, it is too hung up on general carbon flux, data/model integration and comparison to make interesting statements about the variability and accuracy of soil carbon estimation. To me, rather than addressing the estimation of soil carbon fluxes, it seems that the proposed approach is a crop biomass model based on correlations from remote sensing-derived GLAI.

As a matter of fact, the annual carbon budget for croplands that represents the amount of organic C gain or loss depends on the annual $CO_2$ fluxes (with GPP-Rauto conditioning the biomass production) and the lateral fluxes of Carbon as organic amendments (Cimports) and exported at harvest (Cexports) as in the equation 1 (Ceshia et al. 2010, Woodwell and

Whittaker et al., 1968, Chapin et al., 2006) . Except for carbon imports, those variables are estimated by the SAFYE-CO2 model in Agricarbon-EO. We agree that this link is not presented clearly enough in the paper. In order to clarify the relation between soil organic carbon stock changes, $CO_2$ fluxes and biomass, we will add the equation 1 , that connects the different components, into the introduction section of the manuscript with an associated explanation:

$$NECB = \overbrace{\overbrace{GPP \underbrace{- Rauto - Rh}_{Reco}}^{NPP}}^{NEE} + C_{imports} - C_{exports}$$ (eq.1)

ΔNECB is the Net Ecosystem Carbon Budget. It can be divided into two components. First the carbon fluxes as $CO_2$ induced by the biological processes represented by the Gross primary production (GPP) resulting from photosynthesis, autotrophic respiration (Rauto i.e. plant respiration) and heterotrophic respiration (Rh, i.e. soil respiration). The add-up of those fluxes is the Net Ecosystem Exchange (NEE). When the NEE is integrated over a cropping year, it is referred to as the Net Ecosystem Productivity (NEP). The net flux for the plant (GPP-Rauto) is referred to as the  Net Primary production (NPP) which represents the amount of biomass produced. The sum of respiration fluxes (Rauto + Rh) is the ecosystem respiration (Reco) . The Net Ecosystem Carbon Budget also depends carbon fluxes resulting from farming practices, namely the Cimports representing the amount of C brought as organic amendments (e.g. manure, compost) and the Cexports that represent how much C has been exported at harvest (e.g. grain, grain+straw, tubers).

When the crop is harvested, the unharvested plant biomass (litter and roots) are incorporated into the soil, which means ΔNECB = ΔSOC at the end of the cropping year. In the revised version of the manuscript, we will present a ΔSOC variation map to complement the NEP maps over the wheat growing period by considering the following imports and exports conditions:

- In the region of interest, carbon imports are negligible as the fraction of farms practicing animal husbandry is very low. Furthermore the mass of seed carbon for wheat is about 6 to 10 g/m$^2$.
- Cexports are the parts of the plant that are harvested or removed. For wheat, grains are usually the only part of the plant that is exported in the region of interest.

$$C_{exports} = \overbrace{DAM \times HI}^{dry\ Yield} \times Cfrac$$ (eq.2)

HI is the harvest index and Cfrac is the fraction of carbon per unit biomass.

As NEE and NEP (annually cumulated NEE) are computed by Agricarbon-EO, by relying on eq1 and eq2, ΔSOC maps can be computed and added after figure 9 in the manuscript considering the rules mentioned above (no organic amendments, only grain is harvested):

[Figure]

Figure R1: Net Ecosystem Productivity, Carbon exports based on yield and Net Ecosystem Carbon Budget maps between 20161001 and 20171001 over the T31TCJ SENTINEL2 tile and the distribution of those variables.

By highlighting and adding those elements, we hope that the utility of this processing chain for studies regarding soil carbon is now clarified. In the discussion section the limitations related to the soil module are already mentioned but will be also amended. More precisely, alternative solutions include adding more environmental constraints to the Rh terms in SAFYE-CO2, or estimating at high-resolution root and aboveground biomass inputs to the soil with AgriCarbon-EO that will be used as inputs in soil models such as AMG,Daycent or RothC.

The addition of these informations will not lengthen the paper as we are suggesting in the latter answers to remove and reduce parts of the manuscript. The **section 5.3 "Regional scale analysis"** containing **Figure14** and **Figure15** will be removed and the **Figure10** regarding posterior parameter distributions and the analysis related to this figure will be moved to supplementary material.

**Comment II:**

Second, it remains unclear why exactly the study area was chosen and to what extent it has greater relevance for other potential applications in different regions of the world with different crop types, different climatic and soil characteristics.

The study area is used as an application area for the presented method. It was chosen for several reasons that are mentioned across the section "3.1 Study area". We will summarize

them in the updated manuscript at the beginning of section "3 Application in South-West France over Wheat". The reasons are the following:

I. The extensive dataset available from the Space Regional Observatory (OSR - Observatoire Spatial Regional) and the ICOS measurement site of Auradé (FR-Aur). This large evaluation dataset contains multiple variables of interest for carbon stocks, as well as for the verification of the performance and soundness of an agronomic modeling approach. The amount and nature of this dataset corresponds to what is needed to characterize the accuracy/uncertainty of Monitoring Review and Verification(MRV) tools as described in Smith et al. (2010).

II. The spatial variability of pedoclimatic conditions resulting from the slope, aspect, soil properties, and historical land management "Remembrement" policy. This results in high crop growth variability at intra and inter-field scales. This is essential to assess the impact of using high-resolution modeling and assimilation schemes in quantifying the carbon budget components (e.g. biomass, $CO_2$ fluxes), and to assess the ability of the processing chain to reproduce those variations.

III. The area is a dense crop production zone. This is especially true for wheat production (up to 40% of the surface in the simulated area), with a large economic interest (Soft Wheat represents 75% of the Nation soft wheat exports).

These characterics of the study area make the arguments for a benchmark area for both high resolution agronomic modeling exercises and MRV methods. We would also like to recall that the application to study area is to "validate and demonstrate the capabilities of AgriCarbon-EO" (section 3),  and that the focus of the paper is to present the innovative method as mentioned by the referee in his general comment.

**Comment III:**

Furthermore, it is not clear which possible users the presented processing chain has and which questions - apart from comparing/evaluating data and model parameters - it can answer.

The development of the AgriCarbon-EO modeling chain was done based on the needs expressed by the soil carbon communities in several research papers (Smith et al. 2020, Paustian et. al. 2019) or actions like the CIRCASA (Coordination of International Research Cooperation on Soil Carbon Sequestration in Agriculture), the Horizon ORCASA project, and the Eu Commission Expert Group on the implementation of the EU Soil Strategy for 2030…

The current presented approach (AgriCarbon-EO) is a hybrid solution that combines remote sensing, agronomic modeling, and assimilation strategy. It provides a scalable solution that enables the estimation of their main soil carbon budget components as presented and clarified in the first comment (eq.1).  It answers a question related to: How to provide soil carbon budget at regional scale for cropland knowing that farming practices vary between the fields, and that intra-field heterogeneity is present?

Beyond this question the tool can be used to answer more or less specific questions, that were not addressed in this paper, like:

- What trade-off between water consumption and carbon storage is related to the deployment of cover crops?
- How to better define a soil sampling strategy based on the spatial heterogeneity of the carbon budget components?

The current study has limitations presented in the discussion section that open to other questions on how to enhance the approach:

- How to enhance the soil carbon budget by coupling SAFYE-CO2 to process based soil models (AMG, RothC, DayCent) ?
- What type of accuracy can be expected for crops in different regions with different crop rotations and pedoclimatic conditions ?
- **…**

The questions above address many profiles of research users and communities. Clearly there is a multidisciplinary aspect to this tool that can be of interest to the soil, crop modeling and climate impact communities.

We can include those elements in the outlook to illustrate the type of questions that can be answered by Agricarbon-EO for study areas that range from individual sample points to intra field to regional analysis, and may be of interest to agronomic modeling, soil carbon, water management and precision agriculture communities.

**Comment IV:**

Lastly, the paper touches on too many aspects – the discussion of scale (spatial and temporal), inter-comparison of different data/models, different crop-related parameters –so that it is rather difficult to read. The focus on relevant questions within the carbon modeling community and clear answers to those got lost.

We agree that the paper touches many elements. While some analysis are needed to justify driving elements of the method like scalability, accuracy, and high-resolution (figures 11 and 12), we reckon that reducing some elements would enhance the paper. We suggest the following major modifications to the paper to improve the readability:

I.   **Clarifying the link between soil carbon, CO2/carbon fluxes and biomass:**
   A. We will add the NECB equation in the introduction, we will comment on it and make the link with the variables simulated by AgriCarbon-EO (see **comment I**).
   B. We will add the Figure R1 here above that presents the SOC stock change map using this same equation and the hypothesis explained here above.
II.  **Reducing and simplifying parts of the paper:**
   A. Section 2.4 **"Bayesian normalized importance SAmpling using Look out Table - BASALT"** will be fused with section 2.4.2 **"Log-likelihood computations"**.
   B. Section 2.4.1 **"Normalised Importance Sampling and Look-up table"** will be moved to supplementary material.
   C. **Figure 6** and **figure 5** will be fused together as they treat the same years and field.

D. **Figure10** regarding posterior parameter distributions and the analysis related to this figure will be moved to supplementary material.
E. **section 5.3 "Regional scale analysis"** containing **Figure14** and **Figure15** will be removed.

III. **Enhancements:**
A. The "Discussion" section will be amended based on feedbacks from the discussion process.
B. Time series corresponding to the pixel wise validations over the ESU biomass dataset will be added to the supplemental material.
C. We leave it to the editor to decide whether or not part of or all of the equations of the Agronomic model should be put in the supplemental.

IV. **Grammatical and typo:**

The paper will be spell checked by an external service and several native speakers with knowledge in the domain.

We will also integrate the eventual comments provided by the Scientific Editor.

**General conclusion:**

Overall, the paper holds certain potential for wide scientific interest, but needs to be substantially revised for better informative value on soil carbon fluxes and scalability of the method for other regions. I recommend performing major revisions before considering for publication in GMD.

I suggest to address the following major points:

1. the many spelling and grammatical errors in the manuscript (I can't list all of these)

As mentioned above, the paper will be spell checked by an external service and several native speakers with knowledge in the domain.

2. missed topic: the discrepancy between the introduction of the topic of soil carbon fluxes and the presentation of the approach to a crop model that maps biomass and NPP instead of soil carbon.

As mentioned In **comment I.** we will add the NECB equation (eq1) in the introduction and we will use it as a common thread through the manuscript to highlight the relevance of each specific element of analysis to the carbon budget assessment and we will provide a SOC stock change with the NEP and Yield maps that allowed to compute it in section 5.1 **"Large scale simulations".** We will also discuss in more detail how besides providing evaluations of the SOC stock changes, above and below ground biomass maps from ACEO can be used to enhance the spatial representativity of existing soil models.

Clearer designation and focussed answering of relevant research questions/objectives

The main objectives that are:

I. To present a method to represent high resolution crop variability at large scale and its impact on the carbon cycle.

II. To demonstrate its validity through different evaluation exercises for the different components of the carbon budget.

III. To showcase its spatial capabilities through an application over winter wheat to estimate crop characteristics and the resulting SOC stock changes over a cropping year.

Those objectives will be put forward more clearly.

3.  Justification for the selection of the study area in terms of what we can learn from it for other regions

We provided an extensive answer to this remark in **comment II.**

**Specific comments**

**Lines73ff (objectives)**
Please be more specific about the objectives here. What are specifically "the outputs" you aim to check for accuracy and coherence?

The regional analysis in the paper section 5.3 corresponds to checking if outputs present a statistical behavior with respect to altitude, aspect (exposition), and soil properties that is consistent with our expectations.

This section (5.3) will be removed in the updated manuscript to shorten the paper. So the following comment along with the mentions to it in other sections will be removed.

 Instead of throwing in buzz words like "multi-scale validation", name the research questions precisely, e.g. what is the impact of (a) spatial resolution and (b) temporal resolution of the input data?

We agree that we can better show the research question. Still our intention was not to use the "multi-scale validation" because of its popularity. It was used because we argue that a proper presentation and assessment of a method that combines intra-field to regional scales requires validation and comparison at several resolutions.

"verifying the coherency of the outputs through intra-field as well as regional analysis" sounds spongy too me. What exactly is meant by "coherency", what is "intra-field and regional analysis"?

The intra-field analysis designs the evaluation of the models against spatialised aboveground biomass data and high resolution combine harvester yield maps, especially the ability of the model to retrieve intra-field resolution yield patterns. The regional analysis (that we propose to remove from the manuscript) aims at verifying that the model is able to represent the expected patterns when confronted to covariables with known effects on retrieved patterns. For example, we expect plants to grow faster and earlier on south-facing slopes in our region of interest. We  have shown that this is the case in the simulations, lending additional credibility to the overall recovery of vegetation dynamics.

**Line 112/land cover map**

So, only the border extents of the shape file are used here to download the remote sensing and weather forcing data? Please describe how the land cover information is used, if at all, and how the land cover categories should look like.

Yes, in the first step. The extents of the shapefile are used to prepare the remote sensing data and the weather forcing (either they are in the database or they are downloaded).

Then the content of the shape file (land cover map) is used to run the modeling chain over all the pixels covered by the crop of interest. In the current application these are wheat fields.

The modeling chain will use the same vegetation prior parameters file for wheat for each of the pixels.

If the user aims at making estimates for maize, the user will need to provide a shapefile with identified maize fields and the prior parameters of maize.

We discuss limitations related to the land cover map in the Discussion section 6.2 of the paper.

The shapefile used for the simulation is provided in 'ACEO/EXPERIMENT/template/sim_shape/' folder in the zenodo code archive.

**Figure 4**

**What do the different colours mean? Why not use different colours for each dataset groups: input,** validation and regional datasets?

Thank you for the suggestion, the figure will be updated. Also the dataset used in section 5.3 will be removed from the figure.

**Line 571f: "So high resolution allows more accurate estimates of the mean DAM values at field scale which enables more accurate field scale estimates of SOC changes by soil models in the perspective of monitoring SOC stock changes"**

SOC was not really analyzed and assessed here, so how would you know the effect of high-spatial resolution input data on soil organic carbon modeling? I think that this conclusion is simply too far-fetched here.

We consider that more accurate representation of DAM and yield spatial variabilities allow more accurate estimates of the biomass that returns to the soil and of its spatial variability. As the amount of biomass that is returned to the soil is a key input for soil models, reducing its uncertainty or improving its spatial variability will improve the results of soil models.

**Line 665: "an intra-field scale quantification of the DAM and of the biomass that returns to the soil is needed for accurate monitoring of soil organic carbon stock changes"**

Again, this conclusion seems far-fetched here. It is not clear to me how you assessed soil organic carbon with the proposed modeling chain, as it was never computed or derived from the output variables.

As we mentioned above the NECB equation and harvest equations, the carbon from the unharvested dry biomass (NPP or (DAM + DBM) * cfrac ) * (1 - HI) at the time of harvest is the main carbon input to agricultural soils (Soussana et al., 2019, Minasny et al.,2022). As such we argue that it is important to take into account the spatial variability of the biomass that returns to the soil to provide a spatially explicit representation of the evolution of soil organic carbon stocks.

**Technical corrections (selection, please perform comprehensive spelling correction!)**

- Line 8: misspelling – "radiative transfer"
- Line 10/11: misspelling - "a land cover maps"
- Line 14: misspelling - "against"
- Line 103: misspelling – "corpping"
- Line 111: misspelling - "Landcover"
- Line 165: misspelling – "In Contrast"
- Line 268 (and every other occurrence): misspelling "lykelyhoods"
- Line 304: misspelling- "dry Biomass"
- Line 322: "in the field 27" What is 27?
- Line 325: misspelling – "is applied over a for winter wheat"
- Table 3: "Quantify spatial and variability" What variability?
- Line 353: "Dry Aboveground biomass, DAM measurements" Please present abbreviations in a uniform manner.
- Line 453: misspelling - "respiration"
- etc.

Will be corrected.

Bibliography:

E. Ceschia, P. B eziat, J. F. Dejoux, M. Aubinet, C. Bernhofer, B. Bodson, N. Buchmann, A. Carrara, P. Cellier, P. Di Tommasi, J. A. Elbers, W. Eugster, T. Gr¨unwald, C. M. J. Jacobs, W. W. P. Jans, M. Jones, W. Kutsch, G. Lanigan, E. Magliulo, O. Marloie, E. J. Moors, C. Moureaux, A. Olioso, B. Osborne, M. J. Sanz, M. Saunders, P. Smith, H. Soegaard, M. Wattenbach, Management effects on net ecosystem carbon and GHG budgets at European crop sites, Agriculture, Ecosystems & Environment 139 (3) (2010) 363–383. doi:10.1016/j.agee.2010.09.020. URL https://www.sciencedirect.com/science/article/pii/S0167880910002537

F. S. Chapin, G. M. Woodwell, J. T. Randerson, E. B. Rastetter, G. M. Lovett, D. D. Baldocchi, D. A. Clark, M. E. Harmon, D. S. Schimel, R. Valentini, C. Wirth, J. D. Aber, J. J. Cole, M. L. Goulden, J. W. Harden, M. Heimann, R. W. Howarth, P. A. Matson, A. D. McGuire, J. M. Melillo, H. A. Mooney, J. C. Neff, R. A. Houghton, M. L. Pace, M. G. Ryan, S. W. Running, O. E. Sala, W. H. Schlesinger, E.-D. Schulze, Reconciling Carbon-cycle Concepts, Terminology, and Methods, Ecosystems 9 (7) (2006) 1041–1050. doi:10.1007/s10021-005-0105-URL https://doi.org/10.1007/s10021-005-0105-7

G. M. Woodwell, R. H. Whittaker, Primary Production in Ter-restrial Ecosystems, American Zoologist 8 (1) (1968) 19–30.24 doi:10.1093/icb/8.1.19. URL https://academic.oup.com/icb/article-lookup/doi/10.1093/icb/8.1.19

J.-F. Soussana, S. Lutfalla, F. Ehrhardt, T. Rosenstock, C. Lamanna, P. Havlʹık, M. Richards, E. L. Wollenberg, J.-L. Chotte, E. Torquebiau, P. Ciais, P. Smith, R. Lal, Matching policy and science: Rationale for the '4 per 1000 - soils for food security and climate' initiative, Soil and Tillage Research 188 (2019) 3–15. doi:10.1016/j.still.2017.12.002. URL https://www.sciencedirect.com/science/article/pii/S0167198717302271

B. Minasny, D. Arrouays, R. Cardinael, A. Chabbi, M. Farrell, B. Henry, L.-S. Koutika, J. K. Ladha, A. B. McBratney, J. Padarian, M. Romʹan Dobarco, C. Rumpel, P. Smith, J.-F. Soussana, Current NPP cannot predict future soil organic carbon sequestration potential. Comment on "Photosynthetic limits on carbon sequestration in croplands", Geoderma 424 (2022) 115975. doi:10.1016/j.geoderma.2022.115975. URL https://linkinghub.elsevier.com/retrieve/pii/S0016706122002828

---

## Author Comment (AC3)

Dear Referee1,

In addition to our clarifications that were sent in the previous reply, and in order to specify our answers to the two main criticisms, that are the perceived disconnection of the manuscript from the subject of agro-ecosystem carbon budgets as well as its length.

First, concerning the relevance of the study regarding carbon budgets, we provide the following elements:

The annual carbon budget for croplands that represents the amount of organic C gain or loss depends on the annual $CO_2$ fluxes (with GPP-Rauto conditioning the biomass production) and the lateral fluxes of carbon as organic amendments (C imports) and exported at harvest (C exports) as in the equation 1 (Ceshia et al. 2010, Woodwell and Whittaker et al., 1968, Chapin et al., 2006). Except for carbon imports, those variables are estimated by the SAFYE-CO2 model in Agricarbon-EO. We agree that this link is not presented clearly enough in the paper. In order to clarify the relation between soil organic carbon stock changes, $CO_2$ fluxes and biomass, we will add the equation 1 , that connects the different components, into the introduction section of the manuscript with an associated explanation:

$$NECB = \overbrace{\underbrace{\overbrace{GPP \underbrace{- Rauto}_{Reco} - Rh}^{NPP}}_{}}^{NEE} + C_{imports} - C_{exports} \qquad \text{(eq.1)}$$

ΔNECB is the Net Ecosystem Carbon Budget. It can be divided into two components. First the carbon fluxes as $CO_2$ induced by the biological processes represented by the Gross primary production (GPP) resulting from photosynthesis, autotrophic respiration (Rauto i.e. plant respiration) and heterotrophic respiration (Rh, i.e. soil respiration). The add-up of those fluxes is the Net Ecosystem Exchange (NEE). When the NEE is integrated over a cropping year, it is referred to as the Net Ecosystem Productivity (NEP) . The net flux for the plant (GPP-Rauto) is referred to as the Net Primary production (NPP) which represents the amount of biomass produced. The sum of respiration fluxes (Rauto + Rh) is the ecosystem respiration (Reco) . The Net Ecosystem Carbon Budget also depends carbon fluxes resulting from farming practices, namely the Cimports representing the amount of C brought as organic amendments (e.g. manure, compost) and the Cexports that represent how much C has been exported at harvest (e.g. grain, grain+straw, tubers).

When the crop is harvested, the unharvested plant biomass (litter and roots) are incorporated into the soil, which means ΔNECB = ΔSOCat the end of the cropping year. In the revised version of the manuscript, we will present a ΔSOC variation map to complement the NEP maps over the wheat growing period by considering the following imports and exports conditions:

- In the region of interest, carbon imports are negligible as the fraction of farms practicing animal husbandry is very low. Furthermore the mass of seed carbon for wheat is about 6 to 10 g/m$^2$.
- Cexports are the parts of the plant that are harvested or removed. For wheat, grains are usually the only part of the plant that is exported in the region of interest.

$$C_{exports} = \overbrace{DAM \times HI}^{\text{dry Yield}} \times Cfrac \qquad \text{(eq.2)}$$

HI is the harvest index and Cfrac is the fraction of carbon per unit biomass.

As NEE and NEP (annually cumulated NEE) are computed by Agricarbon-EO, by relying on eq1 and eq2, ΔSOC maps can be computed and added after figure 9 in the manuscript considering the rules mentioned above (no organic amendments, only grain is harvested):

[Figure]

Figure R1: Net Ecosystem Productivity, Carbon exports based on yield and Net Ecosystem Carbon Budget maps between 20161001 and 20171001 over the T31TCJ SENTINEL2 tile and the distribution of those variables.

By highlighting and adding those elements, we hope that the utility of this processing chain for studies regarding soil carbon is now clarified. In the discussion section the limitations related to the soil module are already mentioned but will be also amended. More precisely, alternative solutions include adding more environmental constraints to the Rh terms in SAFYE-CO2, or estimating at high-resolution root and aboveground biomass inputs to the

soil with AgriCarbon-EO that will be used as inputs in soil models such as AMG,Daycent or RothC.

Second, concerning the length of the paper. The additions of eq.1 to the introduction and the carbon budget maps will not lengthen the paper as we suggest below to remove and reduce parts of the manuscript. While some analysis is needed to justify driving elements of the method like scalability, accuracy, and high-resolution (figures 11 and 12), we reckon that reducing some elements would enhance the manuscript.

In summary, we suggest the following major modifications to the paper to enhance the readability and focus on the main research questions:

I. **Clarifying the link between soil carbon, carbon fluxes and biomass:**
   A. We will add the change of soil carbon and net ecosystem carbon budget in the introduction with the NECB equation presented in **comment I**.
   B. We will add the Figure R1 here above that presents the SOC stock change map using this same equation and the hypothesis explained here above.

II. **Reducing and simplifying parts of the paper:**
   A. Section 2.4 **"Bayesian normalized importance SAmpling using Look out Table - BASALT"** will be fused with section 2.4.2 **"Log-likelihood computations"**.
   B. Section 2.4.1 **"Normalised Importance Sampling and Look-up table"** will be moved to supplementary material.
   C. **Figure 6** and **figure 5** will be fused together as they treat the same years and field.
   D. **Figure10** regarding posterior parameter distributions and the analysis related to this figure will be moved to supplementary material.
   E. **section 5.3 "Regional scale analysis"** containing **Figure14** and **Figure15** will be removed.

III. **Enhancements:**
   A. The "Discussion" section will be amended based on feedback from the discussion process.
   B. Time series corresponding to the pixel wise validations over the ESU biomass dataset will be added to the supplemental material.
   C. We leave it to the editor to decide whether or not part of or all of the equations of the Agronomic model should be put in the supplemental.

IV. **Grammatical and typo:**

   The paper will be spell checked by an external service and several native speakers with knowledge in the domain.

We will also integrate the eventual comments provided by the Scientific Editor.

Bibliography:

E. Ceschia, P. B eziat, J. F. Dejoux, M. Aubinet, C. Bernhofer, B. Bodson, N. Buchmann, A. Carrara, P. Cellier, P. Di Tommasi, J. A. Elbers, W. Eugster, T. Gr¨unwald, C. M. J. Jacobs, W. W. P. Jans, M. Jones, W. Kutsch, G. Lanigan, E. Magliulo, O. Marloie, E. J. Moors, C. Moureaux, A. Olioso, B. Osborne, M. J. Sanz, M. Saunders, P. Smith, H. Soegaard, M. Wattenbach, Management effects on net ecosystem carbon and GHG budgets at European crop sites, Agriculture, Ecosystems & Environment 139 (3) (2010) 363–383. doi:10.1016/j.agee.2010.09.020. URL https://www.sciencedirect.com/science/article/pii/S0167880910002537

F. S. Chapin, G. M. Woodwell, J. T. Randerson, E. B. Rastetter, G. M. Lovett, D. D. Baldocchi, D. A. Clark, M. E. Harmon, D. S. Schimel, R. Valentini, C. Wirth, J. D. Aber, J. J. Cole, M. L. Goulden, J. W. Harden, M. Heimann, R. W. Howarth, P. A. Matson, A. D. McGuire, J. M. Melillo, H. A. Mooney, J. C. Neff, R. A. Houghton, M. L. Pace, M. G. Ryan, S. W. Running, O. E. Sala, W. H. Schlesinger, E.-D. Schulze, Reconciling Carbon-cycle Concepts, Terminology, and Methods, Ecosystems 9 (7) (2006) 1041–1050. doi:10.1007/s10021-005-0105-URL https://doi.org/10.1007/s10021-005-0105-7

G. M. Woodwell, R. H. Whittaker, Primary Production in Ter-restrial Ecosystems, American Zoologist 8 (1) (1968) 19–30.24 doi:10.1093/icb/8.1.19. URL https://academic.oup.com/icb/article-lookup/doi/10.1093/icb/8.1.19

---

## Referee Report (RR1)

**General comments**

Concerning the content and the structure, the manuscript is improved, its structure is more concise and objectives are more comprehensive now. The connection between soil organic carbon stocks and the components of the carbon budget is now clearer to see. The authors addressed my previous comments and e.g. added an improved justification of the study area for applying the presented approach.

However, there are still minor points to improve. Some sections are still very lengthy, e.g. the first part of the discussion is too descriptive and repeats the findings from the previous section instead of placing them in the wider research context. E.g., the finding that the number of image acquisitions leads to reduced accuracy, but specific sensors increase the accuracy, needs to be discussed more clearly. What are the limitations and also steps forward to improve the approach? Shall users focus on careful image pre-processing (cloud filtering) or sensor selection, what advices could you give based on your findings? Further, the abstract lacks the most important research findings.

Unfortunately, the manuscript still needs to be linguistically revised. It is not so much the spelling mistakes, but often sentence structure and transitions as well as few grammatical errors that make it difficult to read. Please check the references and short citations in the next, as they are numerous redundant brackets and inconsistent formatting!

**Specific comments**

**Abstract**
**Lines 15ff**

What are the main findings here? You rather describe what was done but I am missing an interpretation/discussion of what we can learn from the findings. Please add some insights from the discussion (on data comparison, the difference between pixel- and field-scale and the impact of remote sensing images). How are your results, e.g. the accuracy values, the bias or the affected percentage of uncertainty, to be evaluated? Are the values high, surprising, low - what is new? What are the ways forward?

**Line 25 – "soil organic carbon (SOC) storage has the potential to remove 0.6 to 9.3 Gt CO2 yr-1"**
At which scale – global, Europe-wide or regional? Please specify.

**Line 34ff – "..shows the importance"**
The equation rather shows the linkage between those components and, as a consequences, it is important to quantify them in order to better understand the effect of changing farming practices on the carbon budget. Please rephrase that accordingly and provide more information on the overall objective of the quantification.

**Discussion**
**Section 4.1** Instead of an almost 1-page descriptive text about the results from other studies, a figure or table with an overview of the selected studies their results in $R^2$, RMSE, site characteristics etc. compared against the metrics of this study would be useful here.
In addition, please highlight more clearly what we can learn from that. You write that performances "are close to or better than existing state-of-the-art evaluations" but what does that mean? How high is the agreement or the improvement? Does that mean with less input variables you can yield better results? Please provide more information on what we can learn from the experiment!

**Line 548ff**: **"The same approaches may be penalized when applied to areas with high spatial variability, such as the hilly countryside in southwestern France"**
How about your approach applied to an area with low spatial variability? Please provide information on the scalability of your approach to other regions!

**Technical corrections (please check again! This is just a selection)**

- Line 26: remove ")"
- Line 28: remove " )" and redundant spaces and brackets in the references
- Line 30: remove brackets in the references
- Line 46: insert space in "theregional"
- Line 47: change references into readable format
- Line 47ff: insert hyphenation and commas where needed
- Line 58/59: remove brackets in reference
- Line 561ff: wrong sentence structure! please change to "our approach will benefit from improvements…"
- Figure 5: typo in the axis labels
- Figure 6A: Please use different colour scaling for the different maps, as the spatial differences in the middle plot are not visible at all.

---

## Author Response (AR2)

**Reply to report 1 from Referee2**

**General comments**

Concerning the content and the structure, the manuscript is improved, its structure is more concise and objectives are more comprehensive now. The connection between soil organic carbon stocks and the components of the carbon budget is now clearer to see. The authors addressed my previous comments and e.g. added an improved justification of the study area for applying the presented approach.

However, there are still minor points to improve. Some sections are still very lengthy, e.g. the first part of the discussion is too descriptive and repeats the findings from the previous section instead of placing them in the wider research context. E.g., the finding that the number of image acquisitions leads to reduced accuracy, but specific sensors increase the accuracy, needs to be discussed more clearly. What are the limitations and also steps forward to improve the approach? Shall users focus on careful image pre-processing (cloud filtering) or sensor selection, what advices could you give based on your findings?

Further, the abstract lacks the most important research findings.
Unfortunately, the manuscript still needs to be linguistically revised. It is not so much the spelling mistakes, but often sentence structure and transitions as well as few grammatical errors that make it difficult to read. Please check the  references and short citations in the next, as they are numerous redundant brackets and inconsistent formatting!

We thank Referee2 for his constructive and detailed comments that improved the readability of the paper  and teh presentation of the findings.  We modified the paper according to his recommendations. The first section of the discussion (section 4.1) was enhanced with a better discussion of the results and a clearer transition between the findings and limitations. Discussion on the impact of remote sensing data was also enhanced by adding the recommendations on selection of complementary remote sensing sources and the .  The abstract was modified and more insights about the results, discussion and recomendations  were added. The paper was revised. The citations formatting was corrected. At the final stage of publishing, we will also recheck with the publishing editor if incompatible formatting is still present.

Point-by-point answers on the specific points are provided below.

**Specific comments**

**Abstract Lines 15ff**

What are the main findings here? You rather describe what was done but I am missing an interpretation/discussion of what we can learn from the findings. Please add some insights from the discussion (on data comparison, the difference between pixel- and field scale and the impact of remote sensing images). How are your results, e.g. the accuracy values, the bias or the affected percentage of

uncertainty, to be evaluated? Are the values high, surprising, low - what is new? What are the ways forward?

The abstract has been updated to better highlight the main findings quantiativly and summarise the elements of the discussion.

**Line 25 – "soil organic carbon (SOC) storage has the potential to remove 0.6 to 9.3 Gt CO2 yr-1"**

At which scale – global, Europe-wide or regional? Please specify.

It is at the global scale. We specified it in the text.

**Line 34ff – "..shows the importance"**

The equation rather shows the linkage between those components and, as a consequences, it is important to quantify them in order to better understand the effect of changing farming practices on the carbon budget. Please rephrase that accordingly and provide more information on the overall objective of the quantification.

Agree, modified to "shows the linkage". The corresponding paragraph has also been enhanced.

**Discussion**
**Section 4.1** Instead of an almost 1-page descriptive text about the results from other studies, a figure or table with an overview of the selected studies their results in $R_2$, RMSE, site characteristics etc. compared against the metrics of this study would be useful here.

In addition, please highlight more clearly what we can learn from that. You write that performances "are close to or better than existing state-of-the-art evaluations" but what does that mean? How high is the agreement or the improvement? Does that mean with **less input** variables you can yield better results? Please provide more information on what we can learn from the experiment!

In the submission after first review process, the section "4.1" was added following the request from the previous review. In the current submission, we have rewritten this section based on the additionnal recommendations of the reviewer. We added the scores for the AgriCarbon-EO in the text of this section, and added conclusions and recommendations. We didn't add a table to this section. Because there is a high diversity in the conditions of each study making it difficult to add to a concise table that takes into account the conditions (resolution, reference, area of application…) and results (stats for each variable), knowing that not all variables are covered by each of the studies. The paragraphe is now mentions:

A Comparison with previous iterative retrieval with the SAFYE-CO2 model that is included in AgriCarbon-EO, showing that the assimilation scheme doesn't degrade (enahnces in some cases) retrieval values. Comparison with other studies on NEE and GPP showing that it is possible to have similar to better results considering lesser inputs and constrains. A comparison for Reco pointing the

need for a better soil module in AgriCarbon=-EO. A comparison for biomass and yield. A summary of the conclusions and the need for more comparative studies in the future.

**Line 548ff**: **"The same approaches may be penalized when applied to areas with high spatial variability, such as the hilly countryside in southwestern France**"
How about your approach applied to an area with low spatial variability? Please provide information on the scalability of your approach to other regions!

We don't see an issue with the application of the approach to a lower spatial variability. Generally, the challenges arise in high spatial variability. Other impacts may arise in other regions, like for example a greater presence of clouds. These are discussed in the relevant discussion sections of the manuscript. Considering the genericity of the input data in AgriCarbon-EO, large-scale and multi-site testing of the chain will provide more insights into the robustness of the approach across regions and ecosystems. In this sense the comments by the reviewer are valuable, but certainly, this is beyond the scope of this paper where we presented and validated the approach at a regional scale.

**Technical corrections (please check again! This is just a selection)**

- Line 26: remove ")"

  Done.

- Line 28: remove " )" and redundant spaces and brackets in the references

  Done.

- Line 30: remove brackets in the references

  Done.

- Line 46: insert space in "theregional"

  Done.

- Line 47: change references into readable format

  The reference to the studies were added.

- Line 47ff: insert hyphenation and commas where needed

  Done.

- Line 58/59: remove brackets in reference

Modified, we will also recheck during the publishing phase with the publishing editor for all requirements.

- Line 561ff: wrong sentence structure! please change to "our approach will benefit from

  Improvements..."

  The corresponding paragraph was modified.

- Figure 5: typo in the axis labels

  Corrected.

[Figure]

Figure 6A: Please use different colour scaling for the different maps, as the spatial differences in the middle plot are not visible at all.

Modified. Each variable has it's own colour scaling and colour bar which matches the main color in the histogram.

[Figure]

**Reply to Report2 from Referee3**

We would like to thank the Reviewer for their constructive comments that contributed to improving our manuscript. We updated the figures and the text as requested. Furthermore, we also provide point-by-point answers to the raised comments.

Specifically:

Table 2: How was the range of priors defined? Were they empirically adopted?

The baseline for the determination of the priors was based on the previous implementation of SAFYE-CO2, namely in **Pique et al. (2020a)**. In fact, the BASALT - Bayesian approach - in AgriCarbon-EO requires statistical distributions for the priors in contrast to initial values and range in the iterative retrievals implemented in **Pique et al. (2020a)**. Consequently, we added a mean and standard deviation to the minimum-maximum of the range of the parameters. The value of the standard deviation for the priors is based on a sensibility analysis. The minimum and maximum values are based on physical limits. The priors are generated based on a Gaussian distribution that respects the conditions for the mean and standard deviation and constrained to the minimum and maximum values. The fixed parameters for the wheat crops were not modified from the baseline in this study.

Line 26: The ")" should be removed.
Done.
Table 5: "R2, 2" should be in superscript.
Modified.
Figure 4: There are no y-tick values.
The figure has been corrected as presented below.

[Figure]

Figure 7: The numbers/digits in the wind rose plot should be enlarged or removed, as they are currently too small for readers.

[Figure]

The figure has been regenerated with bigger fonts as presented below.

Figure 9: There is an incomplete display of text on the images; "images" needs attention.

Figure 9 was regenerated as presented below.

[Figure]

---

## Author Response (AR3)

Dear Editors,

You can find below some corrections we made to the document and details on the LateX file.

A compilation error seems to have occurred in the previous version submitted before acceptance. Figures 6 and 7 were respectively switched with another figure and were missing. This was corrected in this version to correspond to the answers given to the minor revision.

Furthermore, the TOSCA POLYPHEME project financed by the CNES was added to the funding as it allows to pay the publication fees.

Finally in the technical end of the LateX file provided this .tex document was compiled using pdfLateX and the 2020 version of TeX Live on the Overleaf platform